# Ecological niche modelling does not support climatically-driven dinosaur diversity decline before the Cretaceous/Paleogene mass extinction

Alfio Alessandro Chiarenza [1], Philip D. Mannion [1,2], Daniel J. Lunt [3], Alex Farnsworth [3], Lewis A. Jones [1], Sarah-Jane Kelland[4] & Peter A. Allison [1]

In the lead-up to the Cretaceous/Paleogene mass extinction, dinosaur diversity is argued to have been either in long-term decline, or thriving until their sudden demise. The latest Cretaceous (Campanian–Maastrichtian [83–66 Ma]) of North America provides the best record to address this debate, but even here diversity reconstructions are biased by uneven sampling. Here we combine fossil occurrences with climatic and environmental modelling to quantify latest Cretaceous North American dinosaur habitat. Ecological niche modelling shows a Campanian-to-Maastrichtian habitability decrease in areas with present-day rock-outcrop. However, a continent-wide projection demonstrates habitat stability, or even a Campanian-to-Maastrichtian increase, that is not preserved. This reduction of the spatial sampling window resulted from formation of the proto-Rocky Mountains and sea-level regression. We suggest that Maastrichtian North American dinosaur diversity is therefore likely to be underestimated, with the apparent decline a product of sampling bias, and not due to a climatically-driven decrease in habitability as previously hypothesised.

[1] Department of Earth Science and Engineering, Imperial College London, South Kensington Campus, London SW7 2AZ, UK. [2] Department of Earth Sciences, University College London, Gower Street, London WC1E 6BT, UK. [3] School of Geographical Sciences, University of Bristol, University Road, Bristol BS8 1SS, UK. [4] Getech, Elmete Hall, Elmete Lane, Leeds LS8 2LJ, UK. Correspondence and requests for materials should be addressed to A.A.C. (email: a.chiarenza15@gmail.com)

Reconstruction of the palaeodiversity of Mesozoic dinosaurs has a long tradition in palaeontology, with a growing number of studies over the last 40 years[1–5]. However, many aspects of their macroevolutionary trajectory remain contentious. In particular, a number of contrasting interpretations have been proposed regarding the diversity trends of dinosaurs in the lead-up to the Cretaceous/Paleogene (K/Pg) mass extinction, 66 million years ago (Ma). These can be simplified into two end-member scenarios: a sudden extinction; or a gradual decline. A recent review argued that there is little evidence for a global, long-term decline[5]. Yet, these authors concluded that there was a latest Cretaceous (Campanian–Maastrichtian; ~83–66 Ma) decrease in the diversity of large-bodied herbivores (primarily ceratopsid and hadrosaurid ornithischian dinosaurs), at least in North America[5,6]. In contrast, Sakamoto et al.[7] found evidence for a long-term (~40 million years) global decline of speciation rate in dinosaurs that began in the mid-Cretaceous, with the exception of ceratopsids and hadrosaurids, which apparently maintained a high diversification rate throughout the Late Cretaceous. One purported cause of this apparent decline has been linked to climatic drivers and habitat degradation[1,8]. Choosing between these competing hypotheses, as well as the potential effects of environmental and tectonic processes on long-term diversity trends, remains a central goal of studies on dinosaur macroevolution and macroecology.

Fossils preserved in sedimentary rocks provide an invaluable record of life on Earth that has driven our understanding of macroevolutionary patterns, associated processes, and biodiversity through time. Early attempts to determine deep time diversity dynamics were largely based on simple counts of the numbers of species (or higher taxa) in each time interval[9,10]. However, the extent to which these raw data have been biased by preservation and sampling artefacts has long been debated (e.g. refs. [9,11,12]). Biases include the incomplete preservation of delicate bones or soft-bodied animals, low preservation potential of some biotopes, erosion of fossil-bearing sedimentary rocks, and incomplete sampling by palaeontologists[13], which could lead to erroneous inferences, especially when compounded over geologic timescales.

Statistical methods developed to mitigate these biases typically employ subsampling (e.g. ref. [12]) or modelling approaches (e.g. ref. [11]). Others have attempted to utilise information on the evolutionary interrelationships of a fossil group in reconstructing palaeodiversity, via the inference of phylogenetic ghost lineages (e.g. ref. [14]), morphological disparity (e.g. ref. [6]), and birth-death models (e.g. ref. [7]). Despite the widespread application of these techniques to a large range of fossil organisms, these methods are still heavily constrained by their inability to deal with the absence of data, especially when the spatial distribution of the fossil record in a particular time interval is strongly heterogeneous[15,16]. Any palaeobiological investigation needs to take into account the completeness of the data set. If the primary data that comprise the fossil record, for example, are spatially variable in completeness, then any attempt to extract a meaningful signal from this biased data set will tend to deliver a view of the past that is artefactual. This is the case with the North American dinosaur diversity record, which is skewed towards better preserved areas.

Currently, North America provides the best available sampled, accurately dated, and stratigraphically continuous record of latest Cretaceous dinosaurs[5], and shows a decline in the numbers of genera and species from the Campanian to the Maastrichtian (Fig. 1). Taken at face value, this record implies a diversity zenith during the middle–late Campanian (~78–72 Ma), a decline in the early Maastrichtian (72–69 Ma), and a nadir in the late Maastrichtian (69–66 Ma). In the Campanian, exceptionally productive fossil localities from the Western Interior Basin (WIB), extending

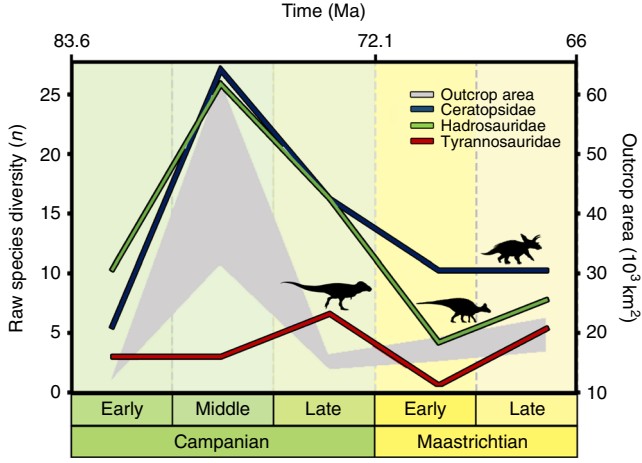

**Fig. 1** Raw diversity trends for the three clades of dinosaurs in this study plotted against outcrop exposure area. The plot shows the apparent correlation of this sampling proxy with diversity curves for these clades of dinosaurs (Ceratopsidae, Hadrosauridae, and Tyrannosauridae). Tyrannosauridae silhouette by Jack Mayer Wood (CC BY 3.0 license: https://creativecommons.org/licenses/by/3.0/ CC BY 3.0); Hadrosauridae silhouette by Pete Buchholz (under CC BY-SA 3.0 license: https://creativecommons.org/licenses/by-sa/3.0/); and Ceratopsidae silhouette by Mariana Ruiz (modified by T. Michael Keesey) under the Public Domain Mark 1.0

along a large latitudinal belt (ranging from Canada to Mexico), expose extensive, and fossil-rich sedimentary successions (Fig. 1). In the Maastrichtian, on the other hand, exposures are smaller and less extensive, with optimal preservation only met in localised areas, such as the Hell Creek Formation in Montana (and lateral equivalents in Alberta, Wyoming, and the Dakotas). These relatively productive Maastrichtian localities occupy a restricted latitudinal belt (~40–50°), whilst sites at higher and lower latitudes do not meet the same ideal preservation or sampling criteria (i.e. they are generally remote places, far away from research centres, and are characterised by climatic extremes).

Furthermore, there is also a major longitudinal bias: nearly all these dinosaur-bearing localities are located on the western side of the continent, where sediments have accumulated in the WIB (Fig. 2a, b). This western subcontinent, Laramidia, stretching from present-day Alaska to Mexico, was separated from the eastern landmass, Appalachia, by the epicontinental Western Interior Seaway (WIS, Fig. 2c–f). Despite forming approximately two-thirds of present-day North America, this eastern subcontinent has a considerably poorer fossil record (e.g. ref. [17]), making the dinosaurian record reliant on Laramidian occurrences[18]. One of the reasons for the poorer sampling of Appalachian localities might be attributable to geological biases: the small number of latest Cretaceous dinosaur-bearing localities in this area (primarily Mississippi, Alabama, and New Jersey) are predominantly represented by marine depositional settings. In addition to most dinosaurs living more inland[19], these marine palaeoenvironments tend to represent unsuitable taphonomic conditions for dinosaur preservation (e.g. due to transportation and disarticulation). These factors result in a poor Appalachian terrestrial vertebrate fossil record, which contrasts starkly with the more suitable fluvial-floodplain settings that characterise most western North American deposits[18,20]. Furthermore, much of the potentially preservable Cretaceous terrestrial sedimentary rock record from Appalachia is thought to have been subsequently eroded[21]. Consequently, due to little fossil material from this region, we have scant means to assess the taxonomic composition of Appalachian dinosaur communities. Was this region truly

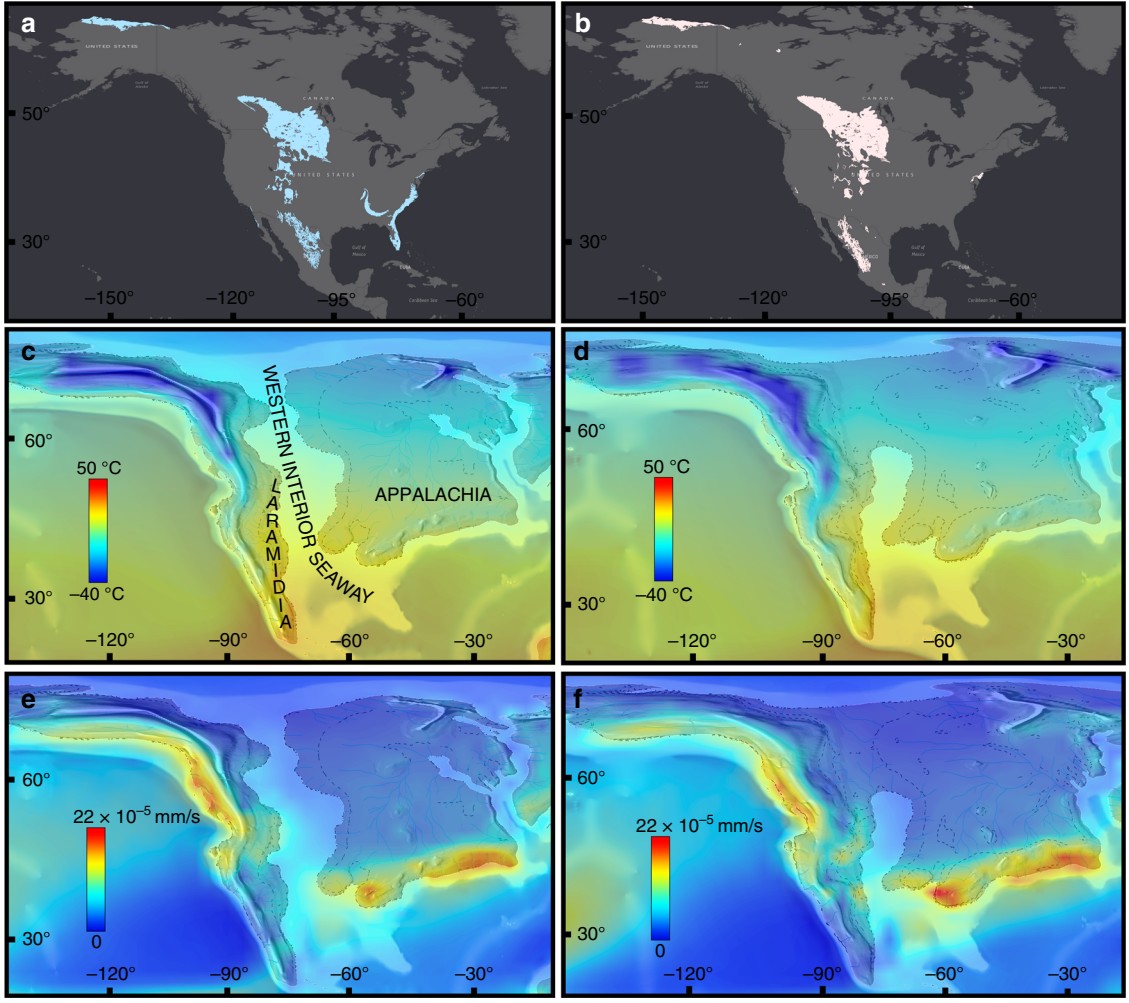

**Fig. 2** Environmental layers used as raw data for this study. Outcrop of Campanian (**a**) and Maastrichtian (**b**) aged terrestrial sedimentary units in North America. Palaeoclimatic outputs from a General Circulation Model configured to the Late Cretaceous (Lunt et al.[34]) with modelled near-surface (1.5 m) annual mean air temperatures (°C) for the Campanian (**c**) and Maastrichtian (**d**). Annual mean precipitation (mm/s) for the Campanian (**e**) and Maastrichtian (**f**). Model outputs have been bilineary interpolated. All the environmental predictors and the statistical operations to select them can be found in Supplementary Note 1

depauperate of dinosaurs, or did it include a viable dinosaurian habitat that has subsequently been lost via a preservational filter? Previous studies were unable to incorporate this critical aspect of data absence into reconstructions of dinosaur diversity, or how it might have been affected by environmental perturbations, regardless of their analytical approach.

One way to examine the impact of data absence is to apply statistical methods, developed by ecologists, which account for spatial biases in diversity data in modern habitats[22]. Ecological niche modelling (ENM) uses correlative statistical algorithms of taxonomic units (e.g. species), coupled with environmental and climatic parameters, to provide a multivariate representation of the hyperspace in which a species is physiologically and reproductively stable[23]. Once a mathematical representation of this fundamental niche is obtained, it can be projected into space to provide an explicitly predictive spatial map of the current geographical location of habitats suitable for these taxa, i.e. the so-called potential niche[24]. These models can then provide a distribution map of niche suitability under different geographic and climatic scenarios, yielding a vital tool for investigating the effects of environmental changes on the potential ecological niches of taxa. By modelling niche space availability using biotic records (fossil occurrences) and abiotic parameters (climatic predictors),

ENM can also be used to map potential ecological niche space dynamics through time in response to physical drivers, refining our knowledge on possible fluctuations in the spatiotemporal distribution of species[25]. Correlative ENMs can use taxonomic occurrences and climatic-environmental layers. For this reason, the modelled niche only indirectly takes into account the constraints to which the potential niche is affected (i.e. biogeographic agents such as dispersal, clade origination, and biotic interaction[26]). Because habitat suitability models are projected into areas that lack geological sampling, they can provide an independent tool for reproducing possible spatially explicit biogeographical trends through time, without the limitations of an imperfect fossil record. Thus, because biogeographical patterns are spatially sensitive to abiotic constraints, ENM can provide an additional metric for modelling deep time responses of organisms to environmental changes[27].

In the last decade, this approach has begun to be applied to palaeobiological problems (e.g. deep time ENM[28] and Paleo-ENM[29]). Examples include: (1) tracking niche fragmentation of the Pleistocene woolly mammoth[28]; (2) the role of climate on diversification and distribution of Neogene horses[30]; (3) the effect of Cenozoic cooling on Eocene planktonic Foraminifera diversity[31]; (4) niche conservatism in Cretaceous

turtles[32]; and (5) niche evolution in Late Ordovician marine invertebrates[33].

Using state-of-the-art Digital Elevation Models (DEMs[34]) of the Cretaceous world, and results from the HadCM3L climate model (Fig. 2), here we apply ENM to deduce dinosaur habitability in North America during the latest Cretaceous (Campanian–Maastrichtian [83.6–66 Ma]), and then used this to simulate and quantify modelled habitat suitability for three diverse and abundant dinosaur clades (Ceratopsidae, Hadrosauridae, and Tyrannosauridae). We then create virtual taphofacies (using taphonomically relevant physical parameters such as sediment flux and surface runoff), and identify areas suitable for potential vertebrate fossil preservation. We use these taphofacies to test statistically significant associations between these parameters and fossil hotspots, to better quantify spatial heterogeneity in the quality of the North American dinosaur fossil record, as well as changes in preservational regimes during the latest Cretaceous. We find no support for the hypothesis of progressive habitat degradation as the mechanism for dinosaur diversity decline[1] in the lead-up to the K/Pg mass extinction. We also highlight the uncertainty associated with a spatially biased fossil record, as well as the physical drivers that influenced dinosaur habitat, biodiversity, and our sampling of their fossil record.

## Results and discussion

### Dinosaur habitability through the latest Cretaceous of North America.
All ENMs scored above 0.90 for the area under the curve (AUC) statistic, indicating strong model performance[35] and that they are able to discriminate presence from background locations[36]. For both ceratopsids and hadrosaurids, temperature of the coldest quarter, precipitation of the driest quarter, and annual temperature standard deviation provided the greatest contribution to the niche models. Tyrannosaurids have almost equal responses to three variables (temperature of the coldest quarter, precipitation of the coldest quarter, and temperature of the warmest quartile; Supplementary Note 1, Supplementary Figures 1–6).

Grid cells with at least one climatic variable outside the univariate range between the Campanian and Maastrichtian are confined to high palaeolatitudes. No fossil occurrence falls within the non-analogue regions; therefore, we retained these areas in the environmental predictor layers, as the models in these regions are not interpreted herein. Habitat suitability overlaying training outcrop area shows ENMs in areas overlying latest Cretaceous terrestrial sedimentary exposure (Fig. 3a). Comparison between suitability in different time bins is reported following shared thresholds of 0.45 and 0.7; values above these thresholds are regarded as highly suitable (see Methods). In areas with outcrop (i.e. training region), intervals of highest habitat suitability (particularly in the threshold above 0.7) correspond to the middle and late Campanian (Fig. 3a). In the early Campanian, wider areas of habitat suitability are shown only in thresholds > 0.45. Similarly, a substantial drop in the area of maximum suitability is observed in the early Maastrichtian, with wider suitable areas only in thresholds > 0.2 (Fig. 3a). The Maastrichtian

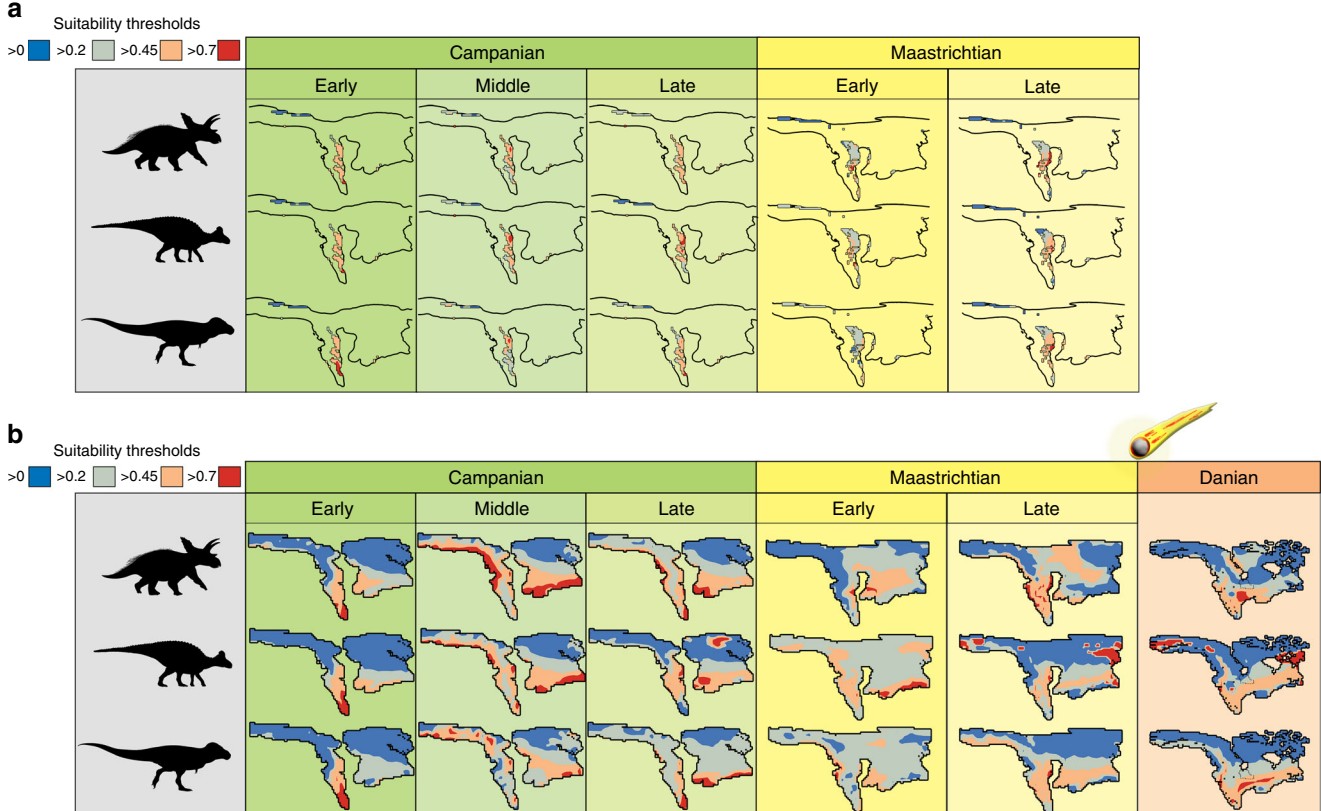

**Fig. 3** Ecological niche models for the three major clades of non-avian dinosaurs in the latest Cretaceous of North America. From top to bottom: Ceratopsidae, Hadrosauridae, and Tyrannosauridae. Niche dynamics in outcrop areas (**a**) show a progressive decrease of high-suitability areas (orange and red) towards the Maastrichtian compared to unsuitable areas (blue) while overall niche stability or increase is shown in a continental setting (**b**). Tyrannosauridae silhouette by Jack Mayer Wood (CC BY 3.0 license: https://creativecommons.org/licenses/by/3.0/ CC BY 3.0); Hadrosauridae silhouette by Pete Buchholz (under CC BY-SA 3.0 license: https://creativecommons.org/licenses/by-sa/3.0/); and Ceratopsidae silhouette by Mariana Ruiz (modified by T. Michael Keesey) under the Public Domain Mark 1.0

shows highly suitable areas (>0.7) in the northeastern margins of the WIB, in areas now occupied by the fossil-rich assemblages of Montana, Wyoming, and the Dakotas (Fig. 3a). Compared to the higher suitable intervals of the middle and late Campanian, the minimal suitable areas (>0.2) occupy more space in the Maastrichtian ENMs than in other time bins (Fig. 3a).

ENMs projected onto the whole terrestrial extent of the North American continent (i.e. projection region) show a different pattern (Fig. 3b). Although in the outcrop model we see a decrease in higher suitability areas towards the end-Cretaceous, the continental model shows a more stable and consistent pattern between substages, where suitability is constant, if not more widespread in the Maastrichtian (Fig. 3b). Although high-suitability areas in the northeastern margin of the WIB are still present, there is a latitudinal expansion southward of higher suitability area (both >0.45 and >0.7 thresholds; Fig. 3b). Interestingly, the greatest reduction in habitable space, in which the only suitable habitats are shown with the 0.45 threshold, is seen in the early substages of both the Campanian and Maastrichtian (Fig. 3b). In the lower threshold of habitability (>0.2) suitable space increases in the early Maastrichtian (Fig. 3b), possibly as an effect of lower occurrence numbers in this time bin, making inference on habitability in this interval more uncertain (Table 1).

The quantification of habitability in the outcrop models (Fig. 4a) shows a peak suitability in both thresholds for ceratopsids in the

middle Campanian, followed by a drop, which reaches its minimum in the early Maastrichtian, before rising in both thresholds (>0.45 and >0.7) in the late Maastrichtian. Hadrosauridae shows a similar trend, with almost equally high peaks in the middle–late Campanian (particularly in the maximum suitability threshold, >0.7, with a somewhat more marked drop in the >0.45 one), followed by a drop in the early Maastrichtian, slightly rising again in the late Maastrichtian (Fig. 4a). This rise is more markedly reached for the relatively lower threshold (>0.45), approaching a similar suitability level to the Campanian one (Fig. 4a). Tyrannosaurid habitability peaks in the middle–late Campanian, before dropping in the early Maastrichtian, and recovering before the K/Pg boundary (Fig. 4a). As for the spatial projection (Fig. 3b), continental quantification shows a different pattern than that of outcrop area (Fig. 4b). Ceratopsidae has its highest habitability of the lower threshold (>0.45) in the late Maastrichtian, with the higher threshold of habitability peaking in the middle Campanian, but this is almost equivalent to late Maastrichtian values (Fig. 4b). There is a habitat contraction in the early Maastrichtian, where suitability reaches the low level of the equally undersampled early Campanian (Fig. 4b). Similar patterns are shown by both Hadrosauridae and Tyrannosauridae, although these two taxa have their absolute peak in habitability (for both thresholds) in the middle Campanian, late Campanian, and late Maastrichtian (Fig. 4b). The consistent presence of low levels of habitability in the early Campanian and early Maastrichtian (Fig. 4) is probably best explained by the lower number of unique spatial occurrences present in these two substages compared to the other intervals, rather than representing a genuine macroecological signal (Table 1). However, we caveat this with a note of caution: although lower suitability threshold patterns might indicate relatively less favourable conditions for dinosaur habitats to persist, they might also highlight uncertainty in assessing spatiotemporal patterns for dinosaur climatic niches with maximum confidence.

A late Maastrichtian ENM was also projected into the first stage of the Cenozoic (the Danian) to test the effect of early

**Table 1 Number of unique occurrences per time bin used as training sample for the ecological niche modelling**

|  | Ceratopsidae | Hadrosauridae | Tyrannosauridae |
|---|---|---|---|
| Early Campanian | 6 | 21 | 9 |
| Middle Campanian | 20 | 39 | 18 |
| Late Campanian | 40 | 58 | 33 |
| Early Maastrichtian | 7 | 17 | 22 |
| Late Maastrichtian | 43 | 46 | 29 |

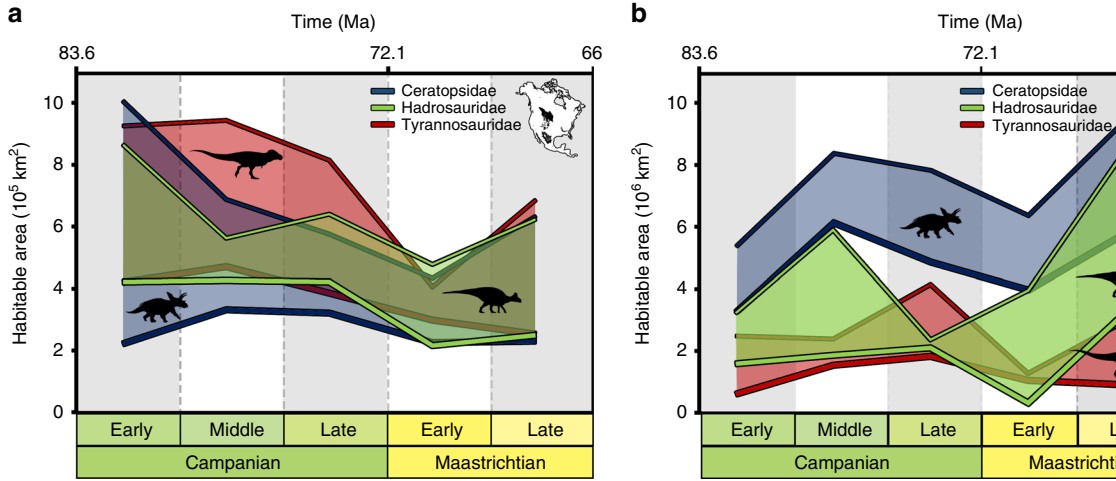

**Fig. 4** Time-bin quantification of habitat suitability of ecological niche models. Quantification is shown for only outcrop area (**a**) and for the whole latest Cretaceous North American palaeocontinent (**b**). Both sets of models have been trained with the same extent (outcrop area), but while **a** shows quantification in training region, plot in **b** shows original models projected to North America. Thick line represents higher suitability threshold quantification (>0.7), while thinner line is lower suitability threshold one (>0.45). An overall decrease in habitat suitability in available outcrop areas is shown in **a** while an increase is obtained for all the three clades in North America (**b**). Orange column in **b** represents habitat stability when niche models are projected after the K/Pg boundary, showing potential habitability for these clades after the end-Cretaceous mass extinction. Numeric values on the y-axes are in $10^5$ km$^2$ in **a** and $10^6$ km$^2$ in **b**. Numeric values to build this figure are in Supplementary Table 1 and Supplementary Table 2. Tyrannosauridae silhouette by Jack Mayer Wood (CC BY 3.0 license: https://creativecommons.org/licenses/by/3.0/ CC BY 3.0); Hadrosauridae silhouette by Pete Buchholz (under CC BY-SA 3.0 license: https://creativecommons.org/licenses/by-sa/3.0/); and Ceratopsidae silhouette by Mariana Ruiz (modified by T. Michael Keesey) under the Public Domain Mark 1.0

Paleocene climate in defining the abiotic niche of these three dinosaur clades, including the possibility of a long-term decrease in habitability (Fig. 3b). We observe a southern migration of suitable dinosaur habitat in the case of Ceratopsidae and Tyrannosauridae, with some peaks of suitability for Hadrosauridae at higher latitudes (Fig. 3b). Highest niche suitability (>0.7) in the Danian slightly decreases from late Maastrichtian levels for Hadrosauridae and Ceratopsidae, but shows a small increase for Tyrannosauridae (Fig. 4b). These Danian levels are comparable to the late Campanian for hadrosaurids and tyrannosaurids, but it reaches the lowest value of the time series for ceratopsids (Fig. 4b). However, the lower habitability threshold (>0.45) shows

suitability levels still comparable to the more habitable intervals (middle–late Campanian and late Maastrichtian; Fig. 4b).

**Spatiotemporal biases in the latest Cretaceous of North America.** Kernel density analyses highlight a significant spatial association of clusters of dinosaur occurrences. In the Campanian, these fossiliferous clusters are grouped together in a few restricted areas, corresponding to Dinosaur Provincial Park in Alberta, Canada, in the north (Fig. 5a, b), and to the southern Kaiparowits, Kirtland, and Fruitland assemblages. These Campanian localities occupy a palaeolatitudinal band between

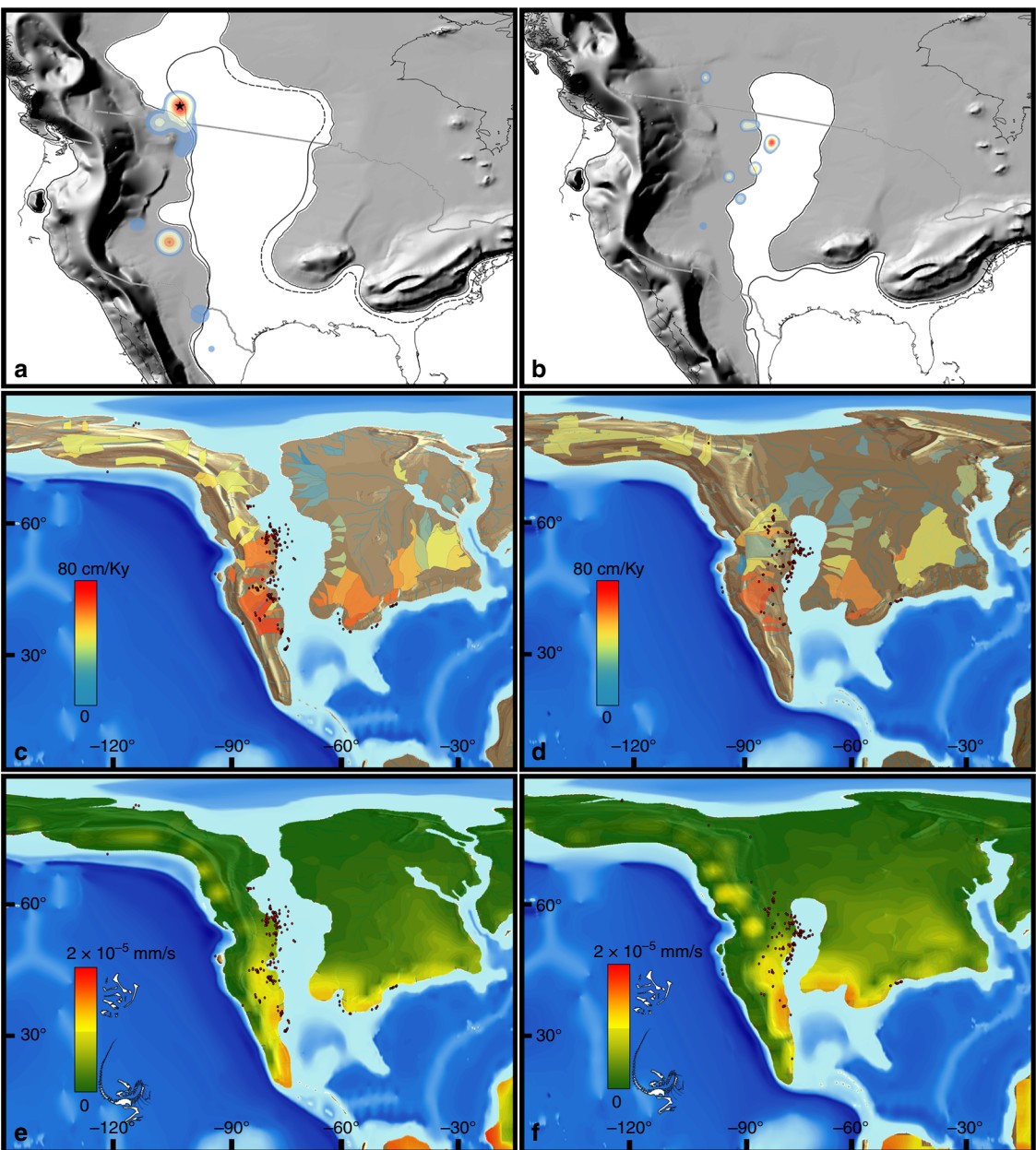

**Fig. 5** Virtual taphofacies and hotspot analysis in the latest Cretaceous of North America. Kernel density in the Campanian (**a**) and Maastrichtian (**b**), with red representing highest-density and blue low-density hotspots. Grey line representing country boundaries overlaid on palaeogeographies. Black dashed line represents sea-level lowstand. Star in **a** is Dinosaur Provincial Park. Sediment fluxes (cm/ky) calculated using basin drainage tools (see Methods section) in the Campanian (**c**) and Maastrichtian (**d**). Surface runoff (mm/s) models overlaid spatially in the Campanian (**e**) and Maastrichtian (**f**). Red dots represent dinosaur occurrences (data points in the middle of the Western Interior Seaway are there because they are associated with lowstand phases). Brown and grey colours represent underlying topography of the digital elevation models. Dinosaur skeletons in figure legends redrawn by A.A.C.

**Table 2 Virtual taphofacies values with statistical support for the $\chi^2$ value of Pearson's chi-squared and Fisher's tests of association with (Y) and without Yates' correction, showing which virtual taphofacies is significantly associated with a higher number of occurrences (in Campanian, Maastrichtian, and total latest Cretaceous hotspots)**

| | Campanian hotspot | Campanian cold spot | Maastrichtian hotspot | Maastrichtian cold spot | Total hotspot | Total cold spot |
|---|---|---|---|---|---|---|
| High sediment flux | 206 | 15 | 0 | 3 | 206 | 18 |
| Low sediment flux | 7 | 3 | 24 | 19 | 31 | 22 |
| $\chi^2$ value, df = 1 | 7.1747 | 7.1747 | 3.5011 | 3.5011 | 38.869 | 38.869 |
| $p$ value, df = 1 | 0.007394* | 0.007394* | 0.06133 | 0.06133 | 0.0000004532* | 0.0000004532* |
| $\chi^2$ value (Y), df = 1 | 4.3077 | 4.3077 | 1.6215 | 1.6215 | 36.207 | 36.207 |
| $p$ Value (Y), df = 1 | 0.03794* | 0.03794* | 0.2029 | 0.2029 | 0.000001774* | 0.000001774* |
| Fisher's exact test $p$ | 0.03389* | 0.03389* | 0.1014 | 0.1014 | 0.00002569* | 0.00002569* |
| High surface runoff | 15 | 14 | 0 | 6 | 15 | 20 |
| Low surface runoff | 125 | 0 | 84 | 3 | 209 | 3 |
| $\chi^2$ value, df = 1 | 66.379 | 66.379 | 59.862 | 59.862 | 110.48 | 110.48 |
| $p$ value, df = 1 | 3.72E-16* | 3.72E-16* | 1.017E-14* | 1.017E-14* | 2.2E-16* | 2.2E-16* |
| $\chi^2$ value (Y), df = 1 | 60.666 | 60.666 | 49.326 | 49.326 | 103.98 | 103.98 |
| $p$ value (Y), df = 1 | 6.8E-15* | 6.8E-15* | 0.000000002168* | 0.000000002168* | 2.2E-16* | 2.2E-16* |
| Fisher's exact test $p$ | 2.946E-12 | 2.946E-12 | 1.102E-07 | 1.102E-07 | 2.2E-16* | 2.2E-16* |

*Significant $p$ values at $\alpha = 0.05$

approximately 30° and 60°N, with observed diversity peaks at 40° and 55°N (Fig. 5a). In the Maastrichtian, clusters correspond to the dinosaur-rich deposits in eastern Montana and South Dakota (e.g. the Hell Creek Formation). A complete set of hotspot analyses is included in the supplementary material (Supplementary Note 2 and Supplementary Figures 7–21). During the Campanian, these assemblages are statistically associated with higher sediment flux areas (~280–700 cm/ky) (Table 2), bordering the eastern margin of the WIS, from Canada (Dinosaur Park Formation) to Mexico (Aguja Formation). Localities with high quality preservation (e.g. in the Dinosaur Park Formation) are also associated with relatively low surface runoff values ($\leq 7 \times 10^{-4}$ mm/s), whereas less well-sampled southernmost localities (e.g. in the Aguja Formation) are characterised by elevated values of surface runoff ($\geq 5 \times 10^{-3}$ mm/s). Maastrichtian sediment flux models show clusters (Fig. 2d) that correspond to a narrower palaeolatitudinal band (~50°N; e.g. the Hell Creek Formation), and coincide with lower sediment fluxes (~80–200 cm/ky). As in the Campanian, elevated values of surface runoff characterise lower palaeolatitude watersheds in the Maastrichtian, with lower values at higher palaeolatitudes (Fig. 2c, d). The number of occurrences in significant hot and cold spots was used to compile $2 \times 2$ contingency tables with taphofacies intervals and quantitatively evaluated with correlative statistics (Pearson's $\chi^2$ and Fisher's tests). The $\chi^2$-test on Campanian occurrences ($n = 231$) found a non-random preferential distribution (by a 206:7 ratio) of fossil hotspots with respect to high sediment fluxes (Fig. 5c; Table 2). The same result is supported by Fisher's exact test. However, in the Maastrichtian (Fig. 5d), the same correlation with high sediment fluxes is not statistically supported, possibly as a consequence of lower sample size in these clusters ($n = 46$). The $\chi^2$-test on the totality of latest Cretaceous (Campanian + Maastrichtian) hotspots shows a similarly high association of high sediment fluxes and hotspot occurrences by a ratio of 206:31. Campanian hotspots ($n = 154$) show a strong association with low surface runoff (Fig. 5e; Table 2), by a 25:3 ratio. Similarly, the same association is found in the Maastrichtian ($n = 94$; Fig. 5f), by a statistically significant ratio of 84:0. The relationship is maintained when we combine the Campanian and Maastrichtian hotspots (by a ratio of 209:15). All of these results on surface runoff taphofacies are also supported by the use of Fisher's test. It is also notable that the number of occurrences falling in

significant hotspots is greater for both taphofacies in the Campanian than in than Maastrichtian (sediment flux by a ratio of 213:24, and surface runoff by a ratio of 70:42). This highlights the reduction in spatial extent of favourable taphonomic conditions, which is greater in the Campanian, enabling a more widespread preservation along the eastern coastline of the WIB, in contrast to the more localised deposits observed in the Maastrichtian.

**The current view of abiotically driven latest Cretaceous diversity decline.** It has been argued that the purported latest Cretaceous diversity decline of non-avian dinosaurs was due to a suite of abiotic drivers, specifically climatic, prior to the mass extinction event at the K/Pg boundary[8,37]. In particular, the apparent Campanian peak and Maastrichtian decline in North America coincide with major tectonic events[38]. The latest Cretaceous of western North America was characterised by the Sevier and Laramide orogenies (forming the proto-Rocky Mountains), as well as the expansion (and eventual retraction) of the WIS[20]. These might have formed abiotic barriers that led to allopatric speciation, ultimately resulting in high late Campanian diversity[38,39]. Apparent differences in the composition between presumably coeval faunal assemblages in the late Campanian of Laramidia have been interpreted as evidence of biogeographic provincialism between northern versus southern communities[39], possibly indicating the presence of an environmental barrier, either of physical or climatic nature[40]. Subsequently, under this scenario, the Maastrichtian sea-level regression removed a major barrier to west–east dispersal (and perhaps facilitated north–south dispersal too[41]), therefore reducing levels of regional endemism[39] (see also ref. [42]), and eventually leading to depressed Maastrichtian diversity. A prolonged episode of climatic cooling throughout the latest part of the Cretaceous (from the Cenomanian/Turonian boundary onward, 93.9–66 Ma[43]) has also been also proposed as a major driver for declining trends in dinosaur diversity up to their final extinction at the K/Pg boundary[1,8,44].

**The impact of heterogeneous sampling on diversity trends.** As discussed above, some authors contend that the rich latest Cretaceous North American record means that the apparent drop in numbers of dinosaur species from the Campanian to the Maastrichtian can be interpreted as genuine. This is reasoned because

of a purportedly better representation in the geological record of Maastrichtian stratigraphic units relative to those from the Campanian[3], and evidenced by the numbers of dinosaur-bearing formations (DBFs[14]) and outcrop area[38] (Fig. 1). However, other authors have shown that there is little change in the numbers of DBFs (or dinosaur-bearing collections) from the Campanian to the Maastrichtian[4,5], and outcrop area is not always a good proxy for sampling[45]. Furthermore, middle–late Campanian units are over-sampled compared to other latest Cretaceous terrestrial units, largely because of the exceptionally fossiliferous localities in Dinosaur Provincial Park, where geographical, climatic, topographic, historical, and sociological factors make it a uniquely palaeontologically productive area[46] (Fig. 5a, b). When outcrop exposure is plotted against raw diversity of the three dinosaur clades examined in this study, peaks in diversity for Ceratopsidae and Hadrosauridae correspond to the highest levels of exposures (Fig. 1).

The Late Cretaceous North American dinosaur record is not only chronologically averaged but also spatially biased towards a few areas[6,14,41,47]. Kernel density (Fig. 5a, b) reveals that just a small number of groups of geographically localised collecting sites account for most of the Cretaceous North American dinosaur record. This low spatial variance can have implications for diversity estimates[16]. Our ENM simulations (Fig. 3) suggest the presence of a relatively large and unsampled area of habitat suitability in the Maastrichtian, equivalent to, if not wider, than that seen in the Campanian, highlighting a possible major loss of sampled localities. As we demonstrate, an extensive expanse of suitable terrestrial areas are not preserved or sampled in the geologic record, meaning that we are likely to exclude a great number of habitats from our estimates, potentially missing many diverse communities. Furthermore, most dinosaur-bearing collections in the WIB are represented by lowland floodplain environments, and therefore preserve a limited subset of depositional environments and thus fossil-bearing lithologies[47]. As such, ultimately we need to more extensively sample a wider range of biotopes if we want to provide a more complete picture of Late Cretaceous North American faunas, and inferences on diversity dynamics should take into account the uncertainty due to the lack of such vast and potentially habitable, but unsampled, areas.

Heterogeneity in terrestrial sampling is pervasive both between stages but also within stages of the Late Cretaceous North American record, skewing our interpretation of palaeobiological patterns. For example, the proposed biogeographic provinces in the Campanian of the WIB[38,39,41] are potentially an artefact of differential taphonomic and collection regimes. This might be caused by sampling bias between northernmost localities (e.g. the highly productive Dinosaur Park Formation[46]) compared to the relatively less well-sampled southern localities (e.g. the Aguja Formation[40]). In particular, there is a clear distinction in taphonomic suitability between northern and southern localities (Fig. 2c, d). Episodes of climate-induced mass mortality, probably due to seasonal precipitation patterns, led to the creation of high density and hyper-productive sites in the Dinosaur Park Formation[46]. On the other hand, southern localities were more often characterised by warmer and drier conditions, with periodic flooding. Sedimentation rates were generally lower as a consequence, whilst erosion from surface runoff was elevated, often resulting in disarticulated and incomplete dinosaur remains[48].

A key argument for the existence of discrete northern and southern biogeographic provinces is the purported penecontemporaneity of late Campanian faunal assemblages[38]. However, detailed chronostratigraphic studies of the terrestrial stratigraphy of the WIB indicate that many of these dinosaur-bearing strata are likely to be diachronous, giving us the false impression of

dealing with hyper-diverse, disparate coeval faunal assemblages[49]. Lehman[41] hypothesised an even higher level of provincialism in the Maastrichtian. However, a subsequent analysis recovered strong statistical support for low beta-diversity (i.e. low endemism) in the Maastrichtian of western North America[42], with previous suggestions for provincialism reinterpreted as a product of heterogeneous sampling. A model of latitudinally arranged biogeographic provinces in the WIB[38–41] might therefore be the result of differential sampling and preservational patterns, as well as time-averaging[49]. In light of such spatiotemporal biases, we are also highly sceptical of recent claims of faunal provinces in poorly sampled Appalachia[16].

ENM outputs restricted to areas where Campanian–Maastrichtian terrestrial sedimentary rocks outcrop at the surface show that non-avian dinosaur habitat suitability decreased from the late Campanian to the late Maastrichtian (Fig. 4a), which broadly mirrors the reduction in the group's observed (raw) diversity (Fig. 1), as documented in previous studies (e.g. refs. 4,5,14). However, when a continental terrestrial projection is considered (i.e. via modelling suitable dinosaur habitats across the whole of North America), a different picture emerges, highlighting the uncertainty that must be considered when extrapolating macroecological signals from palaeontological data at continental or global extents. These results show that dinosaur habitat suitability was stable or actually increased throughout the Maastrichtian, with no evidence for climatically driven habitat degradation; as such, we contend that there is no clear abiotic driver for a long-term decline in dinosaur diversity.

**An alternative abiotic scenario for latest Cretaceous dinosaur diversity.** Although a literal reading of the fossil record suggests that eustatic and tectonic drivers were responsible for shaping dinosaur diversity dynamics in the latest Cretaceous of North America, possibly causing a diversity decline as a result of a reduction in the dinosaur species' abiotic habitat, here we propose a different interpretation based on the spatial spread of fossil occurrences and our deep time ENM results (Fig. 6). The relationship between eustasy and the dinosaur fossil record has been the subject of several studies (e.g. refs. 50,51), with a richer record present during sea-level highstands[50]. The rise and fall of sea level corresponds with peaks and troughs in the deposition of terrigenous sediments within the inner shelf. Eustatic processes have been considered in the past as an example of common cause effect[51], with their rhythmic fluctuations causing either drops or increases in biodiversity, whilst at the same time regulating cycles of deposition and erosion of sediments, shaping palaeodiversity according to the biological record preserved in those sedimentary layers. Although a maximal contraction of terrestrial habitat space can occur in the transgressive phase, alluvial fans and deltaic deposits start to expand basinward on the inner shelf, leading to sediment accumulation at relatively shorter distances of transportation, which in turn promotes the rapid burial and eventual fossilisation of vertebrate remains. Conversely, the opposite is caused by a significant and rapid fall in sea level: terrestrial erosion increases and terrigenous sediments accumulate in more localised areas on the inner shelf. As sea level continues to fall, recently deposited sediments will be eroded and preservation potential will be reduced.

In the Campanian, the WIS was at a highstand phase[52], extending north to south along the eastern coast of Laramidia. This created high accommodation space for deposition of sediments eroding out from the Campanian phase of the Sevier orogeny and the contemporaneous onset of the Laramide orogeny. The early–middle Campanian establishment of the main thrust deformation of the Sevier orogeny rearranged basin

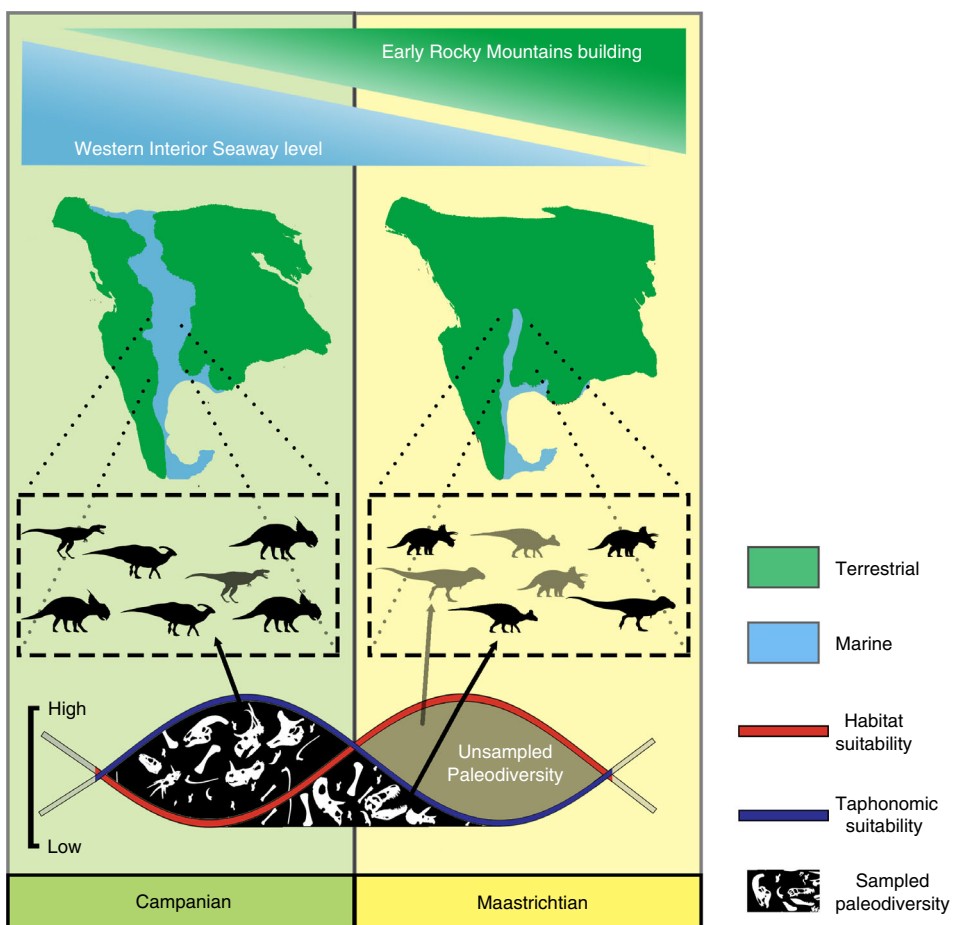

**Fig. 6** Conceptual integration of the results from niche modelling and geological modelling of fossil occurrences. Eustatic (blue shaded bar on top) and tectonic (top green shaded bar) drive the distribution of depositional environment affecting fossil preservation. Campanian palaeogeography (left) fosters increased and more widespread preservation of fossil communities than the Maastrichtian (right) due to the higher accommodation space provided by the highstand of the Western Interior Seaway (left) than during the late Maastrichtian regressive phase (right). On the other hand, the latter time interval (right) may have provided equal if not greater habitable space in terrestrial areas, which are not captured by the geologic record. Diagram at the bottom shows how a high Maastrichtian habitat suitability (red line) is not in phase with the lower preservation potential of this stage, causing lower taphonomic suitability than in the Campanian. This combination of conditions provide a depauperate raw diversity record for non-avian dinosaurs towards the K/Pg. Chasmosaurinae silhouette by Mariana Ruiz (modified by T. Michael Keesey) under the Public Domain Mark 1.0. Saurolophinae silhouette by Pete Buchholz (under CC BY-SA 3.0 license: https://creativecommons.org/licenses/by-sa/3.0/); Tyrannosaurinae silhouette by Jack Mayer Wood (CC BY 3.0: https://creativecommons.org/licenses/by/3.0/ CC BY 3.0); Centrosaurinae silhouette by Andrew A. Farke (under CC BY 3.0: https://creativecommons.org/licenses/by/3.0/); Lambeosaurinae silhouette by Jack Mayer Wood (under CC0 1.0); and Albertosaurinae silhouette by Craig Dylke (under CC0 1.0)

drainage geometry and increased subsidence rates, further increasing accommodation space and promoting sediment flow to a wider area[53] (Fig. 3). This elevated sedimentation increased the likelihood of fossil preservation. It also meant that biodiversity was preserved in a latitudinally widespread area.

The Maastrichtian forced regression of the WIS freed a vast amount of wide inland area previously covered by the epicontinental seaway. These topographic lowland areas contained suitable areas for terrestrial vertebrates, compatible with the fundamental niches of dinosaurs, and thus enabled their areal expansion. The Sevier-Laramide orogenies pushed the thrust belt eastward. This might have caused a habitat bottleneck effect[54] towards more inland areas, which is supported by our modelled sustained habitability in these regions. Dinosaur habitats gained new spaces as a result of the interplay between three main factors: (1) the fundamental stability of dinosaurian climatic niches; (2) topographic changes due to the Sevier-Laramide tectonism; and (3) the forced regression of the WIS. This vast and widespread habitat likely had a broadly similar palaeoclimate to the rich dinosaur-bearing deposits in present-day Montana and Alberta.

Several authors[2,54,55] have proposed that increased terrestrial area during this regressive phase would have promoted higher terrestrial biodiversity. However, the opening of new habitats in the Maastrichtian was not accompanied by the same favourable physical conditions for fossilisation that characterised the Campanian (Fig. 6). Occurrence data show that fossil-yielding localities are progressively displaced southeastwards from the Campanian to the Maastrichtian. This shift is almost certainly the result of basinal subsidence and sea-level changes, leading to the contraction of the WIS and the resultant displacement of the fossil-preserving coastal environments[20,50,52,53]. Reduced sediment fluxes (Fig. 5d), progressive loss of coastlines and inner shelf areas, and relatively higher surface runoff values (Fig. 5f), caused a decrease in relative preservation potential, as well as an increase in erosion, with the consequent loss of a wider pool of dinosaur-bearing localities contributing to a depauperate pre-extinction dinosaurian record.

Our results suggest that the observed raw dinosaur diversity decline in the lead-up to the K/Pg boundary is unlikely to reflect a true biological signal, and is instead likely to be the product of a

geographically uneven fossil record. During the Campanian, peak dinosaur habitat was coincident with areas of high sedimentation and suitable taphonomic conditions, in which fossils were likely to be preserved. ENM show no evidence of habitat degradation from the Campanian to Maastrichtian. However, the depositional environments in the Maastrichtian contained relatively less-favourable taphonomic conditions, providing a spatially less rich vertebrate fossil record. Although we do not exclude a biotic influence of tectonic and eustatic changes on true dinosaur diversity in tandem (e.g. allopatric speciation in the Campanian, and faunal mixing and greater area in the Maastrichtian), we contend that this played a relatively minor role when compared with their abiotic effects on observed diversity.

**Latest Cretaceous dinosaur diversity and the K/Pg mass extinction.** The rise in areal extent of suitable dinosaur habitat throughout the latest Cretaceous, coupled with the increasingly limited spread of dinosaur-bearing localities, suggests caution when interpreting North American dinosaur diversity trends and their response to abiotic changes from the Campanian to the Maastrichtian. A potential Maastrichtian diversity increase is supported by morphological disparity studies[6,56], as well as diversity analyses that use subsampling[5] or modelling approaches[4]. Although stratigraphic resolution is far poorer, a Campanian-to-Maastrichtian increase in dinosaur diversity and disparity has also been reconstructed for other palaeocontinental regions (e.g. Asia[6] and Europe[57]).

A clade-specific habitat suitability model trained in the late Maastrichtian and projected into the first stage of the Cenozoic shows little change in niche space through the K/Pg boundary (Fig. 3b). A rearrangement of hypothetically suitable dinosaur habitats is probably influenced by a changing geography (e.g. the complete disappearance of the epicontinental WIS). Although our models do not incorporate any external perturbing effects that might have altered the normal climatic pattern in the Late Cretaceous or Paleocene, we interpret this result as evidence of a relatively long-term trend of habitat stability both in the lead-up and across the K/Pg boundary. This would suggest that climatic changes did not affect the ecological niche of dinosaurs over prolonged timescales during this critical period. In contrast, the conditions immediately after the K/Pg event would have been very unsuitable for these clades, and this unsuitability would have persisted for some time, from <10 years up to >10 kyr[58] or even 100–300 kyr[59,60], leading to the demise of the non-avian dinosaurs. These geologically rapid climatic fluctuations are not detected by our stage-resolution general circulation climatic models. However, coupled with our results for the Campanian and Maastrichtian, we contend that there is little evidence to evoke a long-term decline in dinosaur diversity. Instead, our results provide support for a kill-mechanism with a high temporal resolution, compatible with an instantaneous catastrophic event, as has been proposed for the Chicxulub post-impact scenario[5,58]. Given the lack of evidence for a long-term degradation of habitat suitability, we suggest that there would have been no K/Pg extinction of the dinosaurs without this geologically instantaneous, catastrophic event[5]. The likely presence of an extensive suitable niche space in the earliest Cenozoic also suggests that dinosaurs might have been able to recover if they had survived the aftermath of this mass extinction event. Instead, following their demise, this available niche space likely contributed to the explosive radiations of placental mammals and neornithine birds[61,62]. This new perspective on spatial bias expands our set of tools in the study of palaeobiogeography, shedding new light on diversity patterns in the lead-up to the K/Pg mass extinction, a pivotal time interval that shaped the evolution of the modern biota.

## Methods

**Dinosaur occurrence data set.** We downloaded a comprehensive database (>4000) of latest Cretaceous (Campanian–Maastrichtian: 83.6–66 Ma) North American dinosaur body fossil occurrences from the Paleobiology Database (https://paleobiodb.org) on 20 September 2017. All occurrences of isolated specimens (i.e. single bones) were excluded, as were those described as displaced or transported in the database. We also only retained occurrences belonging to Ceratopsidae, Hadrosauridae, and Tyrannosauridae: these three clades were the most diverse and abundant non-avian dinosaur groups in the latest Cretaceous of North America. Our pruned data set consists of 1973 occurrences (see Data availability). Although the value of supra-generic taxonomic ranks in ENM has been debated[63], the use of family-level clades to model environmental niches of modern[64] and extinct[32] taxa is not uncommon, and has been widely applied for characterising the climatic niches of modern taxa[63–65]. Given the limitations of an incomplete fossil record, palaeontologists have used family-level groups to investigate palaeogeographical coverages of fossil taxa[66,67]. Although niche conservatism probably acts in different ways at different taxonomic levels, this phenomenon has been shown to occur at supraspecific and particularly family-level clades, notably in species level traits such as geographical range sizes, even when affected by events such as range shifts and local extinctions[68,69]. In addition, each of the terrestrial clades investigated in this study is represented by species with comparable ecomorphological traits (i.e. large-bodied, obligate herbivorous quadrupeds, or moderate to large-sized bipedal obligate predators), which show reproductive[70], ecomorphological[71–73], and life history similarities[70], providing enough within-clade functional consistency to assume close similarity of their climatic niches. These data were also used to produce the diversity curves shown in Fig. 1.

Occurrence layers were processed with ArcGIS 10.2.2 (ESRI) to reduce high-density clusters, a procedure necessary for using MaxEnt (see below), as this particular machine-learning algorithm is highly sensitive to high-density data points[24]. In addition, a procedure for subsampling the records of regularly distributed samples in space was selected to minimise biasing effects[74]. A systematic sampling approach[75] was used to solve the effect of using spatially biased occurrence data in ecological niche models. In addition, MaxEnt automatically discards redundant occurrences in a single cell that might create a density-dependant bias in training the distribution model. We split the occurrence data set to make it compatible to sub-stage-level stratigraphic resolution, dividing it into three substages for the Campanian (early, middle, and late), and two for the Maastrichtian (early and late), such that each has an approximately ~3.5 Myr duration.

**Palaeogeographic DEMs.** Detailed palaeogeographic maps are fundamental to geospatial studies. Getech Plc. has provided a global atlas of 1:20,000,000 scale palaeogeographic maps for regional-scale palaeogeographic interpretations. These maps are created from publicly available stratigraphic literature, supplemented by fieldwork, including both broad scale facies identification and lithology (Fig. 2). It is important to note, however, that these maps are time-averaged (to stage level) approximations, and that this impacts upon the level of inference that can be ascertained from model results. Reconstruction of the positions of tectonic plates were achieved using Getech's in-house global plate model. This rigid plate model, in which the shape of tectonic plates does not change over time, comprises a global distribution of present-day tectonic plates and a set of finite rotations that describe relative motion between each plate. Tectonic plate boundaries and major structural features have all been defined from field data. The motion of one plate relative to another during a given time interval can be represented by an angle of rotation about a fixed semi-axis (Euler Pole) on Earth's surface. The absolute past position of any given plate, relative to Earth's spin axis, is calculated by adding finite rotations within a plate hierarchy. This relative motion information is based on a variety of sources: oceanic magnetic anomalies, fracture zone orientations, palaeomagnetic poles, geological relationships, and tectonic histories of onshore structural features. Regarding the palaeogeographic reconstructions used as boundary conditions, they allow for far greater temporal resolution than other reconstructions, unlike many previous studies that only use one palaeogeography. The palaeo-digital elevation models used as boundary conditions in the model for each stage are informed by these reconstructions, which are in turn constrained by extensive geological databases (both public and private from in-house exploration and cores from the oil industry). These data include published lithologic, tectonic, and fossil studies, the lithologic databases of the Palaeogeographic Atlas Project (University of Chicago), and deep sea (Deep Sea Drilling Project [DSDP/ODP]) data, as well as data from the Atlas projects databases.

These palaeogeographies were initially created on a 0.5° × 0.5° grid and then upscaled to the model resolution (3.75° × 2.5°). This means topographic and bathymetric information is broadly conserved, as it is resolved at a lower resolution, causing less uncertainty incorporated into the downscaled model climate variables. The methods used to build these palaeogeographies are described in Markwick[76]. Additional information is available at http://www.getech.com/.

**Palaeoclimatic general circulation models.** One of the main limitations for the application of ENM in deep time has been the lack of high-resolution climatic data[29]. Previous work using ENM in deep time has tended to use interpolated layers based on localised sedimentological and isotopic proxies[30,77]. In this study, we used climatic model outputs (e.g. near-surface [1.5 m] temperature, annual average precipitation; Fig. 2, Supplementary Note 1) from the fully coupled atmosphere-ocean GCM HadCM3L version 4.5 Atmospheric–Ocean General Circulation Model[78]. The specific version of the model used is HadCM3BL-M2.1aE, in the nomenclature of Valdes et al.[78], wherein a full description of the model can be found. The simulations of the Campanian and Maastrichtian used in this study are described in full by Lunt et al.[34]. In brief, the boundary conditions consist of the same Getech Palaeogeographic DEMs described above, and an atmospheric CO$_2$ concentration of 1120 ppmv, which is within the range of uncertainty provided by the latest proxy pCO2 reconstructions of Foster et al.[79].

The sub-grid-scale orographic features of the topography are calculated within the model, enabling finer-scale features to have an impact on the climate signal. Both regional and large-scale circulation (and associated energy and momentum fluxes), as well as temporal fluctuations, are also resolved in the model, which are important determinants of the climate signal.

The model simulations are run for a total of 1422 years, and the climate variables used in the ENM are an annual average of the last 30 years of these simulations. Solar luminosity is stage-specific and calculated with the methods of Gough[80]. For model evaluation, terrestrial model-data comparisons have been carried out with specific time periods, for example in the Eocene[81], the Oligocene[82], and the Miocene[83]. In general, the model does a reasonable job of reproducing terrestrial climates, given the uncertainties in the data. However, there are indications, in common with many models of this complexity, that the high latitude temperatures during the warmest periods of Earth's history are unrealistically cold. For examples in the literature of other palaeontological-geological validations of these models (HadCM3 and HadCM3L), see: Markwick and Valdes[84], Sellwood and Valdes[85], Waterson et al.[32], and Fenton et al.[31]. Climate variables from the model that are used in ENM analyses for the Campanian and Maastrichtian (Supplementary Note 1 and Supplementary Figures 1–6) are available at: http://www.bridge.bris.ac.uk/resources/simulations.

**Ecological niche modelling.** ENM is a quantitative approach to predict species distributions according to abiotic requirements, creating a correlative model that can be projected in space. It requires geographically explicit information on species occurrences and the suites of environmental conditions experienced at each occurrence point. The ability to incorporate spatial biases in the ENM modelisation phase makes MaxEnt (maximum entropy algorithm[86]) the recommended presence-only algorithm to work with ENM using fossil occurrences[22,29,87,88]. We used downscaled climatic data from 3.75° × 2.75° to 1.25° × 1.25° in order to provide a closer match to the resolution of the square grid areas overlapping outcrop areas. To minimise extrapolations that would have caused erroneous interpretation of our explicitly spatial models, we subsampled occurrences, keeping one for each climatic grid. This was implemented in ArcGIS 10.2.2 (ESRI), using a neighbour index method[89], eliminating spatially associated clusters, until a random distribution of points was obtained. We then used the resultant spatial distribution data (Table 1) to create a correlative model trained on the area where only outcrop occurs, and then projected it to the whole terrestrial extent of latest Cretaceous North America. MaxEnt compares the environmental conditions at locations of occurrence records with randomly selected points from a background extent to create a machine-learning model of habitat suitability. The dinosaur occurrence data for each time interval were randomly split, using 75% of localities to calibrate the models and 25% to evaluate the models' predictive accuracy. A 50-fold sub-sample procedure was used to calculate AUC statistics (predictive performance measure[24,35]). Jackknife tests and % variable contribution were used to estimate variable importance. In the models reported in the Results section, extrapolation was preferred over clamping to allow more reliable (and biologically congruent) response curves[90]. Replications using clamping to test how truncation of response curves was affecting the distribution of habitat suitability spaces were also produced. Different run types (crossvalidate, bootstrap, and subsample) were also attempted to test for marginal deviations from a single response model. The models that provided the best AUC values (>0.9), and minimised extrapolation in the curves, providing the biologically most sensible responses, were retained to fit the ENMs. As an additional sensitivity test, we used comparisons between forward and backward projections (from Campanian substages to Maastrichtian and vice versa) as an independent test to assess major discrepancies in the spatial distribution models. To quantify deviations from the distributions between the better fitted models in these comparisons, difference maps (in ESRI ArcGIS 10.2.2) were used; models with a coarse consensus (exceeding 15% of deviation) were discarded (the same comparison made with forward-backward projections in sensitivity tests was also implemented). The discarded models only affected forward-backward projections from the early Campanian and early Maastrichtian, probably because of the low number of unique training occurrences for those two time slices (Table 2).

As threshold choice can dramatically alter niche model results (and their interpretation based on binary conversion of suitable/unsuitable areas[91]), we used True Skill Statistic (TSS[92]) as a measure of accuracy, which provides a threshold-dependent statistic for ENMs. TSS values can range from −1 to +1, whereby +1

indicates perfect agreement and values of zero or less indicate a performance no better than random. It has the advantage over other metrics used in ENM of being unaffected by prevalence. We selected the threshold with the highest TSS score (max TSS[92]), which could be detected in all models (0.7). As interpretations of suitability trends can be biased by a single threshold, we also selected the minimum shared TSS score (0.2) and the average between the minimum and maximum (0.45). Selecting multiple thresholds allows to get an idea of the niche dynamics in most and least favourable habitats in our study area for the time series investigated. To allow more conservative comparisons in the binary conversions, only the >0.45 and >0.7 values were used to convert ENMs into binary files, using a script written by us (see Code availability section). Consequently, to quantify suitability in time bins, we measured habitability on each of the binary files produced at the two selected thresholds, obtaining suitability area values with another script written by us (see Code availability section). Model outcrop area layers from United States Geological Survey (USGS: https://www.usgs.gov/) and Paleobiology Database were overlain onto Campanian and Maastrichtian climate layers. After training in an outcrop setting as described above, we projected ENM models in outcrop to continental areas for each of the substages of the Campanian and Maastrichtian, with subsequent suitability quantification in both outcrop training and continental projection areas. ENM analyses were performed with the package dismo[93], using default settings in R version 3.4.4 (R Development Core Team, 2017). Palaeorotation of Cretaceous dinosaur occurrences and outcrop areas is based on the Getech Plc. plate model and methodologies of Markwick and Valdes[84] and Markwick[76]. For our study, we originally included 12 climatic variables, including annual average temperature, annual average precipitation, and splitting the year-simulated parameters of temperature and precipitation in equal time quartiles (coldest, warmest, driest, and wettest) with their relative standard deviations. We used Pearson's correlation test to explore colinearity between variables, retaining only the ones showing a Pearson's correlation coefficient of <0.7 in order to minimise multicolinearity between variables (Supplementary Note 1). The climatic variables included in our modelling were temperatures of the coldest and warmest quartiles, precipitations of the driest and wettest quartiles, and annual temperature standard deviation. These analyses were run in R version 3.4.4. Correlation tables, figures, and further discussion on variables choice are provided in Supplementary Note 1.

**Hotspot analysis.** To define spatial clustering of a geographically biased sample in the latest Cretaceous fossil record of North America, we used hotspot analyses. Identifying hotspots of occurrences is of fundamental importance to understanding spatial biases that might affect the fossil record of a particular taxonomic group. By identifying occurrences hotspots using Geographical Information Systems, a more detailed understanding of that spatial distribution can be gained, including causal effects. The use of hotspot analysis methods is justified following the assumption that the existence of hotspot concentrations suggests spatial dependence between individual fossil occurrences, as their particular occurrence in one place might be due to a set of common taphonomic and diagenetic causes. The most common method used to understand causal distributions of point occurrences is kernel density estimation[94,95], which has several advantages over classic statistical hotspot and clustering techniques such as K-means. In particular, this approach enables the explicit spatial representation of the probability spread of an occurrence. The probability spread is defined as the area around a cluster in which there is an increased likelihood for an occurrence to be there because of spatial dependency. A kernel density space is produced by calculating the mean centre of the occurrence point, generating a symmetrical surface around each marginal point, and then calculating the distance from the mean centre. The same process is repeated for progressively more distant points in the cluster, placing a kernel value for each observation. Summing up these individual kernels then provides the density estimate for the distribution of occurrence points in a cluster[96]. We used the Kernel density tool in ArcGIS 10.2.2 (ESRI) to calculate hotspots, using a 10 m search radius. Additional information on kernel density estimation can be found in Silverman[97]. A complete set of hotspot analysis plots is shown in Supplementary Note 2. In order to classify the incidence numbers as hot and cold spots, we used the optimised hotspot analysis tool in ArcGis 10.2.2 (ESRI). This technique enables the creation of a map of statistically significant hot and cold spots (respectively characterised by positive and negative Z values).

The density surface from the kernel density analysis was used as a raster analysis mask in the environment settings. Significant ($p < 0.05$) clusters scoring Z values higher than 1 for hotspots and lower than −1 for cold spots were used to build a 2 × 2 contingency tables with the virtual taphofacies intervals (see Modelling taphofacies using palaeogeography and palaeoclimate section below).

**Modelling taphofacies using palaeogeography and palaeoclimate.** To explore palaeoenvironmental controls on spatial heterogeneity in depositional environments, we analysed a suite of physical parameters related to fossil preservation. Climatic outputs were used from the same palaeoclimatic general circulation models described above, and using Getech palaeogeographies. In particular, we used sediment flux and surface runoff as they are both parameters that act at several diagenetic phases (transport, disarticulation, burial, and erosion); they also incorporate other climatic agents that in death assemblages affect their preservation in the fossil record (rainfall, temperature, topographic slopes, and palaeo-rivers).

Sediment flux is dependent on geomorphological, tectonic, geographical, and climatic inferences (i.e. basin surface area, topographic relief, temperature, and runoff). We used the BQART predictive model[98], which integrates these agents into a mathematical formula to estimate sediment load in DEM-modelled watersheds. As topographic height is a product of the tectonic relief modelled in the DEM, and climatic models were already in use in this study for the niche modelling, we used these palaeogeographic and palaeoclimatic data to model the sedimentological fluxes in palaeo-river basins in the Campanian and Maastrichtian of North America. Modelling sediment fluxes in a palaeogeographic context provides insights into sediment production, accumulation, and release through the hydrological cycle due to tectonic activity in the sedimentary basin. As such, this provides a metric of sedimentation occurring at a given (or $n$ given) site(s), which fosters the burial of a regional death assemblage and its preservation in the fossil record. Superficial surface runoff, which is known to affect skeleton disarticulation and rock erosion, and thus the preservation of vertebrate remains, was obtained as an output of palaeoclimatic modelling and basin drainage analysis, and projected in space. Palaeoenvironmental data were then categorised into discrete intervals from lower to higher values using Jenks natural break optimisation, which clusters values into different classes, minimising the average deviation of each class, and in so doing reducing both the variance within and between each class. Sediment fluxes (Fig. 5c, d) were split into two intervals (0–280.32 and 280.32–777.68, values in cm/kyr), and surface runoff (Fig. 5e, f) was categorised into two classes (0–1.97 × 10$^{-7}$ and 1.97 × 10$^{-7}$–2 × 10$^{-5}$, all values in mm/s). As our working hypothesis was that every specific interval of this parameter was associated with differential fossil preservation, we named each class a virtual taphofacies. The number of fossil occurrences falling in each taphofacies was then counted for each stage (Campanian and Maastrichtian). We used $\chi^2$-tests (both with and without Yates' continuity correction[99]) to assess any significant association between fossil occurrences and specific physical parameters[19]. Pearson's $\chi^2$-test is a statistical procedure to test whether the observed distribution deviates from the hypothesised null assumption of independence between variables and observations. Fisher's exact test of independence was also used, as it is recommended for contingency tables with low sample sizes[99]. A full table of results of the correlation tests is reported in Table 2, presenting the taphofacies showing statistical association with fossil occurrences.

**Code availability**. The R codes used to perform the statistical tests are available on FigShare (https://doi.org/10.6084/m9.figshare.7609229 and https://doi.org/10.6084/m9.figshare.7609226).

**Reporting summary**. Further information on experimental design is available in the Nature Research Reporting Summary linked to this article.

## Data availability
The authors declare that all the data supporting the findings of this study are available within the paper and its Supplementary Information files, on FigShare (https://doi.org/10.6084/m9.figshare.7609937), and at: http://www.bridge.bris.ac.uk/resources/simulations.

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

## Acknowledgements

We are grateful for the efforts of all those who have generated latest Cretaceous North American dinosaur fossil data, as well as those who have entered these data into the Paleobiology Database, especially Matthew Carrano and John Alroy. Contributors to the silhouettes database PhyloPic.org are also thanked, in particular: T. Michael Keesey, Pete Buchholz, Jack Mayer Wood, Andrew A. Farke, Mariana Ruiz Villarreal, Craig Dylke, and Raven Amos. Fruitful discussions with Erin Saupe (University of Oxford), Susannah Maidment (Natural History Museum), Christopher Dean (Imperial College London), Davide Foffa (National Museum of Scotland), Neftalí Sillero (University of Porto), and Soledad De Esteban Trivigno (Institut Català de Paleontologia) improved this work, and we also thank Stephen Watkins, Sinéad Lyster, and Alexander Whittaker (all from Imperial College London) for providing helpful discussion on sediment fluxes, especially pertaining to implementation of the BQART model. A.A.C. was supported by an Imperial College London Janet Watson Departmental PhD Scholarship. P.D.M. was supported by a Leverhulme Trust Early Career Fellowship (ECF-2014–662) and a Royal Society University Research Fellowship (UF160216). D.J.L. and A.F. acknowledge NERC grant NE/K014757/1, Cretaceous-Paleocene-Eocene: Exploring Climate and Climate Sensitivity. L.A.J. was supported by an Imperial College London President's PhD Scholarship. This is Paleobiology Database official publication number 336.

## Author contributions

A.A.C., P.D.M., and P.A.A. conceived and designed the research; A.A.C., D.J.L., A.F., and S.-J.K. produced and collected data; A.A.C. and L.A.J. analysed the data; A.A.C. produced the figures; A.A.C., P.D.M., and P.A.A. wrote the manuscript. All authors provided critical comments on the manuscript.

## Additional information

**Competing interests:** The authors declare no competing interests.

