## [Peer Review File · Nature Communications]

Reviewers' Comments:

Reviewer #1:

Remarks to the Author:

General Thoughts:

In this contribution the authors attempt to test the hypothesis that dinosaur diversity declined in the several million years leading up to the K/Pg extinction event. They utilize a two-pronged approach to tackling this question: (1) cluster analysis of dinosaur occurrence data along with(?) sediment fluxes and runoff intensity; (2) ecological niche modeling (ENM) based (I think) on available GCM data from the Hadley Center. Their results suggest that the majority of dinosaur diversity throughout the Campanian and Maastrichtian stages is observed from a few, spatially clustered localities. This is interpreted (along with stratigraphic data) to indicate a decrease in preservation potential in the western WIB thru time. ENM results are first presented as clipped to available outcrop area (i.e., a model only testing localities that have the potential to be sampled), and later projected(?) from this clipped training region to the entire North American continent. Whereas a decrease in suitable habitat (SH) is observed in the training regions of these models, the model projections to North America show substantial increase in SH. This is interpreted as evidence for a potential increase in dinosaur diversity leading up to the K/Pg boundary. Taken together, the authors conclude that preservation biases (cluster analysis) obscure a pattern of increasing diversity (ENM analysis).

This is an interesting application of ENM, different from the traditional mode of testing hypotheses of species distributions. However, I am unclear about how these models were constructed (e.g., are they sed-proxy-based PaleoENM models as in the work of Myers et al 2015, Stigall 2012, etc. or are they paleo-applications of climate-model-based ENMs using the temperature and precipitation layers provided by Hadley Center GCMs as in the work of Saupé et al 2014 or Nogues-Bravo et al 2008?) and I also am not convinced that using ENMs for diversity estimation is at all a good idea, particularly when developed using clade-level (vs. species-level) data; this is not their intended purpose and although the biases associated with using these models to interpret distribution patterns are well known and debated, neither the community (nor the authors) have explored the impact of ENM biases on diversity estimation. All in all, I think it's an interesting study question and system, but I have concerns about the methods and interpretation of results based on those methods. For me it is not unequivocal that dinosaur diversity increased and is masked by preservation biases. However, even if I was, I am not convinced that this represents a fundamental change in the way that the field of paleobiology thinks, which is cited as requisite for publication in Nature Communications. Therefore, I recommend rejection of this submission.

Specific Thoughts:

- lines 92-94: downscaling environmental data is a tricky thing to do without introducing artifacts. How was this done and was this potential bias tested for?

- line 199ff: there are some real dangers with using ENMs in this fashion without extreme caution....First, the training region model is a snapshot of what environments taxa are found in ("existing niche", but doesn't necessarily reflect ALL the environments in which those taxa are found (which would reflect the true "fundamental niche"). This is why it is impossible to ever conclude "niche expansion" because one may never know if expansion of a niche in environmental space means the taxon is sampling environments that it always liked, but didn't exist on geography before, or actually experienced modification of those environmental tolerances. Therefore, projection of an ENM to a broader space with no rock record can give you a sense of other SH, but is limited by what environments are present in the training region (i.e., almost certainly an underestimation). Second, this approach is only as good as the climate data and its resolution! Some discussion of the boundary

conditions or simulations of these climate data would help to better understand if these GCMs are doing a good job at reconstructing climate (because without a rock record, there is no independent check of the climate models). Finally, I would never use ENMs to estimate diversity. Their purpose is to estimate suitable abiotic environments...both from the perspective of how the models are run and their biases, this is not a strong way to estimate diversity – it is a strong way to estimate suitable habitat area and its change thru time and space.

- line 131ff: I am unclear under what conditions the Hadley Center GCM-based environmental layers, vs. sed-based layers (DEM, sed flux, runoff) are being used to construct niche models. The former indicates a method identical to modern applications of ENMs (with its inherent biases) and the latter represents the PaleoENM framework described in Myers et al 2015 (with somewhat different biases). How these data were used separately or in concert to train niche models will affect how one is able to interpret the results...please clarify.

- line 148ff: Are the reconstructed “virtual taphofacies” used to constrain how the ENMs are built? If so, please clarify because I missed it in the text. If not, this would be a good idea to reduce bias introduced by different preservation regimes.

- line 173ff: AUC values between 0.7-0.8 do not actually indicate particularly good model performance. For example, see Elith et al 2006 or Franklin 2009 who describe “good” models as those with AUC > 0.9. However, AUC scores also known to be artificially reduced in presence-only models (since the calculation involves metrics of both sensitivity and specificity with the latter not applicable in a presence-only model). A better choice would be the partial-ROC test described in Peterson et al 2008.

- Table 1: I’m not certain why stage-level ENM results are reported here, when the temporal resolution of the study is actually the sub-stage level. This is not just an issue of averaging, these models could show fundamentally different patterns when environments are binned over 2 Myrs vs. 5 Myrs. There is also a pretty significant interpretation difference when one is looking at SH estimated over such different time scales, particular for vertebrate taxa that don’t live more than a few Myrs.

- line 191: I cannot discern where non-analogue climates are shown in Fig 3?

- lines 194-196: I’m not sure that I agree with this justification. The fact that non-analogue climates are distributed in the high latitude N seems like a good enough reason to not worry about them because you will not interpret the models in this region anyway. However, suggesting that the lack of species occurrences in non-analogue climates is a reason for no bias is not correct. The model will still sample the non-analog climates to determine suitability, therefore they can affect the model (most likely by biasing those regions towards being deemed unsuitable).

- line 199: What does “relative area standardization” mean? Are you referring to calculating the percent of available habitat deemed “suitable” compared to the full model extent?

- line 208: Why does Fig 4 come before Fig 3?

- Fig 4: What is the y-axis in this figure – is it percent predicted SH? How are you determining a threshold for suitability? Un-thresholded models cannot be directly compared because the internal calibration of the Maxent output scores is unique for each model. See Liu et al 2013 for a discussion of thresholding options...Many folks use “mean model value” or (more conservatively) “least training presence”. There is also missing a reference to “B” in the caption. This figure is one of the focal points of the paper wherein it shows a decrease in the SH metric when models are trained (and projected?)

only to areas with preserved geologic record, however, models projected to the entire North American continent show substantial increase in the SH metric. I could use some clarification in the text regarding which ENMs are fueling these figures. That is, (1) are both sets of models trained with the same extent and clipped to available outcrop area? (2) in A, am I looking at the training region or a model projected to the North American continent but still clipped to available outcrop area? (3) in B, am I looking at models that were trained with the same clipped-to-outcrop data, but then are projected to the continuous area of North America? The stark contrast of these two estimates of SH is used to support missing diversity because of lack of outcrop...however, all ENMs show is that climate models predict expanding climates within the predicted SH zone for each of their taxa. The connection to diversity is based on a loose reference to the "species-area effect" (line 305), which needs more explicit discussion to be convincing.

- line 224-225: It would be more accurate to stick to ENM terminology vs. "species distribution modeling" because this analysis is at the clade-level and includes an environmental space analysis in Ecospat (vs. purely geography-based analysis as in most SDMs); also helpful to use the same terminology thru-out.

- line 243ff: I'm a fan of using Ecospat to look at niche similarity and equivalency in environmental space (vs. geographic space which is biased by existing environments). However, the meat and potatoes of this test lies in the value of the D-metric and whether the test is statistically significant -- this could be emphasized and interpreted more. The test can also be useful in tracking how niche estimates are changing thru time and/or space. Whereas niche contraction can be easily identified in this setting, niche expansion cannot because the figures (Fig 5 and in the SI) do not separate the environmental cloud unique to each time period (the suite of environments is binned as a single cloud). Therefore, I can't discern whether changes in niche dimensions are due to expansion in environments that existed at both time periods or occupation of (potentially previously) suitable environments that simply didn't exist in the preceding time period. It would be helpful to separate these environmental clouds. Further, why are the Ecospat tests run at the stage-level vs. the sub-stage? Similar to the SH analyses, binning environmental data into larger temporal bins changes how these results can be interpreted.

- Table 2: It would be most helpful to know how many occurrences are being used for each individual model being run. That is, number of occurrences per family, per sub-stage. Without this information it's hard to interpret whether model fidelity may be affected by low sample size.

- line 301: "North AmericaN" ?

- line 302ff: Why are the cluster analyses performed separately for latitude and longitude? Species don't move in only one direction, it seems like you would want to consider the effects of both in concert?

- line 308: I'm not certain that you are performing PaleoENM...? These models are based on GCM climate data, which is standard ENM just applied in deep time (e.g., Nogues-Bravo 2009).

- lines 323-324: Where is a "clear distinction in taphonomic suitability between northern and southern localities" tested? Did I miss this quantification somewhere?

- line 339: "provincialism AS a..." ?

- lines 349ff: How are you discriminating between whether your clipped vs. continuous models are "better"? Based on what data? My understanding is that the same training region (clipped to outcrops)

was used for both projections. If that's the case, then the only difference between these projections is area of outcrop  less outcrop area, less SH in the clipped projections vs. continuous projections. An easy check would also be to confirm that the continuous projections are showing the same decrease in SH in the area of outcrops. If the models are biased by numbers of occurrences (i.e., less occurrences  less SH predicted) then this will bias both projections....

- line 400: "...WIS AND freed..." ?

- lines 423ff: A lot of this discussion is based on the assumption that the species-area effect is real and ubiquitous...the authors have not convinced me that this is necessarily the case.

- lines 431-435: Tectonism and eustasy are also abiotic factors, and happen to be ones that significantly influence GCMs....

- lines 450-451: ENMs are not the best way to statistically compare environments because they are linked to where taxa are (and are not). Better to do a direct comparison of environments thru time using more standard multivariate statistics since you have the GCM outputs.

- lines 484-486: What types of environments are these fossils coming from? Is it reasonable to assume that just because the PBDB record doesn't say "transported" that they haven't been transported?

- lines 489-492: A single paper using family-level data does not seem like substantial evidence of the "wide use" of clade-level analyses. Whereas I understand the utility, and even necessity, of using clades vs. species in deep time ENMs (due to occurrence number constraints), this should not be done without caution. Some explicit discussion here regarding why dinosaur families are good proxies for species-level responses to environmental change, would be very useful.

- line 549: Fenton et al is reference 32 (not 30)

- line 554: "biased sampleS" ?

- lines 591-592: Should cite Owens et al 2013; what do your individual response curves look like? Is there significant (wacky) extrapolation in your models?

- line 592: What does "modelled outcrop ENMs" refer to? How is this different from other models in the study?

- lines 602ff: A more explicit explanation of variable choice would be helpful. There is a substantial body of literature supporting temperature and precipitation as important abiotic variables constraining modern vertebrate distributions - probably even some specific to birds and reptiles which would be important analogs to reference here and support your variable choices.

- line 626: What kind of "comparisons" ?

- lines 631-635: I don't understand how/what is being standardized here; please clarify.

- Regarding potential biases, it would be helpful to include somewhere (could be the SI) a discussion of the effects of temporal bin size on the models and their interpretation. The Campanian and Maastrichtian are substantially different in length and their sub-stages vary from ~ 2.4 Myrs to 5 Myrs in duration. This is a 2-fold difference in time bin size. Does the size of the time bin correlate to

number of occurrence points? (suggesting potential bias) How about outcrop area and bin duration?

Refs Mentioned

Elith, J., C. H. Graham, R. P. Anderson, M. Dudík, S. Ferrier, A. Guisan, R. J. Hijmans, F. Huettmann, J. R. Leathwick, A. Lehmann, J. Li, L. G. Lohmann, B. A. Loiselle, G. Manion, C. Moritz, M. Nakamura, Y. Nakazawa, J. M. Overton, A. T. Peterson, S. J. Phillips, K. Richardson, R. Scachetti-Pereira, R. E. Schapire, J. Soberón, S. Williams, M. S. Wisz, and N. E. Zimmermann. 2006. Novel methods improve prediction of species' distributions from occurrence data. *Ecography* 29:129–151.

Franklin, J. 2009. *Mapping species distributions: spatial inference and prediction*. Cambridge University Press, Cambridge, UK.

Liu, C., M. White, and G. Newell. 2013. Selecting thresholds for the prediction of species occurrence with presence-only data. *Journal of Biogeography* 40:778–789.

Nogués-Bravo, D. 2009. Predicting the past distribution of species climate niches. *Global Ecology and Biogeography* 18:521-531.

Owens, H. L., L. P. Campbell, L. L. Dornak, E. E. Saupe, N. Barve, J. Soberón, K. Ingenloff, A. Lira-Noriega, C. M. Hensz, C. E. Myers, and A. T. Peterson. 2013. Constraints on interpretation of ecological niche models by limited environmental ranges on calibration areas. *Ecological Modelling* 263:10–18.

Peterson, A. T., M. Papeş, and J. Soberón. 2008. Rethinking receiver operating characteristic analysis applications in ecological niche modeling. *Ecological Modelling* 213:63–72.

Reviewer #2:

Remarks to the Author:

Chiarenza and colleagues tackle the oft-debated question of whether dinosaur diversity was in long-term decline prior to the Cretaceous/Paleogene boundary. Instead of focusing on diversity itself, they used ecological niche modelling to examine the extent to which dinosaur habitat changed in concert with observed diversity changes. Chiarenza and team also examined sediment flux and surface runoff in both the Campanian and Maastrichtian of North America to quantify potential taphonomic biases affecting understanding of dinosaur diversity over this interval. The study will be of broad relevance to readers given deep interest in the evolutionary dynamics of this clade. Moreover, Chiarenza and colleagues use a novel approach to try to understand potential taphonomic biases influencing our interpretation of diversification patterns through time. I note a number of concerns and suggestion below.

Overall comments

1. Perhaps my most relevant comment is that ecological niche modelling (ENM), or species distribution modelling (SDM), can provide insight into whether the distributional potential of a group was declining or expanding in terms of climatic niche footprint, but this decline or expansion does not necessarily translate into diversity decrease or increase, respectively. More specifically, ENM can tell you that the climatic conditions for a clade were broadly suitable, and therefore potential declines in diversity were not likely due to environmental degradation. However, and importantly, diversity can decline for other

reasons (for example, from reduced rates of speciation or extinction from biotically-mediated forces). The ENM analyses may be best shown in concert with sample-standardized diversity curves (which have been done previously) and sediment flux and surface runoff models, which are almost more convincing as to why we may have reduced ability to sample dinosaur diversity in the Maastrichtian. The focus is, at the moment, strongly on the ENM analyses, but perhaps it should be more equally-weighted among all three analyses (sample-standardization, taphonomic modelling, and ENM modelling), all three together of which show why we cannot say that diversity declined prior to the K/Pg.

2. SDMs/ENMs are modelled at the clade level for the Ceratopsids, Hadrosaurids, and Tyrannosaurids to provide a general overview of the degree to which suitable habitat changed over time for these clades. My question is whether similar patterns would be obtained if models were calibrated at the species level and summed to the clade level. Would increases in suitable habitat for each group still be obtained? Or, would a more nuanced picture be obtained as some species' habitat decreased and other species' habitat increased?

3. Was any statistical analysis done to show the association of fossil occurrences with lower sediment flux?

4. The analyses on niche occupation and niche stability are interesting, but they seem out of place and do not contribute to the argument at hand (although apologies if I am missing something critical here). These analyses should perhaps be better integrated and/or clarified, or simply removed.

Other comments

1. Fig 2: some specimens are found in the middle of the WIS. Why this occurs should be briefly mentioned.

2. More introduction is needed on why a cluster analysis was performed (even if just briefly) in both the main text results and methods and in the supplementary materials. With regard to the latter, the results could be better presented (for example in a Table?).

3. It might be useful to include discussion of the number of dinosaur taxa currently found from Laramidia in each of the three clades. A similar table could be produced as to Table 2, but focused on different regions.

4. It might be easier for readers to comprehend the different modelling projections/analyses with a chart or table. For example, the authors calibrated models on outcrop area only, and then performed numerous projections in space and time. The series of analyses could potentially be more easily understand with a schematic?

5. Line 92: The authors indicated palaeoenvironmental data had been interpolated to a finer resolution: what resolution? Also, this interpolation should be reiterated in the palaeoclimate model section.

6. Line 198: How do you quantify suitability here? The methods section indicates cells with values > 0.5 were chosen, but no rationale was provided. Choice of threshold may dramatically alter niche model results (see, e.g., Lui et al. 2005 or 2013), and ideally multiple thresholds should be compared and results assessed for congruence.

7. Line 199: How was relative area standardized?

8. Line 236: Perhaps include a brief explanation / introduction of niche gradients with ordination techniques.

9. Line 259: Were occurrences used for the niche overlap analyses, or were the model results? Please clarify.

10. Line 295: Was rock outcrop area in the time slices quantified? This would be interesting and relevant to present.

11. Line 445: A parenthesis is missing.

12. Lines 451-8: The wording and reasoning here is a bit unclear.

13. Line 489: Are the number of occurrences cited spatially unique? It seems unlikely from the

resolution of the data. How many spatially-unique occurrences (at the resolution of the climate data) were used for each clade in each time slice? Perhaps these data should be listed in a table.

14. Lines 554-559: Other authors have found clustered fossil occurrences through time (see, for example, Plotnick, 2017, *Geology*).

15. Table 2 – Could diversity counts be shown after standardising for sampling (e.g., using SQS)? This may elucidate patterns and further augment your position (see overall comment #1).

16. Fig S1 and S2 – It's difficult to see the occurrences in these figures. Could they be made bigger?

17. Fig S3 – A solid red contour line is referred to in the ECOSPAT analyses, but it does not seem to be present on the figures.

18. Fig S4+ – These figures should be explained in more detail here, in the Methodology, and in the Results. From my reading, it seems there is no support for niche equivalency, which makes sense, but that there is mild evidence for niche stability through time (at the clade level). While these analyses are certainly interesting, it's unclear how they add to the manuscript in its current form (see overall comment #4).

19. Does the variable contribution to each model make sense based on inferences of the biology/ecology of these clades? Perhaps this should be discussed. A large table of these results is presented, but they are not discussed or made much of with regard to the main arguments (and indeed, they do not seem particularly relevant to the main points of the paper). This entire discussion and the associated results could likely go in the supplementary materials? Unless, of course, the idea is to comment on what conditions are important for each clade, but then some discussion is needed with regard to whether the particular variables make sense based on what is known of the clades in question.

20. The authors modelled suitable habitat into the Danian and showed an increase or at least stability in amount of available habitat. However, the conditions immediately after the K-Pg impact would have been very unsuitable, and this unsuitability would have persisted for some time, leading to the demise of the non-avian dinosaurs. Perhaps this should be made clearer?

21. The authors found an increase in suitability for Hadrosaurids into the Maastrichtian when considering just outcrop area, but hadrosaurids show a decline in diversity similar to the other clades using the raw taxon count data. Given that the authors claim there is a link between suitability and diversity, shouldn't a decline in hadrosaurid diversity be surprising here? Perhaps more discussion of this is warranted?

22. The authors project their models in both space and time, with extrapolation enabled. This means the models will extrapolate suitability when encountering combinations of climatic conditions not observed in the model calibration region. When extrapolation is enabled, the Maxent algorithm will produce maps of these regions in space. It might be useful for the authors to present these maps. Moreover, what happens with regard to amount of suitable habitat if extrapolation is NOT allowed?

Reviewer #3:

Remarks to the Author:

Review of Chiarenza et al. "Ecological niche 1 modelling indicates no decline in dinosaur diversity prior to the Cretaceous/Paleogene mass extinction"

This is an excellent, innovative study addressing a longstanding debate in dinosaur paleobiology: to what extent were non-avian dinosaurs 'on the decline' prior to the end-Cretaceous mass extinction? This question has been flogged for many years with conflicting results, and in many ways has grown stale since the underlying data for all such studies are subject to the same sampling biases.

Chiarenza and colleagues attempt to circumvent these limitations by incorporating an ecological niche modelling approach, allowing them to investigate how apparent declines in raw dinosaur diversity in

the latest Cretaceous of North America compare with changes in suitable dinosaur habitat, and changes in the abundance of suitable dinosaur-bearing rock through this interval.

The results they present are convincing, and move the study of Late Cretaceous dinosaur diversity trends substantially forwards. Moreover, I think the study will serve as an exemplar of the power of integrating cutting-edge niche modeling approaches with the study of paleodiversity.

The authors illustrate convincingly that the apparent decline in suitable dinosaur habitat from the Campanian to the Maastrichtian is likely to be an artifact of limited rock exposure, and that, when continental-scale trends are considered, habitat suitability for the major clades of non-avian dinosaurs may have increased throughout the latest Cretaceous of North America.

Importantly, the authors argue that the same low-stand eustatic conditions that generated an increase in habitat suitability towards the Maastrichtian yielded poorer overall conditions for fossil preservation. This trade-off provides a satisfying explanation for a significant amount of the longstanding disagreement among palaeontologists related to whether or not non-avian dinosaurs were on a long-term decline throughout the latest Cretaceous.

A secondary, but additionally interesting insight from the authors' work shows that ENM estimates of habitat suitability in the earliest Palaeocene show no declines for the three major lineages of non-avian dinosaurs under investigation, providing additional support for a geologically sudden and catastrophic event as the principal driver of non-avian dinosaur extinction.

One caveat: although I am familiar with ENM approaches and paleobiology database-style analyses, I am not an expert in either of these techniques, and feel that the manuscript will need to be appraised on a more technical level by ENM and paleoDB specialists.

The study is generally well-written, although it could benefit from more careful proof-reading for style and technical errors at times. For example, the first line of the Introduction is somewhat confusingly worded, and should be rewritten in my opinion.

Last, the trans-K-Pg results presented in this study are interesting, and I agree with the authors that these results bear on the explosive post-extinction radiation of crown birds and mammals. In addition to the classic references listed, I think referencing more recent and convincing evidence for rapid early Cretaceous radiations of mammals and birds would be suitable here, such as Prum et al. 2015 for birds and O'Leary et al. 2013 for mammals.

I've flagged a small number of minor grammatical issues below, but I didn't pay particularly close attention to these details. The authors should give the writing a close look to expunge similar errors that I'm sure I've missed.

Line 19: remove comma after window

Line 51: delete 'for' after mitigate

Line 445: close parenthesis.

10 August 2018

Dear Editor and Reviewers,

Below, we respond to each individual point raised. Our modifications to the original manuscript are attached in a tracked format, and a “clean” version is also submitted. Comments from the referees here are in *italics*, while our responses are in **bold**.

Yours sincerely,

Alfio Alessandro Chiarenza

On behalf of all the other authors

Reviewers' comments:

Reviewer #1 (Remarks to the Author):

General Thoughts:

In this contribution the authors attempt to test the hypothesis that dinosaur diversity declined in the several million years leading up to the K/Pg extinction event. They utilize a two-pronged approach to tackling this question: (1) cluster analysis of dinosaur occurrence data along with(?) sediment fluxes and runoff intensity; (2) ecological niche modeling (ENM) based (I think) on available GCM data from the Hadley Center. Their results suggest that the majority of dinosaur diversity throughout the Campanian and Maastrichtian stages is observed from a few, spatially clustered localities. This is interpreted (along with stratigraphic data) to indicate a decrease in preservation potential in the western WIB thru time. ENM results are first presented as clipped to available outcrop area (i.e., a model only testing localities that have the potential to be sampled), and later projected(?) from this clipped training region to the entire North American continent. Whereas a decrease in suitable habitat (SH) is observed in the training regions of these models, the model projections to North America show substantial increase in SH. This is interpreted as evidence for a potential increase in dinosaur diversity leading up to the K/Pg boundary. Taken together, the authors conclude that preservation biases (cluster analysis) obscure a pattern of increasing diversity (ENM analysis).

This is an interesting application of ENM, different from the traditional mode of testing

hypotheses of species distributions. However, I am unclear about how these models were constructed (e.g., are they sed-proxy-based PaleoENM models as in the work of Myers et al 2015, Stigall 2012, etc. or are they paleo-applications of climate-model-based ENMs using the temperature and precipitation layers provided by Hadley Center GCMs as in the work of Saupe et al 2014 or Nogues-Bravo et al 2008?) and I also am not convinced that using ENMs for diversity estimation is at all a good idea, particularly when developed using clade-level (vs. species-level) data; this is not their intended purpose and although the biases associated with using these models to interpret distribution patterns are well known and debated, neither the community (nor the authors) have explored the impact of ENM biases on diversity estimation. All in all, I think it is an interesting study question and system, but I have concerns about the methods and interpretation of results based on those methods. For me it is not unequivocal that dinosaur diversity increased and is masked by preservation biases. However, even if I was, I am not convinced that this represents a fundamental change in the way that the field of paleobiology thinks, which is cited as requisite for publication in Nature Communications. Therefore, I recommend rejection of this submission.

First of all, we would like to thank the reviewer for finding our study system interesting and worth reviewing. We have tried to make the manuscript clearer. As we have been using outputs from General Circulation Models to build our ecological niche models, we have replaced any reference of Palaeo-ENM (*sensu* Mayers et al 2015) with deep time ENM (e.g. as in Saupe et al 2014 or Nogues-Bravo et al 2008). As stated below in response to more specific comments, we have also added details to description of the GCM, also providing R plots for the environmental predictors used in this study and the statistics to select them as a supplementary file (S.3). As dinosaur palaeontologists, we disagree in not retaining investigation on macroecological dynamics toward one of the most important (and surely the most iconic) mass extinction in the history of life on Earth an interesting topic.

We would also like to remark that palaeontological applications of Earth System Modelling and Ecological Niche Modelling are quite rare so far, particularly regarding vertebrates. We not only hope to take the lead in a more widespread use of these methodologies to Deep Time studies, but also make sure that the novelties brought from quantitative ecology to palaeontology may add some more experimental component in a subject which has always been considered as an historical science, with the impossibility of testing experimentally ad hoc conceived hypotheses.

Specific Thoughts:

- lines 92-94: downscaling environmental data is a tricky thing to do without introducing artifacts. How was this done and was this potential bias tested for?

Regarding the downscaling to 1.25° x 1.25° – although we know from Waterson et al. 2016 that downscaling to this resolution does not alter ENM outputs, we have run some preliminary tests on the original grid format resolution of the outputs (3.75° x 2.5°). Although the coarse resolution reduces the number of unique training datapoints, negatively affecting model evaluation metrics (AUC between 0.6 and 0.8 and low TSS

~0.5), they still preserve the spatial spread as in the finer resolution run, and they broadly agree in areal extent with the higher resolution outputs used in the study. We would also like to remark that the paleogeographic boundary conditions the GCM are run with, originally created on a $0.5^\circ \times 0.5^\circ$ degree grid and then upscaled to the model resolution ($3.75^\circ \times 2.5^\circ$). This means that topographic and bathymetric information is broadly conserved, as it is resolved at a lower resolution and strengthens interpretations at low resolution in the climatic grid, meaning that there is less uncertainty incorporated into the downscaled model climate variables. Furthermore, the sub-grid scale orographic features of the topography are still calculated within the model (and used, for instance, in the gravity wave-drag scheme in the model, allowing for finer scale features to have an impact on the climate signal). As this paper also looks at the regional scale features by using a novel data-model approach, the regional and large-scale circulation (with associated energy, momentum fluxes and how they change stage-on-stage), are resolved in the model, which is the important determinant on the climate signal. Please note that in this contribution, we have included as supplementary file (S.3) R plots with the environmental predictors used in this study and the correlation analyses executed on the full set of GCM output variables.

- line 199ff: *there are some real dangers with using ENMs in this fashion without extreme caution....First, the training region model is a snapshot of what environments taxa are found in (“existing niche”, but does not necessarily reflect ALL the environments in which those taxa are found (which would reflect the true “fundamental niche”). This is why it is impossible to ever conclude “niche expansion” because one may never know if expansion of a niche in environmental space means the taxon is sampling environments that it always liked, but did not exist on geography before, or actually experienced modification of those environmental tolerances. Therefore, projection of an ENM to a broader space with no rock record can give you a sense underestimation). Second, this approach is only as good as the climate data and its resolution! Some discussion of the boundary conditions or simulations of these climate data would help to better understand if these GCMs are doing a good job at reconstructing climate (because without a rock record, there is no independent check of the climate models). Finally, I would never use ENMs to estimate diversity. Their purpose is to estimate suitable abiotic environments...both from the perspective of how the models are run and their biases, this is not a strong way to estimate diversity – it is a strong way to estimate suitable habitat area and its change thru time and space.*

1) We have toned down our MS to better highlight our consideration of uncertainty and of bias intrinsic in our dataset and methodology, as we try to explain in these replies. We do agree that outcrop training may restrict the full possibility of training points in space. For this, we have altered our main experimental system by training on the whole North American extent (as we hope to have made clear throughout the method and discussion of the updated manuscript) and then, as a comparison with suitability trends in outcrop area still remains central to the discussion in this paper, projected to available geological extent in each time bin. We would also like to remark that, although we do agree that “training region model is a snapshot of what environments taxa are found in (“existing niche”, but does not necessarily reflect ALL the environments in which

those taxa are found (which would reflect the true “fundamental niche”)” this may be more the case in modern-day ENM. As already mentioned by Meyers et al 2015, it is a different matter in deep time ENM. Although it is true that training region may be computationally only a snapshot, the same “time-averaging” component stated as a limitation of the palaeoclimatic data and to the fossil occurrences (depending on their time resolution) might actually be a key point of advantage here: representing occurrences and training regions distributed across several million years can properly capture ranges shifting in response to abiotic drivers in extended time windows, approximating ALL the environments in which those taxa have been living (their fundamental niche) at an evolutionary time scale. Hence, we contend that this is appropriate for this experimental setting and this kind of deep time investigation.

2) How model-dependent are our results? This question is challenging to address without repeating our simulations with several different climate models. Unfortunately, this is not possible in the scope of this manuscript (for instance, these simulations took over two years to complete on a high performance super computer). However, confidence in the robustness of our results can be obtained by the fact that (a) the internal dynamical changes in the model are self-consistent and agree with fundamental theory. For instance the relative small changes in paleogeography between different geologic stages do not cause fundamental shifts in global or regional climate between period where there is a more robust rock record to compare against; and (b) the model (used extensively in IPCC AR3-5) itself is well evaluated for the Modern (see Valdes, et al. 2017) when compared to observations and against other IPCC CMIP5 models of which it still out perform many more contemporary, higher fidelity models (Valdes, et al. 2017).

3) CO₂ concentrations are within the range of uncertainty provided by the latest proxy pCO₂ reconstructions of Foster, et al. (2017).

4) Solar luminosity is stage specific and calculated using the methods of Gough (1981).

5) These paleogeographic reconstructions allow for far greater temporal resolution than other reconstructions, unlike many previous studies that only use one palaeogeography. The palaeo-digital elevation models used as boundary conditions in the model for each stage are informed by these reconstructions, which are in turn constrained by extensive geological databases (both public and private from in-house exploration and cores from the oil industry). These data include published lithologic, tectonic and fossil studies, the lithologic databases of the Palaeogeographic Atlas Project (University of Chicago), and deep sea (Deep Sea Drilling Project (DSDP)/ODP) data, as well as data such as from the Atlas projects databases.

6) See comment above.

- line 131ff: *I am unclear under what conditions the Hadley Center GCM-based environmental layers, vs. sed-based layers (DEM, sed flux, runoff) are being used to*

construct niche models. The former indicates a method identical to modern applications of ENMs (with its inherent biases) and the latter represents the PaleoENM framework described in Myers et al 2015 (with somewhat different biases). How these data were used separately or in concert to train niche models will affect how one is able to interpret the results...please clarify.

We apologise for the lack of clarity on this main point in our study, and we have added relevant information both in the discussion and the material and methods section. Although this has been discussed already in previous comments, we restate here that this is a GCM based study, where output from the HadCM3L version 4.5 Atmospheric–Ocean General Circulation Model (AOGCM60) has been used. The specific version of the model is HadCM3BL-M2.1aE following the nomenclature of Valdes et al. 2017. We have added other relevant references of case study and model description, where all information on the models can be found throughout the text.

- line 148ff: Are the reconstructed “virtual taphofacies” used to constrain how the ENMs are built? If so, please clarify because I missed it in the text. If not, this would be a good idea to reduce bias introduced by different preservation regimes.

We thank the reviewer for this suggestion. Originally we had set our study system using the taphofacies to constrain the training region for the study, encountering two main problems for which we opted in the end to the current study setting. Firstly, analogously to what happened in outcrop training models, fitting the niche on outcrop extent provided a relatively lower estimate for model evaluation metrics (AUC and TSS), and plenty of spurious extrapolation in the response curves. Second, as one of the main aims of this study is to constrain and better detail the effect of spatial bias in the fossil record for our understanding of palaeodiversity, we suspected that constraining the niche models in this way for then using virtual taphofacies for highlighting “poorer and better” areas for fossil preservation potential, might have created a circularity problem, making redundant and poorly constrainable our ecological model outputs with the taphonomical ones and vice versa.

- line 173ff: AUC values between 0.7-0.8 do not actually indicate particularly good model performance. For example, see Elith et al 2006 or Franklin 2009 who describe “good” models as those with AUC > 0.9. However, AUC scores also known to be artificially reduced in presence-only models (since the calculation involves metrics of both sensitivity and specificity with the latter not applicable in a presence-only model). A better choice would be the partial-ROC test described in Peterson et al 2008.

We agree with the reviewer and this is one of the reasons why in this resubmission we opted for using continental training and projected to outcrop rather than the original opposite approach. Following this, all of the AUC values are now above the recommended 0.9 threshold. In addition, to follow the reviewer’s suggestion in using a second metric of model evaluation, instead of the now updated (particularly in light of the newest MaxEnt update [3.4.4]) ROC test we opted for the TSS test (Allouche et al. 2006), which also makes it easier to establish which suitability threshold to use for

binary conversion by using the max TSS (particularly for presence only methods). Notice that this meets another reviewer's request in using a clearer method to establish a threshold to make models comparison and quantification in time bins easier and more objective, as choice of threshold may dramatically alter niche model results (Liu et al. 2005, 2013), and using multiple thresholds is better in order to interpret the results.

- Table 1: I'm not certain why stage-level ENM results are reported here, when the temporal resolution of the study is actually the sub-stage level. This is not just an issue of averaging, these models could show fundamentally different patterns when environments are binned over 2 Myrs vs. 5 Myrs. There is also a pretty significant interpretation difference when one is looking at SH estimated over such different time scales, particular for vertebrate taxa that do not live more than a few Myrs.

We removed the table following the suggestion of another reviewer and instead replaced it with a unique spatial occurrence one, highlighting the single cells occupied by occurrences for each time bin. In addition, we have now run all the analyses to substage level.

- line 191: I cannot discern where non-analogue climates are shown in Fig 3?

Non-analogue climates are only produced during projection on other different time slices/palaeogeographies, so they are only present in the Danian projections (as they have been produced from the training in the late Maastrichtian). The non-analogue projection set in the middle of the North American continent has been added with a dashed line to Figure 3 b.

- lines 194-196: I'm not sure that I agree with this justification. The fact that non-analogue climates are distributed in the high latitude N seems like a good enough reason to not worry about them because you will not interpret the models in this region anyway. However, suggesting that the lack of species occurrences in non-analogue climates is a reason for no bias is not correct. The model will still sample the non-analog climates to determine suitability, therefore they can affect the model (most likely by biasing those regions towards being deemed unsuitable).

We removed the erroneous part of this sentence, and thank the reviewer for the useful suggestion, which we used to complement the correct portion of the lines in “No fossil occurrence falls within the non-analogue regions, therefore, we retained these areas in the environmental predictor layers, as the models in these regions are not interpreted herein.”

- line 199: What does “relative area standardization” mean? Are you referring to calculating the percent of available habitat deemed “suitable” compared to the full model extent?

We agree with the reviewer on the confusion we caused with this method for calculating relative suitability in each time bin. As we have explained in our initial submission, we had originally measured the areas of cells above a 0.5 threshold of suitability and then standardised the amount of suitable area by dividing it to the whole extent of the continent (removing the effect of a different – and wider - land palaeogeography for the

Maastrichtian) and making values between stages comparable. In addition to the request for clarification, we also encountered a further suggestion from another reviewer who made the good point that, not only was this quantification method unclear, but also that the choice of threshold may dramatically alter niche model results (see, e.g., Lui et al. 2005 or 2013), and that ideally multiple thresholds should be compared and results assessed for congruence. For this reason we decided to use a more standard method for quantification based on a simple metric measure of area occupied by suitable cells (like in Nogués-Bravo 2008), and measured taking in consideration multiple thresholds. These thresholds were established using pattern congruence and comparable model performance by measuring TSS values, particularly selecting the maximum TSS value (0.7) which was shared by all the models, and then used to evaluate suitability thresholds to be applied in the different substage models to measure habitability changes between time bins. To select multiple thresholds we also selected the lowermost value 0.2 and an average between the two (0.45). To allow more conservative comparisons plotted in suitability trends quantification in time bins, only the >0.45 and >0.7 values were used to convert ecological niche models into binary files. For binary conversion we used this script written by us, which we included as supplementary information (S. 5) and it is freely available at [https://github.com/LewisAJones/ENM/blob/master/Convert to binary prediction.R](https://github.com/LewisAJones/ENM/blob/master/Convert%20to%20binary%20prediction.R).

Consequently, to quantify suitability in time bins, we measured habitability on each of the binary files produced at the two selected thresholds, obtaining suitability area values with another script written by us, included as well as supplementary information (S.6) and freely available at [https://github.com/LewisAJones/ENM/blob/master/Calculate area.R](https://github.com/LewisAJones/ENM/blob/master/Calculate%20area.R). All ENM models were constructed using maximum entropy algorithms implemented in MaxEnt 3.4.1, and the package ‘dismo’, using default settings in R version 3.4.4 (R Development Core Team, 2017). We integrated all these relevant information on methods in the appropriate section in the manuscript.

- line 208: Why does Fig 4 come before Fig 3?

We have corrected this and fixed a small number of similar issues. To avoid this, we have removed the figures from the main text of the manuscript in this resubmission, and supplied them as separate files.

- Fig 4: What is the y-axis in this figure – is it percent predicted SH? How are you determining a threshold for suitability? Un-thresholded models cannot be directly compared because the internal calibration of the Maxent output scores is unique for each model. See Liu et al 2013 for a discussion of thresholding options...Many folks use “mean model value” or (more conservatively) “least training presence”. There is also missing a reference to “B” in the caption. This figure is one of the focal points of the paper wherein it shows a decrease in the SH metric when models are trained (and projected?) only to areas with preserved geologic record, however, models projected to the entire North American continent show substantial increase in the SH metric. I could use some clarification in the text regarding which ENMs are fueling these figures. That is, (1) are both sets of models trained with the same extent and clipped to available outcrop area? (2) in A, am I looking at the training region or a model projected to the North American continent but still clipped

to available outcrop area? (3) in B, am I looking at models that were trained with the same clipped-to-outcrop data, but then are projected to the continuous area of North America? The stark contrast of these two estimates of SH is used to support missing diversity because of lack of outcrop...however, all ENMs show is that climate models predict expanding climates within the predicted SH zone for each of their taxa. The connection to diversity is based on a loose reference to the “species-area effect” (line 305), which needs more explicit discussion to be convincing.

As discussed in a previous comment, Figure 4 in the former version wasn't clear enough to explain our aim, plus used a metric for habitat suitability that two reviewers found awkward and difficult to follow. We acknowledge this and for this reason we have now plotted habitat suitability metrics in a different way (also check comments and addition to the manuscript for an updated – and hopefully simpler to follow – way of quantifying habitat suitability). We have updated the caption in Figure 3 (formerly Figure 4) to add these information following the indications of the reviewer.

Regarding the scepticism on the Species Area Relationship (SAR) and our use of it in this paper, we acknowledge that the former reference was quite loose but have expanded on that in the paper and we'll try to explain here what we meant. Our argument is that we are effectively predicting wide areas of habitat suitability that did not make it into the geological record, hence leaving a gap in our palaeontological knowledge of that area. As we have shown with hotspot analyses and others have reported in the literature before (e.g. Close et al 2017; Plotnick, 2017), fossil localities are spatially clustered in few areas. We have highlighted that just a small number of groups of geographically localised collecting sites account for the Cretaceous North American dinosaur record. This low spatial variance has implications for diversity estimates, particularly when we highlight how much of the hypothetically habitable area of one of the most highly sampled and most highly diverse palaeoareas is missing in the rock record. Species area seem to covaries with species richness, although it is still not clear whether diversity increases purely as a function of greater area, and/or because sampling of more heterogeneous habitats increases with a larger area – this is far beyond the aim and scope of our paper, but it has been repeatedly demonstrated that species richness positively covaries with area, and seems to find support from modern ecologists (e.g. Killimanis 2008 [and references therein]). We have shown that Maastrichtian fossil occurrences become increasingly more localised than Campanian ones. As they occupy an overall more limited area, this more restricted geological sample may also hide a heterogeneity of habitat. This variety of habitats potentially could have hosted a suite of species which has now become undetectable to palaeontologists (Lyson and Longrich 2011). Most dinosaur-bearing collections in the Western Interior Basin are restricted to lowland-floodplain environments, and therefore preserve a limited subset of depositional environments and thus fossil-bearing lithologies (and palaeoenvironments). As the chance for suitable biotopes increases, we wonder how much of the species richness of the Maastrichtian record are we missing.

- line 224-225: It would be more accurate to stick to ENM terminology vs. “species distribution modeling” because this analysis is at the clade-level and includes an environmental space analysis in Ecospat (vs. purely geography-based analysis as in most SDMs); also helpful to use the same terminology thru-out.

We have changed all the “species distribution” terminology into “ecological niche modelling” terms accordingly.

- line 243ff: I'm a fan of using Ecospat to look at niche similarity and equivalency in environmental space (vs. geographic space which is biased by existing environments). However, the meat and potatoes of this test lies in the value of the D-metric and whether the test is statistically significant -- this could be emphasized and interpreted more. The test can also be useful in tracking how niche estimates are changing thru time and/or space. Whereas niche contraction can be easily identified in this setting, niche expansion cannot because the figures (Fig 5 and in the SI) do not separate the environmental cloud unique to each time period (the suite of environments is binned as a single cloud). Therefore, I can't discern whether changes in niche dimensions are due to expansion in environments that existed at both time periods or occupation of (potentially previously) suitable environments that simply did not exist in the preceding time period. It would be helpful to separate these environmental clouds. Further, why are the Ecospat tests run at the stage-level vs. the sub-stage? Similar to the SH analyses, binning environmental data into larger temporal bins changes how these results can be interpreted.

As another reviewer has made a similar suggestion, we have made some changes in the current version of the manuscript to focus more and justify the use of ecospat in this study. First of all we have changed the analyses to substage level, following the suggestion of temporal consistency by the reviewers. In addition, we compared the two most suitable and highly diverse substages, the late Campanian and the late Maastrichtian. We have reported D-metric and *p*-values both in the main text and the figures. We think that these analyses support a strong similarity of environment between the two substages, supporting also our view that habitat degradation during the last ten million years of the Cretaceous was not responsible for a diversity drop and that the pre-extinction environments could have potentially hosted a high richness of dinosaur taxa.

In addition, as it has been pointed out also by another reviewer, we acknowledge that a single ecospat plot showing the combined environmental cloud for each of the three taxon might not be the clearest figure to follow. So as to keep to the figure limit, without compromising the clarity of the plots, we have decided to move entirely all the plots to the supplementary information, adding some additional description to the analyses in the Method section of the manuscript and the supplementary material (S.4). You can find the full set of niche ordination analyses in the supplementary information file S.4.

- Table 2: It would be most helpful to know how many occurrences are being used for each individual model being run. That is, number of occurrences per family, per sub-stage. Without this information it is hard to interpret whether model fidelity may be affected by low sample size.

We thank the reviewer for their suggestion, and we have updated Table 2 accordingly.

- line 301: “North AmericanN” ?

Thanks for spotting this: fixed.

- line 302ff: Why are the cluster analyses performed separately for latitude and longitude?

Species do not move in only one direction, it seems like you would want to consider the effects of both in concert?

We are fully aware that terrestrial animals can move around (and eventually die and be preserved) in two dimensions, but in the previous version of this submission, we used the cluster and outlier analyses (using the Anselin Local Moran's and k-means clustering on palaeocoordinate data). This method has to be weighted under a single parameter for which the clustering is weighted (hence, why we were using latitude and longitude separately as input fields). On the other hand, we acknowledge the request of the reviewer and designed a different method to satisfy this request. Instead of a classic cluster analysis, we moved for this resubmission to hotspot analyses. In this way, the relative position (a combination of both coordinates) is taken into account to classify groups of occurrences, rather than using a single input to weight the cluster. We detail the methodology (listing relevant references) in the method section, but just briefly here: we used the hotspots analyses guided by the assumption that the existence of concentrations of fossil occurrences in one place are due to a set of common preservational causes. We used Kernel density estimation, which has several advantages on classic statistical hotspot and clustering techniques such as K-means. It provides a 'probability spread' of an occurrence (the area around a cluster in which there is an increased likelihood for an occurrence to be there because of spatial dependency—a common spatial cause). A Kernel density space is produced by calculating the mean centre of the occurrence point, generating a symmetrical surface around each marginal point, and then calculating the distance from the mean centre. The same process is repeated for progressively more distant points in the cluster, placing a kernel value for each observation. Summing up these individual kernels then provides the density estimate for the distribution of occurrence points in a cluster (Fotheringham et al., 2000; Silverman 1986).

- line 308: I'm not certain that you are performing PaleoENM...? These models are based on GCM climate data, which is standard ENM just applied in deep time (e.g., Nogues-Bravo 2009).

As stated in a previous comment, we have modified the terminology to adapt to a “deep time ENM” approach.

- lines 323-324: Where is a “clear distinction in taphonomic suitability between northern and southern localities” tested? Did I miss this quantification somewhere?

We have now added to our virtual taphofacies modelling statistical support (via Pearson's correlation test) for the association of occurrences with selected quantitative intervals of sediment fluxes and surface runoff. As it can be clearly seen in Figure 2 d, f (and particularly comparing them to Figure 2 c, e), southern localities are characterised by a limited extent of “virtual taphofacies” recognised in this study as more suitable. This has been suggested qualitatively before, as cited in the text.

- line 339: “provincialism AS a...” ?

Fixed.

- lines 349ff: *How are you discriminating between whether your clipped vs. continuous models are “better”? Based on what data? My understanding is that the same training region (clipped to outcrops) was used for both projections. If that’s the case, then the only difference between these projections is area of outcrop  less outcrop area, less SH in the clipped projections vs. continuous projections. An easy check would also be to confirm that the continuous projections are showing the same decrease in SH in the area of outcrops. If the models are biased by numbers of occurrences (i.e., less occurrences  less SH predicted) then this will bias both projections...*

As we have stated before, we have now changed the way we fitted the ecological niche models, the projections, and the quantification process, making it more simple to understand. We hope these changes satisfy the reviewer.

- line 400: *“...WIS AND freed...” ?*

This is correct, “The Maastrichtian forced regression” of the WIS freed up land.

- lines 423ff: *A lot of this discussion is based on the assumption that the species-area effect is real and ubiquitous...the authors have not convinced me that this is necessarily the case.*

As we have mentioned in a previous comment, we have expanded on our use of the species area relationship to link the loss of a wide suitable area with a potential loss of richness of fossil taxa. We have also reported here (as we have done above) references relevant to the investigation of the SAR in the neontological and palaeontological realm. We think that a trial on whether the SAR is an actual and real biogeographic phenomenon is beyond the scope of this paper.

- lines 431-435: *Tectonism and eustacy are also abiotic factors, and happen to be ones that significantly influence GCMs....*

With abiotic factors we meant the effect on generating geological patterns rather than affecting diversity on a biological level (remark: we are not excluding it, just saying there is not support along this view so far and that we think the effect on geology was more important). We are suggesting that these abiotic factors affected mostly fossil preservation and so our record of palaeodiversity. We have provided plenty of details, reference, validation and application of the GCMs (and DEMs that they are built on) throughout the entire paper to understand how palaeogeography affects palaeoclimatic models used in this study.

- lines 450-451: *ENMs are not the best way to statistically compare environments because they are linked to where taxa are (and are not). Better to do a direct comparison of environments thru time using more standard multivariate statistics since you have the GCM outputs.*

We do not see the reason to “compare environments” as our investigation is focused on understanding the distribution of habitats suitable for dinosaurs, palaeodiversity and the bias that affects them. A PCA space quantifying environments through time will show different environments through time bins (particularly between Campanian,

Maastrichtian and Danian, given the stage resolution of our dataset), and we already now they are different (we have made them). In addition, by dropping fossil occurrences, and the spatial dimension associated with them, we also would lose the possibility to investigate substage trends (as our GCMs are stage level); whereas our occurrences can be constrained at substage levels, the GCMs cannot.

- lines 484-486: *What types of environments are these fossils coming from? Is it reasonable to assume that just because the PBDB record does not say “transported” that they haven’t been transported?*

The PBDB provides a detailed and accurate database of fossil occurrences, input by professional palaeontologists, as you can see from the supplementary information file S.7 (S.7. Paleobiology Database occurrences data), these information are not only related to taxonomy and spatial coordinates, but also with stratigraphical position, sedimentological environments the fossils were found in, and usually taphonomical conditions. We have already gone through and ‘cleaned’ the dataset (see Methods), but nonetheless we note that many studies have been based (and are continuously published) on these data.

- lines 489-492: *A single paper using family-level data does not seem like substantial evidence of the “wide use” of clade-level analyses. Whereas I understand the utility, and even necessity, of using clades vs. species in deep time ENMs (due to occurrence number constraints), this should not be done without caution. Some explicit discussion here regarding why dinosaur families are good proxies for species-level responses to environmental change, would be very useful.*

As the reviewer correctly remarks here (echoed by another reviewer), we are guided by necessity for these choices. Vertebrate fossil species are often represented by a few specimens, and dinosaurs are no exception. Using the PBDB on the 4 August 2018, the best samples of a “specific” taxon is *Tyrannosaurus rex*, from the late Maastrichtian of the WIB, with 65 occurrences. However, as discussed abundantly in this paper, this is spatially clustered in a few localities, resulting in no more than a couple of spatially unique occurrences if we were due to subsample them spatially for ENM investigation. Similarly, *Triceratops horridus* (the most common ceratopsian) is represented from 36 occurrences, and *Edmontosaurus annectens* (the most common hadrosaurid) is represented by 16 occurrences. With the possible unique exception of *Allosaurus fragilis*, a theropod dinosaur from the Late Jurassic of North America, with 31 occurrences, all other taxa of Dinosauria are represented by no more than tens of specimens, and in these cases, from hyper productive and very close localities from Asia or North America (source: <https://paleobiodb.org/>; see also Weishampel et al. 2004). Given the limitation of an incomplete fossil record, palaeontologists have widely used “family rank” groups to investigate palaeogeographical coverages of fossil taxa (Benton 1985; Huang et al. 2014). If that wasn’t enough to justify our use of supraspecific taxa for this investigation, we have added some more detail in the manuscript based on other studies that have used supra-generic taxonomic ranks in ecological niche modelling (Waterson et al. 2016) and macroecological investigations (Couce et al. 2012). Although niche conservatism probably acts in different ways at different taxonomic levels, this phenomenon has been shown to occur at supraspecific and particularly “family level” clades, with a certain degree of consistency particularly in species level traits like range

sizes, despite historical events like range shifts and local extinctions have occurred, particularly in closed related taxa (Hadly et al. 2009; Cooper et al. 2011). In addition, one of the reasons we have chosen these taxa for this study is not only the numerical availability of occurrences, their geographical spread in the whole North American continent, and the stratigraphic continuity of their occurrences throughout all the latest Cretaceous, but most importantly the fact that each of these clades is represented by species with comparable ecomorphological traits (i.e. large-bodied, obligate herbivorous quadrupeds, or moderate to large-sized bipedal obligate predators), showing reproductive, ecomorphological and life history similarities which provide enough within-clade functional consistency to assume close similarity of their climatic niches (Benson et al. 2018; Codron et al. 2013; Mallon et al. 2013; Snively and Russel 2007).

- line 549: *Fenton et al* is reference 32 (not 30)

Fixed.

- line 554: “biased sampleS” ?

Fixed.

- lines 591-592: *Should cite Owens et al 2013; what do your individual response curves look like? Is there significant (wacky) extrapolation in your models?*

We have added the suggested reference and we have made sure to avoid spurious extrapolation when fitting the ecological niche models. As added in the method section (Ecological Niche Modelling), we have run and described several sensitivity tests, using multiple model evaluation metrics and methods for comparison between models (like difference maps), to make sure we are using the best niche models given the data available.

- line 592: *What does “modelled outcrop ENMs” refer to? How is this different from other models in the study?*

We meant here a polygon representing the extent of outcrop area, used for clipping environmental layers and niche models to extract values falling inside the “boundaries” of the preserved geological record. Please note that we have now removed this sentence as we have added more information to the method, and changed to a clearer version whenever similar concepts were explained in the text.

- lines 602ff: *A more explicit explanation of variable choice would be helpful. There is a substantial body of literature supporting temperature and precipitation as important abiotic variables constraining modern vertebrate distributions - probably even some specific to birds and reptiles which would be important analogs to reference here and support your variable choices.*

We were forced by reference limit in our former submission to avoid digressions, and we apologise for having missed an important point. We have now added the following passage to the supplementary material (S.3) after a similar recommendation from another reviewer who asked to expand on variable choices in the SI. We hope to have

cleared any doubt on our approach on variables choice. We want to restate though that we are here in the realm of phylogenetic inferences, where this is the only tool that can help us in deciding variables that may affect the physiology of dinosaurs, given our limited knowledge on the biology of this group.

“Variable choice was dictated by the power as direct predictors of these parameters (i.e. which have a direct physiological influence on the organism). Although we are long far from estimating directly the thermophysiology or other autoecological features on extinct members of Dinosauria, coarse phylogenetic inferences can be suggested by their living most closely related organisms, sauropsids (Guisan et al. 2003), particularly crown group archosaurs (crocodiles [Ihlow et al. 2014]+birds [Peterson 2001; Peterson et al. 2002]). Temperature variables have been commonly used in phylogenetic niche conservation across clades (Peterson et al. 2011), as they represent some degree of realism with respect to how maximum and minimum temperature values vary and affect environmental conditioning across landscapes (Saupe et al. 2017). The use of these variables with a “broad” ecologically important determining effect reduces assumptions on the thermophysiology of these dinosaur clades, which is still debated (e.g. Grady et al. 2014). Furthermore, the average precipitation metric might have secondarily impacted the distribution of dinosaur communities by affecting distribution patterns, with a bottom-up effect in structuring the consumers (MacAllister et al 2004). Indirect predictors like topography were not used in this study as we are interested in macroscale patterns, and topography is usually indirectly associated with generating heterogeneity in microclimate that can provide microscale variability in the landscape for organism to live in. In addition, given the large size of the animals investigated in this study, we use the assumption that microclimatic patterns had a minor effect on their spatial distribution at continental scale.”

- line 626: *What kind of “comparisons” ?*

We have largely expanded and restructured both the method section and the results based on them. We have kept still though a comparison with backward vs forward projections to test for major sensitivity in the ENM response to each stage model, and have been using difference maps to compare the results. As added now to the manuscript, to minimise deviations from the distributions between the better fitted models, difference maps (on ESRI ArcGIS 10.2.2) were used, with models exceeding 5% of deviation discarded (a comparison also with forward-backward projections in sensitivity test was tested).

- lines 631-635: *I do not understand how/what is being standardized here; please clarify.*

We have now changed this method, recognising the lack of clarity (please see previous comments).

- *Regarding potential biases, it would be helpful to include somewhere (could be the SI) a discussion of the effects of temporal bin size on the models and their interpretation. The Campanian and Maastrichtian are substantially different in length and their sub-stages vary from ~ 2.4 Myrs to 5 Myrs in duration. This is a 2-fold difference in time bin size. Does the size of the time bin correlate to number of occurrence points? (suggesting potential bias) How about outcrop area and bin duration?*

We have now changed all the experiments in this study to substages (even the Ecospat analyses, now moved to the supplementary information, S.4). Please note that our choice of using substage level analyses is also based on the fact that each time bin in both the Campanian and the Maastrichtian as an approximately equal duration of 3.5 myr.

Refs Mentioned (by the reviewer).

*Elith, J., C. H. Graham, R. P. Anderson, M. Dudík, S. Ferrier, A. Guisan, R. J. Hijmans, F. Huettmann, J. R. Leathwick, A. Lehmann, J. Li, L. G. Lohmann, B. A. Loiselle, G. Manion, C. Moritz, M. Nakamura, Y. Nakazawa, J. M. Overton, A. T. Peterson, S. J. Phillips, K. Richardson, R. Scachetti-Pereira, R. E. Schapire, J. Soberón, S. Williams, M. S. Wisz, and N. E. Zimmermann. 2006. Novel methods improve prediction of species' distributions from occurrence data. *Ecography* 29:129–151.*

Franklin, J. 2009. Mapping species distributions: spatial inference and prediction. Cambridge University Press, Cambridge, UK.

*Liu, C., M. White, and G. Newell. 2013. Selecting thresholds for the prediction of species occurrence with presence-only data. *Journal of Biogeography* 40:778–789.*

*Nogués-Bravo, D. 2009. Predicting the past distribution of species climate niches. *Global Ecology and Biogeography* 18:521-531.*

*Owens, H. L., L. P. Campbell, L. L. Dornak, E. E. Saupe, N. Barve, J. Soberón, K. Ingenloff, A. Lira-Noriega, C. M. Hensz, C. E. Myers, and A. T. Peterson. 2013. Constraints on interpretation of ecological niche models by limited environmental ranges on calibration areas. *Ecological Modelling* 263:10–18.*

*Peterson, A. T., M. Papeş, and J. Soberón. 2008. Rethinking receiver operating characteristic analysis applications in ecological niche modeling. *Ecological Modelling* 213:63–72.*

Reviewer #2 (Remarks to the Author):

Chiarenza and colleagues tackle the oft-debated question of whether dinosaur diversity was in long-term decline prior to the Cretaceous/Paleogene boundary. Instead of focusing on diversity itself, they used ecological niche modelling to examine the extent to which dinosaur habitat changed in concert with observed diversity changes. Chiarenza and team also examined sediment flux and surface runoff in both the Campanian and Maastrichtian of North America to quantify potential taphonomic biases affecting understanding of dinosaur diversity over this interval. The study will be of broad relevance to readers given deep interest in the evolutionary dynamics of this clade. Moreover, Chiarenza and colleagues use a novel approach to try to understand potential taphonomic biases influencing our interpretation of diversification patterns through time. I note a number of concerns and suggestion below.

We thank the reviewer for reviewing our manuscript and are glad of their overall interest and suggestions. We detail a point by point response below.

Overall comments

1. Perhaps my most relevant comment is that ecological niche modelling (ENM), or species distribution modelling (SDM), can provide insight into whether the distributional potential of a group was declining or expanding in terms of climatic niche footprint, but this decline or expansion does not necessarily translate into diversity decrease or increase, respectively. More specifically, ENM can tell you that the climatic conditions for a clade were broadly suitable, and therefore potential declines in diversity were not likely due to environmental degradation. However, and importantly, diversity can decline for other reasons (for example, from reduced rates of speciation or extinction from biotically-mediated forces). The ENM analyses may be best shown in concert with sample-standardized diversity curves (which have been done previously) and sediment flux and surface runoff models, which are almost more convincing as to why we may have reduced ability to sample dinosaur diversity in the Maastrichtian. The focus is, at the moment, strongly on the ENM analyses, but perhaps it should be more equally-weighted among all three analyses (sample-standardization, taphonomic modelling, and ENM modelling), all three together of which show why we cannot say that diversity declined prior to the K/Pg.

2. SDMs/ENMs are modelled at the clade level for the Ceratopsids, Hadrosaurids, and Tyrannosaurids to provide a general overview of the degree to which suitable habitat changed over time for these clades. My question is whether similar patterns would be obtained if models were calibrated at the species level and summed to the clade level. Would increases in suitable habitat for each group still be obtained? Or, would a more nuanced picture be obtained as some species' habitat decreased and other species' habitat increased?

3. Was any statistical analysis done to show the association of fossil occurrences with lower sediment flux?

4. The analyses on niche occupation and niche stability are interesting, but they seem out of place and do not contribute to the argument at hand (although apologies if I am missing something critical here). These analyses should perhaps be better integrated and/or clarified, or simply removed.

We are aware that ENM can provide insight into the climatic niche footprint and should not be used to unequivocally define diversity trends (as also highlighted by

another reviewer). We have now toned down this aspect of the manuscript to be more focused towards what habitat suitability for these dinosaur clades may tell us in terms of spatial bias and potential effects on our interpretation of paleobiodiversity. We have included this information in replies to another reviewer, but restated it for the sake of clarity, particularly when a new suggestion was made by this reviewer. Despite this, we still contend that our results indicate that there is no evidence for a decline in diversity in the lead-up to the K/Pg mass extinction.

We had not included subsampled diversity curves for several reasons: as we have discussed extensively (and referenced), past efforts to reconstruct diversity curve with several subsampling methods (e.g. Residuals and SQS) have been made, and the first two authors of this contribution have co-authored several papers using them in the past (e.g. Mannion et al. 2011 for residuals; Tennant, Chiarenza and Baron, 2018 for SQS). On the other hand, there are a few technical issues to consider. The residual diversity method uses residuals from a modelled relationship between palaeodiversity and sampling (sampling-driven diversity model) as “corrected” diversity estimates. The method has recently been abandoned as it shows spurious inherent statistical fallacies like “unacceptably high rates of incorrect and systematically, directionally biased estimates”. A thorough investigation on the method, using simulations to explore how robust the method is, has shown high error rates and biases in regression model coefficients (Sakamoto et al 2017). We have reported the aforementioned study in the reference after this comment. On the other hand, to satisfy the reviewer curiosity on the effect of rock outcrop area, we have computed a residual diversity analysis on the three clades investigated in this study and included both the raw analyses (as an excel spreadsheet; For reviewers. Residual testing) and a figure (For reviewers_Residuals figure). The resulting residuals show “flattened” diversity trends for Ceratopsidae and Hadrosauridae, and a more emphasised “dip and through” curve for Tyrannosauridae. In order to acknowledge the efforts being made by other authors in the past though disproving the efficiency of this method, we would like to ask the reviewer to keep these data and analyses for their own consideration, as we would like not to include these data in our manuscript.

As for the shareholder quorum subsampling (SQS; Alroy 2010), the method tries to standardize unequally sized samples allowing for comparison between time bins characterised by unequal sampling intensity, adapting the “standardisation quorum” to the level of the best complete sample. Although the method has been used extensively to analyse dinosaur (and generally fossil vertebrate) diversity in the past, we unfortunately cannot use it for our scope for several reasons. Most importantly, SQS needs long time series to work effectively, as it needs to compare several quora to properly standardise for sampling intensity. As our time series is no longer than 5 time bins, with particularly 2 very low sampled ones, we cannot apply it here without violating statistical integrity. Furthermore, as the method is very sensitive to sample size, oftentimes it is binned at either “stage level”, or with time bins of at least 5 million years (again, an approach which would shorten our already short time series). Furthermore, one of the contributions that our methods tries to investigate is the effect of null sample size on estimating diversity: SQS for example cannot deal with absence of raw diversity signal in a bin, often underestimating or altering the diversity signal between sampled and unsampled time bins. This issue is particularly important when data do not cover some specific areas (if you have an unsampled area, like in our study system many areas of Appalachia), the method cannot simply work with it. So our kind of investigation tries to tackle more intimately the issue of spatial bias in paleodiversity estimates. We have

reported at the end of the comment a series of references on SQS and other sample standardisation method on the issue, as many of them had to be kept out of the main manuscript because of journal instructions. Finally, previous usages of these methods suggest that there is little evidence for a substantial decline (beyond background rates) in dinosaur diversity: our approach is an attempt to independently address this issue.

We agree, as mentioned in previous comments, that the initial approach emphasised perhaps too much the ENM component of the study, and we have restructured the paper to highlight more the geological and taphonomical component. Following a comment below by the reviewer, we have included now in the paper (Figure 5) a figure showing raw diversity trends for the three clade of dinosaurs in this study plotted against outcrop exposure.

An additional note to be considered by the editor and reviewers is that we are aware that climatic niche alone and geological modelling cannot conclusively establish that pre-extinction diversity wasn't sustained and not decreasing. On the other hand it does generate predictions and highlights uncertainty in our evaluation of palaeodiversity and sampling that provide "indicative truths" to suggest so.

2. As another reviewer has raised the same scepticism over supra-specific clade analysis, please see comment above.

3. We added, as reported in a previous comment, results from Pearson's square correlation (Table 1 and supplementary file S.2.), plus adding relevant discussion and description of the method in the section.

4. We acknowledge as discussed in other comments that in the original submission, ecospat analyses were not properly integrated with the manuscript. As we have commented before (given that another review made a similar suggestion) we have majorly reworked our ecospat section and integrated it to the manuscript in a different way.

First of all we have changed the analyses to substage level, late Campanian versus late Maastrichtian, time bins approximately of equal duration (~3.5 myr), following the suggestion of temporal consistency of the reviewers. In addition, we compared the two most suitable and highly diverse substages, the late Campanian and the late Maastrichtian. We have reported *D*-metric and *p*-values both in the main text and the figures. We think that these analyses support now a strong similarity environmentally between the two substages, supporting also our view that habitat degradation during the last ten million years of the Cretaceous surely wasn't responsible for a diversity drop induced by changing climate and that the pre-extinction environments could have potentially hosted a high richness of dinosaur taxa.

In addition, as we noticed that, although we do not deny the usefulness of this kind of analysis in this study, as outlined by 2 reviewers in the last round of review, a single ecospat plot showing the combined environmental cloud for each of the three taxon might not be the clearest figure to follow. In order to fit figures limit without compromising the clarity of the plots, we have decided to move entirely all the plots in the supplementary material (S.4), adding some additional description to the analyses in the main version of the manuscript. You can find the full set of niche ordination analyses in the supplementary information file S.3.

Other comments

1. Fig 2: some specimens are found in the middle of the WIS. Why this occurs should be briefly mentioned.

The palaeogeographies used in the figures represent a snapshot of an average of the highstand of the sea level. In addition, to make more visible the occurrence points, we have kept all of the occurrences present in the “un-pruned dataset”, showing many of the fossil occurrences that, when putting together the data for ENM, have been cut off from the dataset as they represent displaced fossils in coastal settings. As regression-transgression rhythms are continuous and alternate between more proximal-distal levels, the coastlines and the eventual points of accumulation of fossils in their basins constantly change and can be seldom pushed deeper in what is the centre of the epicontinental area at a given time. We have now included two figures with Kernel density (Fig. 2a, b) showing with a dashed black line the average sea level lowstand.

2. More introduction is needed on why a cluster analysis was performed (even if just briefly) in both the main text results and methods and in the supplementary materials. With regard to the latter, the results could be better presented (for example in a Table?).

We apologise for the lack of clarity on the cluster analyses. As another reviewer raised a similar point, we have changed the cluster analyses used in the original submission with hotspot analyses – see above

3. It might be useful to include discussion of the number of dinosaur taxa currently found from Laramidia in each of the three clades. A similar table could be produced as to Table 2, but focused on different regions.

We thank the reviewer for the suggestion, but as our emphasis right now is more macroecological than taxonomical, we would like to avoid to rearrange the manuscript with the risk of creating some unbalancing between ecological and geological modelling and this taxonomical discussion. Plus, table limit in the manuscript has already been reached by Table 1, with the spatially unique occurrences per grid cell, and Table 2, with the chi-squared support of association for virtual taphofacies. Also, a detailed and minimally inclusive taxonomy is available exclusively for Laramidian taxa, as the almost totality of the very few Appalachian occurrences (the Eastern Cretaceous North American subcontinent) is represented by oftentimes non diagnosable material, representative *incerta sedis* ID of supraspecific clades (Brownstein 2018). Last, but not least, the main author of the manuscript is currently co-authoring a more detailed and extended taxonomical revision of the latest Cretaceous North American dinosaur diversity on another paper in review elsewhere.

4. It might be easier for readers to comprehend the different modelling projections/analyses with a chart or table. For example, the authors calibrated models on outcrop area only, and then performed numerous projections in space and time. The series of analyses could potentially be more easily understand with a schematic?

We agree that the original concept of this paper was quite complicated, and as answered in previous comments, we have decided to restructure it in a simpler way (as we have now reported in the manuscript in a hopefully clearer and more linear way). Although we think that the idea of a schematic might be good, we have not included it as we already reached the figure limit and believe that the existing figures are more crucial to the reader.

5. Line 92: The authors indicated palaeoenvironmental data had been interpolated to a finer resolution: what resolution? Also, this interpolation should be reiterated in the palaeoclimate model section.

We have added this information. As another reviewer requested a similar clarification, we have now added more description and detail to our GCM section, providing also additional information to the ENM section where needed – see above.

6. Line 198: How do you quantify suitability here? The methods section indicates cells with values > 0.5 were chosen, but no rationale was provided. Choice of threshold may dramatically alter niche model results (see, e.g., Lui et al. 2005 or 2013), and ideally multiple thresholds should be compared and results assessed for congruence.

As we have replied above to another very similar reviewer's comment, please we agree and apologise with the reviewers for the confusion we caused with this method for calculating relative suitability in each time bin. Please see above comment.

7. Line 199: How was relative area standardized?

See comment above.

8. Line 236: Perhaps include a brief explanation / introduction of niche gradients with ordination techniques.

We have removed this figure (and relevant caption) from the main manuscript. As two reviewers have raised concerns toward the clarity of the ecospat analysis, we have added more detail to both the method section, how we presented the results, and to the relative figures plus additional description now all in a separate supplementary information file (S.3).

9. Line 259: Were occurrences used for the niche overlap analyses, or were the model results? Please clarify.

Occurrences, as we have now specified in the relevant section.

10. Line 295: Was rock outcrop area in the time slices quantified? This would be interesting and relevant to present.

This is a great idea and we thank the reviewer for it. We have added this plot in Figure 5, plus a relevant description in the method section on how we built it.

11. Line 445: A parenthesis is missing.

Thanks for spotting this, we fixed it now.

12. Lines 451-8: The wording and reasoning here is a bit unclear.

We apologise for that; we have rewritten that passage for the sake of clarity.

13. Line 489: Are the number of occurrences cited spatially unique? It seems unlikely from the resolution of the data. How many spatially-unique occurrences (at the resolution of the climate data) were used for each clade in each time slice? Perhaps these data should be listed in a table.

We agree with the reviewer and we have added the suggested Table as Table 2 in the main manuscript.

14. Lines 554-559: Other authors have found clustered fossil occurrences through time (see, for example, Plotnick, 2017, Geology).

Thanks for the relevant suggestion. We have added Plotnick 2017 as a reference in the relevant expanded discussion on hotspot analysis in S.1.

15. Table 2 – Could diversity counts be shown after standardising for sampling (e.g., using SQS)? This may elucidate patterns and further augment your position (see overall comment #1).

See detailed reply to similar/related comment above.

16. Fig S1 and S2 – It is difficult to see the occurrences in these figures. Could they be made bigger?

As we have changed the cluster analyses to hotspot analyses (see comment above), we have removed these figures, and changed them with a symbology which does not show occurrence point but probability areas (supplementary material S.1).

17. Fig S3 – A solid red contour line is referred to in the ECOSPAT analyses, but it does not seem to be present on the figures.

Please note that we have now largely updated the whole ecospat section, putting all the outputs in a Supplementary File (S.4). The relevant figures to this comment are S.20, S.22, S.24 where in “c”, a solid contour line illustrates the full range (100%) of climate space in the two time intervals considered and dashed lines the 50%.

18. Fig S4+ – These figures should be explained in more detail here, in the Methodology, and in the Results. From my reading, it seems there is no support for niche equivalency, which makes sense, but that there is mild evidence for niche stability through time (at the clade level). While these analyses are certainly interesting, it is unclear how they add to the manuscript in its current form (see overall comment #4).

Yes, we agreed and we have changed/updated the ecospat analyses and moved it out from the main manuscript (see detailed answer from overall comment #4)

19. *Does the variable contribution to each model make sense based on inferences of the biology/ecology of these clades? Perhaps this should be discussed. A large table of these results is presented, but they are not discussed or made much of with regard to the main arguments (and indeed, they do not seem particularly relevant to the main points of the paper). This entire discussion and the associated results could likely go in the supplementary materials? Unless, of course, the idea is to comment on what conditions are important for each clade, but then some discussion is needed with regard to whether the particular variables make sense based on what is known of the clades in question.*

This is a tricky issue to tackle. As we have detailed in a previous comment, we considered environmental predictors according to what we know of the abiotic requirements using a phylogenetic inference approach (using what is known on crown groups archosaurs; see comment above on using family level occurrences for this study). On the other hand we do not think there is enough knowledge on the biology of these extinct taxa to say anything in detail on whether variable responses make sense. We know that avian dinosaurs (i.e. birds) are today constrained by thermal excursions (hence migration), so a broadly similar response is expected (as we get in this study), but again, with which particular degree, given how uncertain is dinosaurs' thermophysiology (somewhere on the spectrum between heterothermy and endothermy?). We realise that this may cause some confusion to neontologists (it does to us as palaeontologists!), but the best (and most honest approach) we could afford was to highlight (as we have added now) some discussion on variable choices according to phylogenetic inferences, niche conservatism assumptions, and ecomorphological consistency. We are aware that there are a lot of assumptions, and we have expanded on this issue following the recommendation of the reviewer, putting this discussion in the supplementary file S.3.

20. *The authors modelled suitable habitat into the Danian and showed an increase or at least stability in amount of available habitat. However, the conditions immediately after the K-Pg impact would have been very unsuitable, and this unsuitability would have persisted for some time, leading to the demise of the non-avian dinosaurs. Perhaps this should be made clearer?*

Yes, we agree, and we are currently writing a paper on short term drivers of K/Pg mass extinctions, but we concur that it is a confusing phrasing right now and we have added further details, restructuring the passage to state what we meant in clearer way, hoping to make it more comprehensible to the reader.

21. *The authors found an increase in suitability for Hadrosaurids into the Maastrichtian when considering just outcrop area, but hadrosaurids show a decline in diversity similar to the other clades using the raw taxon count data. Given that the authors claim there is a link between suitability and diversity, shouldn't a decline in hadrosaurid diversity be surprising here? Perhaps more discussion of this is warranted?*

As we have now changed the original experimental setting of the ENM and double checked all the results, the evidence has now changed and we have incorporated it in the manuscript accordingly. In particular, Hadrosauridae reach their peak in habitability the middle-late Campanian, then dropping at lower level in the Maastrichtian, especially for the highest suitability threshold (<0.7; see Fig. 4a), a result that can be seen mirrored in Fig. 5.

22. *The authors project their models in both space and time, with extrapolation enabled. This means the models will extrapolate suitability when encountering combinations of climatic conditions not observed in the model calibration region. When extrapolation is enabled, the Maxent algorithm will produce maps of these regions in space. It might be useful for the authors to present these maps. Moreover, what happens with regard to amount of suitable habitat if extrapolation is NOT allowed?*

We agree with the reviewer and tested sensitivities for changing these settings. We have now specified that although we used the results using extrapolation (to allow more reliable response curves) in the discussion, we have also tested for changes due to replications using Clamping (to see how truncation of responses curves was affecting the distribution of habitat suitability spaces). We have also iterated different run types (crossvalidate, bootstrap and subsample) to test whether marginal deviation from a single response model was biasing our interpretations. The models that provided the best AUC values (>0.9), and minimised extrapolation in the curves (showing biologically more sensible patterns), were retained to fit the ENMs. As an additional sensitivity test, we used comparisons between forward and backward projections (from Campanian substages to Maastrichtian and vice versa) as an independent test to assess discrepancies in the spatial distribution models. To further investigate deviations from the distributions between the better fitted models, we used diffmaps (on ESRI ArcGIS 10.2.2), discarding models exceeding 5% of deviation from the majority of better fitted models (the discarded ones were only forward-backward projections from Early Campanian and Early Maastrichtian, probably because, as mentioned in reference to Table 2, for the low number of unique spatial occurrences). As we only presented projection from late Maastrichtian to the Danian (hence showing non-analogue region from the training region to the projection stage), we have marked with a dashed line in Fig. 3b this area, set in the middle of the Danian North America.

Reviewer #3 (Remarks to the Author):

Review of Chiarenza et al. "Ecological niche 1 modelling indicates no decline in dinosaur diversity prior to the Cretaceous/Paleogene mass extinction"

This is an excellent, innovative study addressing a longstanding debate in dinosaur paleobiology: to what extent were non-avian dinosaurs 'on the decline' prior to the end-Cretaceous mass extinction? This question has been flogged for many years with conflicting results, and in many ways has grown stale since the underlying data for all such studies are subject to the same sampling biases.

Chiarenza and colleagues attempt to circumvent these limitations by incorporating an ecological niche modelling approach, allowing them to investigate how apparent declines in raw dinosaur diversity in the latest Cretaceous of North America compare with changes in suitable dinosaur habitat, and changes in the abundance of suitable dinosaur-bearing rock through this interval.

The results they present are convincing, and move the study of Late Cretaceous dinosaur diversity trends substantially forwards. Moreover, I think the study will serve as an exemplar of the power of integrating cutting-edge niche modeling approaches with the study of paleodiversity.

The authors illustrate convincingly that the apparent decline in suitable dinosaur habitat from the Campanian to the Maastrichtian is likely to be an artifact of limited rock exposure, and that, when continental-scale trends are considered, habitat suitability for the major clades of non-avian dinosaurs may have increased throughout the latest Cretaceous of North America.

Importantly, the authors argue that the same low-stand eustatic conditions that generated an increase in habitat suitability towards the Maastrichtian yielded poorer overall conditions for fossil preservation. This trade-off provides a satisfying explanation for a significant amount of the longstanding disagreement among palaeontologists related to whether or not non-avian dinosaurs were on a long-term decline throughout the latest Cretaceous.

A secondary, but additionally interesting insight from the authors' work shows that ENM estimates of habitat suitability in the earliest Palaeocene show no declines for the three major lineages of non-avian dinosaurs under investigation, providing additional support for a geologically sudden and catastrophic event as the principal driver of non-avian dinosaur extinction.

One caveat: although I am familiar with ENM approaches and paleobiology database-style analyses, I am not an expert in either of these techniques, and feel that the manuscript will need to be appraised on a more technical level by ENM and paleoDB specialists.

The study is generally well-written, although it could benefit from more careful proof-reading for style and technical errors at times. For example, the first line of the Introduction is somewhat confusingly worded, and should be rewritten in my opinion.

Last, the trans-K–Pg results presented in this study are interesting, and I agree with the authors that these results bear on the explosive post-extinction radiation of crown birds and mammals. In addition to the classic references listed, I think referencing more recent and convincing evidence for rapid early Cainozoic radiations of mammals and birds would be suitable here, such as Prum et al. 2015 for birds and O'Leary et al. 2013 for mammals.

I have flagged a small number of minor grammatical issues below, but I did not pay particularly close attention to these details. The authors should give the writing a close look to expunge similar errors that I'm sure I have missed.

We are pleased that the reviewer thinks that our paper is worthwhile and sufficiently novel for publication in Nature Communications. We have focussed here on the few suggestions that have definitely improved the previous submission.

Reviewer 3 correctly pointed out that this first sentence of the manuscript was confusingly worded, so we have rephrased and simplified to a more linear version: “Reconstruction of the palaeodiversity of Mesozoic dinosaurs has a long tradition in palaeontology, with a growing number of studies over the last 40 years^{1,2,3,4,5}”. Furthermore, as suggested by the reviewer, we have proof read the manuscript to minimise grammatical errors and typos, and we hope to have avoided missing any. We have updated the final section of the paper with more recent references on the post-K/Pg mass extinction (Prum et al. 2015 for birds and O’Leary et al. 2013 for mammals).

Line 19: remove comma after window

Fixed.

Line 51: delete ‘for’ after mitigate

Fixed.

Line 445: close parenthesis.

Fixed.

References:

Allouche, O., Tsoar, A. and Kamon, R. (2006). Assessing the accuracy of species distribution models: prevalence, kappa and the true skill statistic (TSS). *Journal of Applied Ecology*, 43: 1223-1232. doi:10.1111/j.1365-2664.2006.01214.x

Alroy, J. (2010), Geographical, environmental and intrinsic biotic controls on Phanerozoic marine diversification. *Palaeontology*, 53: 1211-1235. doi:10.1111/j.1475-4983.2010.01011.x

Benton MJ. 1985 Mass extinction among non-marine tetrapods. *Nature* 316, 811–814.

Benson, R. B., Hunt, G., Carrano, M. T., Campione, N. and Mannion, P. (2018), Cope's rule and the adaptive landscape of dinosaur body size evolution. *Palaeontology*, 61: 13-48. doi:10.1111/pala.12329

Bradshaw, C.D., Lunt, D.J., Flecker, R., Salzmann, U., Pound, M.J., Haywood, A.M., Eronen, J.T (2012). The relative roles of CO₂ and palaeogeography in determining late Miocene climate: results from a terrestrial model-data comparison. *Clim. Past*, 8, 1257-1285. [doi:10.5194/cp-8-1257-2012]. [And see corrected Figures 7-16 in Corrigendum]. [And see a further Corrigendum.]

Brownstein, C. D. The biogeography and ecology of the Cretaceous non-avian dinosaurs of Appalachia. *Palaeo. Elect.* 21, 1. (2018).

Chainey, S., Ratcliffe, J., 2005. GIS and Crime Mapping. John Wiley and Sons, UK.

Close, R. A., Benson, R. B. J., Upchurch, P. & Butler, R. J. Controlling for the species-area effect supports constrained long-term Mesozoic terrestrial vertebrate diversification. *Nat. Commun.* 8, 1–11 (2017).

Codron D, Carbone C, Clauss M (2013) Ecological Interactions in Dinosaur Communities: Influences of Small Offspring and Complex Ontogenetic Life

Cohen, K.M., Finney, S.C., Gibbard, P.L. & Fan, J.-X. (2013; updated August 2018) The ICS International Chronostratigraphic Chart. Episodes 36: 199-204.

Couce E, Ridgwell A, Hendy EJ. 2012 Environmental controls on the global distribution of shallow-water coral reefs. J. Biogeogr. 39, 1508–1523. (doi:10.1111/j.1365-2699.2012.02706.x)

Codron D, Carbone C, Clauss M (2013) Ecological Interactions in Dinosaur Communities: Influences of Small Offspring and Complex Ontogenetic Life

Foster, G. L., Royer, D. L. & Lunt, D. J. Future climate forcing potentially without precedent in the last 420 million years. Nat Commun 8, doi:ARTN 14845

Fotheringham, S., Brunson, C., Charlton, M., 2000. Quantitative Geography: Perspectives on Spatial Data Analysis. Sage, Thousand Oaks, CA.

Gough, D. O. Solar Interior Structure and Luminosity Variations. Sol Phys 74, 21-34, doi:Doi 10.1007/Bf00151270 (1981).

Hadly E. A., Spaeth P. A., Li C. Niche conservatism above the species level. Proceedings of the National Academy of Sciences Nov 2009, 106 (Supplement 2) 19707-19714; DOI: 10.1073/pnas.0901648106

Huang S, Roy K, Jablonski D. 2014 Do past climate states influence diversity dynamics and the present day latitudinal diversity gradient. Glob. Ecol. Biogeogr. 23, 530–540.

Kallimanis, A. S., Mazaris, A. D., Tzanopoulos, J. , Halley, J. M., Pantis, J. D. and Sgardelis, S. P. (2008), How does habitat diversity affect the species–area relationship?. Global Ecology and Biogeography, 17: 532-538. doi:10.1111/j.1466-8238.2008.00393.x

Liu C, Berry P, Dawson T, Pearson R. Selecting thresholds of occurrence in the prediction of species distributions. Ecography. 2005; 28: 385–393.

Liu et al. (2013) Liu J, Möller M, Provan J, Gao LM, Poudel RC, Li DZ. Geological and ecological factors drive cryptic speciation of yews in a biodiversity hotspot. New Phytologist. 2013;199(4):1093–1108. doi: 10.1111/nph.12336.

Lunt, D. J., Dunkley Jones, T., Heinemann, M., Huber, M., LeGrande, A., Winguth, A., Loptson, C., Marotzke, J., Roberts, C. D., Tindall, J., Valdes, P., and Winguth, C. (2012). A model-data comparison for a multi-model ensemble of early Eocene atmosphere-ocean simulations: EoMIP, Clim. Past, 8, 1717-1736. [doi:10.5194/cp-8-1717-2012].).

Lyson, T. R. & Longrich, N. R. Spatial niche partitioning in dinosaurs from the latest cretaceous (Maastrichtian) of North America. Proc. R. Soc. B Biol. Sci. 278, 1158–1164 (2011).

Mallon JC, Anderson JS (2013) Skull Ecomorphology of Megaherbivorous Dinosaurs from the Dinosaur Park Formation (Upper Campanian) of Alberta, Canada. PLoS ONE 8(7): e67182.

Mannion PD, Upchurch P, Carrano MT, Barrett PM et al., 2011, Testing the effect of the rock record on diversity: a multidisciplinary approach to elucidating the generic richness of sauropodomorph dinosaurs through time, *Biological Reviews*, Vol: 86, Pages: 157-181

Myers, C. E., Stigall, A. L. & Lieberman, B. S. PaleoENM: Applying ecological niche modeling to the fossil record. *Paleobiology* 41, 226–244 (2015).

Natalie Cooper, Rob P. Freckleton, Walter Jetz. Phylogenetic conservatism of environmental niches in mammals. *Proc. R. Soc. B* 2011 -; DOI: 10.1098/rspb.2010.2207. Published 5 January 2011

Nogués-Bravo et al. Climate Change, Humans, and the Extinction of the Woolly Mammoth. *PLoS Biol* 6(4): e79 (2008).

Roy E. Plotnick; Recurrent hierarchical patterns and the fractal distribution of fossil localities. *Geology* ; 45 (4): 295–298. doi: <https://doi.org/10.1130/G38828.1>

Sakamoto, M., Venditti, C. and Benton, M. J. (2017) Residual diversity estimates' do not correct for sampling bias in palaeodiversity data. *Methods in Ecology and Evolution*, 8 (4). pp. 453-459. ISSN 2041-210X doi: <https://doi.org/10.1111/2041-210X>

Sabel, C., 2006. Kernel Density Estimation as a Spatial-Temporal Data Mining Tool: Exploring Road Traffic Accident Trends, GISRUUK 2006, University of Nottingham.

Silverman, B. W. *Density Estimation for Statistics and Data Analysis*. New York: Chapman and Hall, 1986.

Shufeng Li, Yaowu Xing, Paul J Valdes, Yongjiang Huang, Tao Su, Alex Farnsworth, Daniel J. Lunt, He Tang, Alan Kennedy, Zhekun Zhou: Oligocene climate signals and forcings in Eurasia revealed by plant macrofossil and modelling results, in press, *Gondwana Research*.)

Snively, E. and Russell, A. P. (2007), Functional morphology of neck musculature in the Tyrannosauridae (Dinosauria, Theropoda) as determined via a hierarchical inferential approach. *Zoological Journal of the Linnean Society*, 151: 759-808.

Valdes, P. J. et al. The BRIDGE HadCM3 family of climate models: HadCM3@Bristol v1.0, *Geosci. Model Dev.*, 10, 3715-3743 (2017).

Waterson A. M. et al. Modelling the climatic niche of turtles: a deep-time perspective. *Proc. R. Soc. B* 283: 20161408 (2016).

Weishampel, D.B., Barrett, P.M., Coria, R.A., Loeuff, J.L., Xing, X., Xijin, Z., Sahni, A., Gomani, E.M.P., and Noto, C.R. 2004. Dinosaur Distribution, p. 517-617. In Weishampel, D.B., Dodson, P., and Osmólska, H. (eds.), *The Dinosauria*, 2nd Edition. University of California Press, Berkeley, California, USA.

Information on resubmission

Number of words in the abstract of the submitted manuscript: 149

Number of words in the main text of the submitted manuscript: 5009

Number of words in the Methods of the submitted manuscript: 3267

Number of figures: 6 (included in MS file)

Number of tables: 2 (included in MS file)

Additional materials:, a manuscript checklist, an original 'track changes' version of the manuscript in word, a track changes file with highlighted changes, 8 Supplementary Information files, comprising of 3 tables, 2 scripts in text format and 25 figures (included in SI documents)

Additional number of data for reviewers' discretion only: 1 figure and 1 excel file with additional tables and datasets.

Total number of submitted files: 21

Expected number of published pages in Nature Communications: 12

We thank the referees for their kind and constructive comments they have provided to strengthen this manuscript. It did not only improve the quality of this study, but provided also a great learning opportunity on these fascinating subjects and interesting methodologies.

Yours sincerely,

Alfio Alessandro Chiarenza,

On behalf of all the co-authors.

Reviewers' Comments:

Reviewer #1:

Remarks to the Author:

General Thoughts:

This revision modifies substantial aspects of the original manuscript in a positive way to alleviate confusion, increase transparency, and fix many of the methodological concerns from my first review. The hot spot sedimentological analysis in particular is much clearer and more convincing; a strong result from this work is that the limited and spatially clustered nature of available outcrop and dinosaur sampling impairs the ability to accurately estimate patterns of diversity. From this perspective, previous work citing a decrease in dinosaur diversity leading up to the K/Pg extinction is likely underestimated and even completely artefactual. However, as opposed to interpreting this issue as intractable to diversity analysis, the authors attempt to estimate diversity using ENM predicated suitable habitat area based on the justification of the species-area relationship (i.e., larger area → larger diversity). Whereas I am very comfortable with the conclusions surrounding the rock record showing an inaccurate decrease in dinosaur diversity at this time, I am still very uncomfortable with the application of ENM to infer a diversity increase at this time. Further, the ENM analyses still include some methodological concerns and the Ecospat results are not well-integrated into the overall goal of the study, making these results difficult to interpret. More detailed comments are provided below.

1) I remain unconvinced on the point that the species-area relationship justifies the interpretation of ENM predicted habitat as informative for diversity measurements:

- a. The ENMs in this analysis are trained at the clade (family) level. A clade-level model could result from several different scenarios of species-level patterns: the model could be controlled by a single, widespread taxon, it could reflect a set of species with extremely similar abiotic niche requirements, or it could reflect a summation of species each with much smaller niche dimensions. Which of these scenarios is predominant makes a difference in how these models may reflect diversity. In the first and second, a large predicted suitable habitat area says nothing about species-level diversity. Without some sense of the generality of how clade-level niche models reflect species-level patterns (e.g., with extant clades where both the clade and species' can be tested), these results remain very hard to interpret.
- b. In the text, the species-area relationship is used in conjunction with "larger habitat area = greater number of habitats" (lines 304-305). This would seem to be quite variable with latitude – i.e., an equivalent area in a temperate region has a more depauperate set of abiotic habitats than the same area in a tropical region.
- c. As I mentioned before, the use of ENM to estimate changes in biodiversity has not been applied before. It seems very imprudent to apply this inferred relationship to a deep time system with no possibility for independent validation before testing this concept in a modern setting.

2) ENM methods:

- a. In the revised version of this manuscript, models are trained to the entire extent of North America. As I mentioned before, this introduces a significant potential for biasing the model against any areas that have not been sampled (because the modeling algorithm treats unsampled areas as poorly suitable). This is evident by the significantly decreased predicated suitable habitat in this version of the models compared to the original manuscript. The authors had less biased models in their original version when they trained the models on sampled areas (i.e., those with outcrop) and projected these to the continental scale; I'm not sure why the change was instituted...?
- b. I am confused as to why the authors projected their full continental model onto outcrop. The result should be identical to clipping the outcrop layer from the full continental model. The fact that the outcrop suitability maps (Fig 3a) are somewhat different than the full continental model (Fig 3b)

concerns me – why are these results different?

c. It is important to be clear throughout the paper that the ENM results are providing “predicted” suitable habitat area. At best, models can hope to estimate the “truth” and in this case in particular, where there is no outcrop to ground truth model predictions, I think it’s important to be explicit about uncertainty.

d. It is confusing throughout the results and discussion to refer to the full continental model as “projected”. From my reading of the methods, the model was trained at a continental extent, so the North America maps reflect the training region, not a projected region?

3) Ecospat tests: Whereas I applaud the application of tests of niche stability, especially in deep time where these analyses can be most informative, these tests are not well-integrated into the overall purpose of the study. The purpose seems to be to test for the “reality” of an observed diversity decline leading into the K/Pg mass extinction, particularly from the perspective of sampling biases. How does niche stability play into this goal?

It is also difficult for me to know how to interpret niche stability within a clade for the reasons described in #1a above. It seems reasonable to assume that the higher the taxonomic level, the more “stable” the composite niche dimensions will be regardless of number species...how is this information useful for the current study?

Specific Comments

- Line 98: I’m not sure why the reference to endangered species is relevant? ENMs are useful for predicting species responses to environmental change in all species...? (assuming niche stability)

- Lines 108-110: ENMs are most certainly not exempt from spatial or sampling biases. As a spatial analysis, they are strongly subject to these issues!

- Figure 3a: would be helpful to provide an outline of the continent so that we can see where these areas occur in space. As it is, they sort of look like they concur with the areas in Fig3b, but the model results are not identical (as I would expect) and this makes me wonder if I’m comparing the same areas between these figures.

- S3, figure caption for Fig S20: I have already commented on the misuse of “niche evolution” in running Ecospat or other ENM analyses through time. You simply cannot tell if niche expansion is adaptive or simply a species occupying suitable habitat that was previously inaccessible (e.g., because of constraints imposed by dispersal or biotic interactions).

Reviewer #2:

Remarks to the Author:

The authors have substantially improved their contribution, and I am impressed with the thoroughness of their Response to Reviewer comments. As I said before, the contribution provides an interesting perspective and commentary on potential dinosaur dynamics during the Late Cretaceous. However, I still think the manuscript would benefit from enhanced clarity in methodology and an overhaul of its focus, which I detail below:

1. I do not think you can use ENM to assess how diversity has changed through time, and I disagree with the reasoning and stated aims on lines 106-114 and 130-133. The authors rightly note (line 104) that ENMs do not consider factors such as dispersal ability and biotic interactions (if built only using climatic data and species occurrences). Therefore, one could generate a model that indicates suitable

habitat in an area that a species (or clade) is not found because of barriers. For example, say a terrestrial species with poor dispersal potential originated in North America. Suitable habitat for this species may be present in Asia, but it has very little chance of dispersing there and occupying that region. Thus, you cannot model diversity or say anything about diversity using ENMs, but you can say whether or not habitat suitability for dinosaurs was declining through time. Essentially, IF dinosaur diversity was declining, your analyses can show that this decline was not because of reduction in the dinosaur species' abiotic habitat. The fact that you found that suitable habitat did not decrease is congruent with previous standardized diversity curves, and therefore provides a nice complement to previous analyses using different (and novel) methodology. Again, you cannot assess the distribution of diversity, but rather you can provide a means of assessing whether a decline in suitable habitat could have caused a decline in number of dinosaurs leading up to K/Pg. I think the focus/purpose of the manuscript needs to change, as does the language with regard to diversity/ENM. Your concluding paragraph in the Discussion was excellent (lines 398-409) and was the best summary of what you were trying to accomplish. It also presented the data and information in the correct order: I would try to re-structure your manuscript following this paragraph, which would constitute an overhaul of essentially the entire framework.

2. Line 15 – consider changing 'define' to 'quantify'?

3. Abstract in general is much improved, but see point 1!

4. The authors mention in the Introduction that dinosaur bearing localities are fairly continuous and of best quality in the western half of the US (in Laramidia). But, what IS the diversity pattern if you examine this record? It should be mentioned in the Introduction. And, if it is so well sampled, then why would it not be reflective of the true diversity pattern? The authors need to make an argument for why (or why not) this fairly continuous record is reflective of true dinosaur diversity patterns, which has direct bearing on why they need to perform their current analyses (it is touched on some in the Discussion, but additional explanation needs to be up front and center when these records are first introduced).

5. Line 99: ENMs are not just used to model endangered species: perhaps add 'among other applications' to the end of your sentence?

6. Line 102: ENMs do not need to rely solely on climatic-environmental layers. One could, if desired, add complexity by considering biotic interactions and dispersal dynamics, although this of course is easier to implement for extant species.

7. What is meant by large scale patterns? (line 107)

8. I would recommend including a sentence that says what you did and briefly why before you present the results for each analysis (since Methods are presented at the end). For example, WHY are hot spot analyses performed? How are they linked to taphonomic processes? The latter wasn't made very clear in the Methods themselves. Moreover, I struggled to understand the virtual taphofacies analyses, and their relevance, until the Discussion. You should try to make this clearer in the manuscript.

9. With regard to the hot spot analyses: how did you determine whether an area was well sampled or not? What was the cut off for these categories?

10. I am confused as to the results from the association of fossils with sed flux and surface runoff. That is, it didn't appear as if there was much of a pattern with regard to fossil occurrences and sed flux, since occurrences/clusters are associated with higher sed flux in the Campanian, and lower sed flux in the Maastrichtian?

11. Does the spatial extent of favourable virtual taphofacies in the Campanian equate to greater diversity (line 157-160)?

12. Table 1 not easy to understand, and I thought you were comparing hot spots of occurrences to sed flux and surface runoff, not occurrences in general? Also, are the Chi Squared tests significant at, say, the 0.05 or 0.01 alpha level? This should probably be included in the table.

13. Line 174-183: I think you need to briefly introduce thresholds here, for reader comprehension.

14. Line 595: state what the default search radius is here

15. Line 1057: colours represent, or color represents

16. It is still unclear how the ecospat analyses, which compare cladal niche properties through time, contribute to the manuscript? Again, I might be missing something, but what this something is should be made clearer. I don't see mention of these analyses in the paragraph summing up your results (lines 398-409), and therefore perhaps they are unnecessary?

17. There is a lot of repetition in the ENM methodology section. Language here could be tightened significantly. For example, 615-619 seems to repeat info in the beginning of the ENM section. Also, why would you project the model to outcrop area, if you trained on the entire Cretaceous (see line 613-614)? Do you mean that you masked out the outcrop area from your FULL model that you trained on the entire Cretaceous? This needs to be clarified, since projections would be unnecessary given how you currently describe your analyses.

Dear Editor and Reviewers,

Below, we respond to each individual point raised. Our modifications to the original manuscript are attached in a tracked format, and a “clean” version is also submitted. Comments from the referees here are in *italics*, while our responses are in **bold**. Quotes from the main manuscript text are bracketed.

Yours sincerely,

Alfio Alessandro Chiarenza

On behalf of all the other authors

Reviewers' comments:

Reviewer #1 (Remarks to the Author):

General Thoughts:

This revision modifies substantial aspects of the original manuscript in a positive way to alleviate confusion, increase transparency, and fix many of the methodological concerns from my first review. The hot spot sedimentological analysis in particular is much clearer and more convincing; a strong result from this work is that the limited and spatially clustered nature of available outcrop and dinosaur sampling impairs the ability to accurately estimate patterns of diversity. From this perspective, previous work citing a decrease in dinosaur diversity leading up to the K/Pg extinction is likely underestimated and even completely artefactual. However, as opposed to interpreting this issue as intractable to diversity analysis, the authors attempt to estimate diversity using ENM predicated suitable habitat area based on the justification of the species-area relationship (i.e., larger area → larger diversity). Whereas I am very comfortable with the conclusions surrounding the rock record showing an inaccurate decrease in dinosaur diversity at this time, I am still very uncomfortable with the application of ENM to infer a diversity increase at this time. Further, the ENM analyses still include some methodological concerns and the Ecospat results are not well-integrated into the overall goal of the study, making these results difficult to interpret. More detailed comments are provided below.

First of all, we would like to thank the Reviewer for their thorough and very useful review. We would like to remark that after the preceding round of comments from the Reviewers and fruitful discussion with the Editor, we have decided to drop the concept of an increase in diversity as justified by a modelled larger are of suitability (as predicted by the species-area relationship). In addition, given the similar comments on the topic by both reviewers, we decided to pull out the Ecospat analyses from the paper. Furthermore, given some further requests of clarification from Reviewer 2, we added some additional details and (hopefully) some useful simplification in the virtual taphofacies analyses. We have generally modified and restructured the whole manuscript as well to put more emphasis on the fact that what we are showing is that: 1) if there was a decline in species diversity, it was probably not due to habitat degradation because of climatic drivers; 2) a purported decline in raw diversity of dinosaurs in North America is likely due to geological biases. Lastly, we also want to highlight that we have tried to better emphasise, where relevant, issues related to uncertainty, in

particular pertaining to how our methods and analyses characterise trends or spatiotemporal characterisation of the fossil record.

1) I remain unconvinced on the point that the species-area relationship justifies the interpretation of ENM predicted habitat as informative for diversity measurements:

a. The ENMs in this analysis are trained at the clade (family) level. A clade-level model could result from several different scenarios of species-level patterns: the model could be controlled by a single, widespread taxon, it could reflect a set of species with extremely similar abiotic niche requirements, or it could reflect a summation of species each with much smaller niche dimensions. Which of these scenarios is predominant makes a difference in how these models may reflect diversity. In the first and second, a large predicted suitable habitat area says nothing about species-level diversity.

Without some sense of the generality of how clade-level niche models reflect species-level patterns (e.g., with extant clades where both the clade and species' can be tested), these results remain very hard to interpret.

b. In the text, the species-area relationship is used in conjunction with “larger habitat area = greater number of habitats” (lines 304-305). This would seem to be quite variable with latitude – i.e., an equivalent area in a temperate region has a more depauperate set of abiotic habitats than the same area in a tropical region.

c. As I mentioned before, the use of ENM to estimate changes in biodiversity has not been applied before. It seems very imprudent to apply this inferred relationship to a deep time system with no possibility for independent validation before testing this concept in a modern setting.

As reported in the previous comment, we accept the Reviewer’s concerns regarding our overly simplistic link between species richness and areal of sampling, entirely dropping this argument. We also integrated in the introduction, results and discussion, that showing areas of high suitability in currently geologically undetected areas should be addressed with care, particularly when it affects our interpretation of how environmental disruptions might have affected biotic patterns.

2) ENM methods:

a. In the revised version of this manuscript, models are trained to the entire extent of North America. As I mentioned before, this introduces a significant potential for biasing the model against any areas that have not been sampled (because the modeling algorithm treats unsampled areas as poorly suitable). This is evident by the significantly decreased predicted suitable habitat in this version of the models compared to the original manuscript. The authors had less biased models in their original version when they trained the models on sampled areas (i.e., those with outcrop) and projected these to the continental scale; I’m not sure why the change was instituted...?

b. I am confused as to why the authors projected their full continental model onto outcrop. The result should be identical to clipping the outcrop layer from the full continental model. The fact that the outcrop suitability maps (Fig 3a) are somewhat different than the full continental model (Fig 3b) concerns me – why are these results different?

c. It is important to be clear throughout the paper that the ENM results are providing “predicted” suitable habitat area. At best, models can hope to estimate the “truth” and in this case in particular, where there is no outcrop to ground truth model predictions, I think it’s important to be explicit about uncertainty.

d. It is confusing throughout the results and discussion to refer to the full continental model as “projected”. From my reading of the methods, the model was trained at a continental extent, so the North America maps reflect the training region, not a projected region?

We hopefully have now fixed this ambiguity throughout the text. Following the reviewer’s recommendation, we switched back to outcrop training → continental projection, quantifying this time (differently from the original submission) with the updated methods (and scripts) from our former resubmission. Please note that for the sake of clarity and unambiguity, we have also added to the caption of Figure 4 that: “Both sets of models have been trained with the same extent (outcrop area) but while a shows quantification in training region, plot in b shows original models projected to North America.”.

3) Ecospat tests: Whereas I applaud the application of tests of niche stability, especially in deep time where these analyses can be most informative, these tests are not well-integrated into the overall purpose of the study. The purpose seems to be to test for the “reality” of an observed diversity decline leading into the K/Pg mass extinction, particularly from the perspective of sampling biases. How does niche stability play into this goal?

It is also difficult for me to know how to interpret niche stability within a clade for the reasons described in #1a above. It seems reasonable to assume that the higher the taxonomic level, the more “stable” the composite niche dimensions will be regardless of number species...how is this information useful for the current study?

After recommendations from both Reviewers and discussion with the Editor, we decided to entirely remove this analysis and the related discussion in the current manuscript and associated content in the Supplementary Information.

Specific Comments

- Line 98: I’m not sure why the reference to endangered species is relevant? ENMs are useful for predicting species responses to environmental change in all species...? (assuming niche stability)

We originally included this as Nature Communications is a wide audience journal – so specifying it seemed useful to a broader interest audience that is potentially not familiar with ENM. However, as Reviewer 2 made a similar comment, we have removed this reference, changing the sentence to “...changes on the potential ecological niches of **taxa.”**

- Lines 108-110: ENMs are most certainly not exempt from spatial or sampling biases. As a spatial analysis, they are strongly subject to these issues!

We are aware of the kind of biases we may incur with ecological modelling, but our aim was to highlight the kind of biases in palaeodiversity analyses that we are trying to circumvent with this particular approach – hence the content of this introduction. We agree though that the former structure of the passage might have seemed to oversell the advantages of ecological niche modelling vs any other kind of palaeodiversity analyses. As remarked above, we have tried to emphasise more clearly the uncertainty related to our analyses and the record we try to investigate with them. We have also edited this passage by removing the most critical sentences, de-emphasising biodiversity references, and tightening the connection between

modelling abiotic drivers of biogeographical distribution, hopefully avoiding connotations with species richness.

- *Figure 3a: would be helpful to provide an outline of the continent so that we can see where these areas occur in space. As it is, they sort of look like they concur with the areas in Fig3b, but the model results are not identical (as I would expect) and this makes me wonder if I'm comparing the same areas between these figures.*

We thank the Reviewer for this useful suggestion, which we have followed entirely, updating Fig. 3a. Please notice that the whole Figure 3 has now been updated to the most recent results, which, following the Reviewer's recommendations, now show outcrop training → continental projections.

- *S3, figure caption for Fig S20: I have already commented on the misuse of "niche evolution" in running Ecospat or other ENM analyses through time. You simply cannot tell if niche expansion is adaptive or simply a species occupying suitable habitat that was previously inaccessible (e.g., because of constraints imposed by dispersal or biotic interactions).*

We have removed entirely the Ecospat part of the original submission.

Reviewer #2 (Remarks to the Author):

The authors have substantially improved their contribution, and I am impressed with the thoroughness of their Response to Reviewer comments. As I said before, the contribution provides an interesting perspective and commentary on potential dinosaur dynamics during the Late Cretaceous. However, I still think the manuscript would benefit from enhanced clarity in methodology and an overhaul of its focus, which I detail below:

1. I do not think you can use ENM to assess how diversity has changed through time, and I disagree with the reasoning and stated aims on lines 106-114 and 130-133. The authors rightly note (line 104) that ENMs do not consider factors such as dispersal ability and biotic interactions (if built only using climatic data and species occurrences). Therefore, one could generate a model that indicates suitable habitat in an area that a species (or clade) is not found because of barriers. For example, say a terrestrial species with poor dispersal potential originated in North America. Suitable habitat for this species may be present in Asia, but it has very little chance of dispersing there and occupying that region. Thus, you cannot model diversity or say anything about diversity using ENMs, but you can say whether or not habitat suitability for dinosaurs was declining through time. Essentially, IF dinosaur diversity was declining, your analyses can show that this decline was not because of reduction in the dinosaur species' abiotic habitat. The fact that you found that suitable habitat did not decrease is congruent with previous standardized diversity curves, and therefore provides a nice complement to previous analyses using different (and novel) methodology. Again, you cannot assess the distribution of diversity, but rather you can provide a means of assessing whether a decline in suitable habitat could have caused a decline in number of dinosaurs leading up to K/Pg. I think the focus/purpose of the manuscript needs to change, as does the language with regard to diversity/ENM. Your concluding paragraph in the Discussion was excellent (lines 398-409) and was the best summary of what you were trying to accomplish. It also presented the data and information in the correct order: I would try to re-structure your manuscript following this paragraph, which would constitute an overhaul of essentially the entire framework.

We thank the Reviewer for the useful input in the current and former submission. As highlighted in the first reply to Reviewer 1 above, we have overhauled our manuscript according to the Reviewers' criticism and other suggestions from the editor. We agree with the observation herein from the Reviewer, which were similarly pointed out by both the Editor and Reviewer 1. We have reorganised the manuscript accordingly, de-emphasising any strict link between habitat suitability and diversity, and remarking also uncertainty related to both material and methods.

2. Line 15 – consider changing 'define' to 'quantify'?

Changed accordingly.

3. Abstract in general is much improved, but see point 1!

We thank the Reviewer for the appreciative comments. We have now altered the abstract by removing the last sentence and adding: "..., and not due to a climatically-driven decrease in habitability as previously hypothesised..." in order to highlight the incongruence between a habitability increase and a hypothesis of declining dinosaur diversity driven by climate change.

4. *The authors mention in the Introduction that dinosaur bearing localities are fairly continuous and of best quality in the western half of the US (in Laramidia). But, what IS the diversity pattern if you examine this record? It should be mentioned in the Introduction. And, if it is so well sampled, then why would it not be reflective of the true diversity pattern? The authors need to make an argument for why (or why not) this fairly continuous record is reflective of true dinosaur diversity patterns, which has direct bearing on why they need to perform their current analyses (it is touched on some in the Discussion, but additional explanation needs to be up front and center when these records are first introduced).*

On this purpose we have now edited our Introduction and as a fourth paragraph of our manuscript added this passage:

"Currently, North America provides the best available sampled, accurately dated, and stratigraphically continuous record of latest Cretaceous dinosaurs⁵, and shows a decline in the numbers of genera and species from the Campanian to the Maastrichtian (Fig. 1). Taken at face value, this record implies a diversity zenith during the middle-late Campanian (~78–72 Ma), a decline in the early Maastrichtian (72–69 Ma), and a nadir in the late Maastrichtian (69–66 Ma). In the Campanian, exceptionally productive fossil localities from the Western Interior Basin (WIB), extending along a large latitudinal belt (ranging from Canada to Mexico), expose extensive, fossil-rich sedimentary successions (Fig. 1). In the Maastrichtian, on the other hand, exposures are smaller and less extensive, with optimal preservation only met in localised areas, such as the Hell Creek Formation in Montana (and lateral equivalents in Alberta, Wyoming, and the Dakotas). These relatively productive Maastrichtian localities occupy a restricted latitudinal belt (~40–50°), whilst sites at higher and lower latitudes do not meet the same ideal preservation or sampling criteria (i.e. they are generally remote places, far away from research centres, and are characterised by climatic extremes)."

5. Line 99: *ENMs are not just used to model endangered species: perhaps add 'among other applications' to the end of your sentence?*

As Reviewer 1 made a similar comment, we removed this reference, changing the sentence to “...changes on the potential ecological niches of *taxa*.”

6. Line 102: ENMs do not need to rely solely on climatic-environmental layers. One could, if desired, add complexity by considering biotic interactions and dispersal dynamics, although this of course is easier to implement for extant species.

We have changed this passage to: “Correlative ENMs can use taxonomic occurrences and climatic-environmental layers.

7. What is meant by large scale patterns? (line 107).

We have generally restructured this paragraph, and in the process, we have removed this reference to “large scale patterns”.

8. I would recommend including a sentence that says what you did and briefly why before you present the results for each analysis (since Methods are presented at the end). For example, WHY are hot spot analyses performed? How are they linked to taphonomic processes? The latter wasn't made very clear in the Methods themselves. Moreover, I struggled to understand the virtual taphofacies analyses, and their relevance, until the Discussion. You should try to make this clearer in the manuscript.

To satisfy this comment, we have modified the last paragraph of our introduction, which now is updated to this content:

“Using state-of-the-art Digital Elevation Models (DEMs³⁴) of the Cretaceous world, and results from the HadCM3L climate model (Fig. 2), we applied ENM to deduce dinosaur habitability in North America during the latest Cretaceous (Campanian–Maastrichtian [83.6–66 Ma]), and then used this to simulate and quantify modelled habitat suitability for three diverse and abundant dinosaur clades (Ceratopsidae, Hadrosauridae, and Tyrannosauridae). We then created virtual taphofacies (using taphonomically relevant physical parameters such as sediment flux and surface runoff), and identified areas suitable for potential vertebrate fossil preservation. These taphofacies were used to test statistically significant associations between these parameters and fossil ‘hotspots’, to better quantify spatial heterogeneity in the quality of the North American dinosaur fossil record, as well as changes in preservational regimes during the latest Cretaceous. These results are used herein to test the hypothesis of progressive habitat degradation as the mechanism for dinosaur diversity decline¹ in the lead-up to the K/Pg mass extinction. We also highlight the uncertainty associated with a spatially biased fossil record, as well as the physical drivers that influenced dinosaur habitat, biodiversity, and our sampling of their fossil record.”

9. With regard to the hot spot analyses: how did you determine whether an area was well sampled or not? What was the cut off for these categories?

We know already that the Campanian localities are better sampled than Maastrichtian ones, given the correlation with higher exposure and greater detected species richness (see Figure 1). Our attempt with the hotspots analysis was to highlight the clusters of fossil occurrences and investigate whether they were more localised in one stage than the other (the higher the spatial clustering, the higher the potential bias in palaeodiversity estimates as predicted by

Close et al., 2017 and Plotnick, 2017). In addition, we aimed to test a possible correlation between components of each cluster and taphonomically relevant variables. We recognise though that, as pointed out by the Reviewer in the following 3 comments, this has been done in a unclear way. We also thank their useful suggestion to use components of a cluster, instead of raw occurrences, to process correlative statistics. We then performed an optimised hotspots analyses (as implemented in ArcGIS 10.2.2 by ESRI) using the kernel density masks for raster analysis. We then tabulated these results and reported hotspots units falling in a high vs low fluxes and high vs low surface runoffs. As done in the previous resubmission, taphofacies have been defined based on Jenks natural break optimisation (via ArcGIS). We reported a quantification of our analyses on taphofacies in Table 2, which we hope is now more easily readable than the former version.

References:

1. Close, R. A., Benson, R. B. J., Upchurch, P. & Butler, R. J. Controlling for the species-area effect supports constrained long-term Mesozoic terrestrial vertebrate diversification. *Nat. Commun.* **8**, 1–11 (2017).
2. Plotnick, R. E, Recurrent hierarchical patterns and the fractal distribution of fossil localities. *Geology.* **45**, 295–298 (2017).

10. I am confused as to the results from the association of fossils with sed flux and surface runoff. That is, it didn't appear as if there was much of a pattern with regard to fossil occurrences and sed flux, since occurrences/clusters are associated with higher sed flux in the Campanian, and lower sed flux in the Maastrichtian?

Please see reply to comment 9.

11. Does the spatial extent of favourable virtual taphofacies in the Campanian equate to greater diversity (line 157-160)?

We tried to clarify this by adding in the relevant Results section that the number of significant hot spots' occurrences is greater in the Campanian for both sediment flux and surface runoff. We added the following passage to the Taphofacies results:

“It is also notable that the number of occurrences falling in significant hotspots is greater for both taphofacies in the Campanian than in than Maastrichtian (sediment flux by a ratio of 213:24, and surface runoff by a ratio of 70:42). This highlights the reduction in spatial extent of favourable taphonomic conditions, which is greater in the Campanian, enabling a more widespread preservation along the eastern coastline of the WIB, in contrast to the more localised deposits observed in the Maastrichtian.”

12. Table 1 not easy to understand, and I thought you were comparing hot spots of occurrences to sed flux and surface runoff, not occurrences in general? Also, are the Chi Squared tests significant at, say, the 0.05 or 0.01 alpha level? This should probably be included in the table.

We thank the reviewer for the indirect suggestion in using hotspots of occurrences instead of occurrences in general for our quantification (see reply to comment 9). We have also

simplified and tried to make Table 2 clearer, as recommended, adding the relative information on statistics.

13. Line 174-183: *I think you need to briefly introduce thresholds here, for reader comprehension.*

We have done this, adding the following sentence to this passage “Comparison between suitability in different time bins is reported following shared thresholds of 0.45 and 0.7; values above these thresholds are regarded as highly suitable (see Methods)”

14. Line 595: *state what the default search radius is here*

Added (10 m).

15. Line 1057: *colours represent, or color represents*

As we have simplified the taphofacies analyses, we have removed the colours, and so this correction is no longer required.

16. *It is still unclear how the ecospat analyses, which compare cladal niche properties through time, contribute to the manuscript? Again, I might be missing something, but what this something is should be made clearer. I don't see mention of these analyses in the paragraph summing up your results (lines 398-409), and therefore perhaps they are unnecessary?*

As reported in former comments, we have opted to remove the Ecospat analyses from the revised submission.

17. *There is a lot of repetition in the ENM methodology section. Language here could be tightened significantly. For example, 615-619 seems to repeat info in the beginning of the ENM section. Also, why would you project the model to outcrop area, if you trained on the entire Cretaceous (see line 613-614)? Do you mean that you masked out the outcrop area from your FULL model that you trained on the entire Cretaceous? This needs to be clarified, since projections would be unnecessary given how you currently describe your analyses.*

We have tried to make this method section simpler and less redundant. Also, given similar observations on the ambiguity between cropping/projections, we have now edited these terms throughout the manuscript. Please note that following Reviewer's 1 suggestion, we swapped back to the original experimental setting of training in outcrop and projecting in continental North American palaeogeography, trying to minimising bias. We eventually quantified suitability the same way as we've done in the former resubmission.

Information on resubmission

Number of words in the abstract of the submitted manuscript: 149

Number of words in the main text of the submitted manuscript (with headings): 5287

Number of words in the Methods of the submitted manuscript: 3007

Number of figures: 6 (included in MS file)

Number of tables: 2 (included in MS file)

Additional materials: a manuscript checklist, an original 'track changes' version of the manuscript in word, a clean version of the new resubmission, 6 Supplementary Information files, comprising of 2 tables, 1 spreadsheet with occurrence dataset, 2 scripts in text format and 19 figures (included in SI documents)

Total number of submitted files: 21

Expected number of published pages in Nature Communications: 10

We thank the editor and referees again for the thorough review job, which helped me provided a clearer explanation of our study system and results.

Yours sincerely,

Alfio Alessandro Chiarenza,

On behalf of all the co-authors.

Reviewers' Comments:

Reviewer #2:

Remarks to the Author:

Chiarenza and colleagues have revised significantly their previous contribution. This version of the manuscript focuses on changes in the extent of suitable habitat predicted by ecological niche models from the Campanian to the Maastrichtian. The authors suggest that declines in the extent of habitable area for dinosaur clades was likely not responsible for declines (if there were declines) in dinosaur diversity. This version is much improved from the previous two, although I list some outstanding concerns below:

1. Line 19: what do you mean by 'this reduction in the spatial sampling window'? Do you mean 'reduction of suitable habitat in the spatial sampling window'?
2. Line ~36: Do you mean diversification rate, or do you mean diversity? Ceratopsids and Hadrosaurids could have high diversification rates but low overall species count compared to previous Stages (that is, they could have high turnover, but low overall standing diversity). The finding by Sakamoto et al of a high diversification rate for large-bodied herbivores does not necessarily contrast with the finding of Brusatte et al.
3. Line 63: This section is confusing and needs to be reworded.
4. Line 84: 'Gains more space' is strangely phrased. Consider rewording.
5. Line 125: This sentence is unclear.
6. In general, the results are quite confusing and difficult follow. I needed to reference the figures to understand the main message. I would consider revising them thoroughly. Example suggestions include: making it clearer that lines ~165–174 refer to the training/calibration region (regions with rock outcrop), and not the projections, and making it clear how the section starting with line 186 differs from the results presented in lines 165-174. Currently they seem to be stating essentially the same thing.
7. Line 503: Are references needed here?
8. Line 1015: Incomplete sentence.
9. The authors need to be more explicit regarding some of the assumptions and caveats of their study. For example, they study suitability patterns only in North America, while dinosaurs were found globally. Global patterns may not mirror North American patterns. I think additional verbiage is also warranted on the disconnect between extent of suitable habitat and diversity. The authors test whether declines in the extent of suitable habitat could have caused declines in dinosaur diversity. They did not test whether that diversity was declining. As indicated previously, ecological modelling could suggest that regions are suitable for dinosaurs, but these areas may not have been inhabited by dinosaurs because of dispersal constraints or other biotic factors. This situation would then invalidate the conclusions of the authors.
10. The extent of the projection region within North America was unclear. Did it change through time? Did the authors include only those regions known to be terrestrial, or did they project to the entire North American continent, regardless of whether regions were marine (even though these regions could not support terrestrial dinosaurs)? This calculation will obviously affect amount of suitable habitat through time, since extent of the Western Interior Seaway, among other things, was changing over this interval (noted by the authors themselves).

Dear Editor,

Below, we respond to each individual point raised by the reviewer. Our modifications to the original manuscript are attached in a tracked format, and a “clean” version is also submitted. Comments from the reviewer here are in *italics*, while our responses are in **bold**. Quotes from the main manuscript are placed in parentheses.

Yours sincerely,

Alfio Alessandro Chiarenza

On behalf of all the other authors

REVIEWER COMMENTS:

Reviewer #2 (Remarks to the Author):

Chiarenza and colleagues have revised significantly their previous contribution. This version of the manuscript focuses on changes in the extent of suitable habitat predicted by ecological niche models from the Campanian to the Maastrichtian. The authors suggest that declines in the extent of habitable area for dinosaur clades was likely not responsible for declines (if there were declines) in dinosaur diversity. This version is much improved from the previous two, although I list some outstanding concerns below:

1. Line 19: what do you mean by ‘this reduction in the spatial sampling window’? Do you mean ‘reduction of suitable habitat in the spatial sampling window’?

We mean exactly “reduction of the spatial sampling window”, i.e. the area that we sample is reduced, meaning that the sampling potential through time is diminished by the various agents discussed in the text. We would prefer to retain this wording, as it is also quite familiar to geo-biologists, palaeontologists and other fossil record workers.

2. Line ~36: Do you mean diversification rate, or do you mean diversity? Ceratopsids and Hadrosaurids could have high diversification rates but low overall species count compared to previous Stages (that is, they could have high turnover, but low overall standing diversity). The finding by Sakamoto et al of a high diversification rate for large-bodied herbivores does not necessarily contrast with the finding of Brusatte et al.

We do mean diversification rate here, but we have clarified that it is speciation, rather than extinction, that is meant to be driving this: the Sakamoto et al. study found evidence for a slowdown in speciation rates in all dinosaur clades except ceratopsids and hadrosaurids – and so this contrasts with the Brusatte et al. study. We have therefore replaced diversification with speciation in the text.

3. Line 63: This section is confusing and needs to be reworded.

We have changed this passage from: “If the primary data that comprises the fossil record is spatially variable in its completeness, then any attempt to extract a signal from this biased

data set will tend to deliver a view of the past that is skewed towards ‘better preserved areas’.” to “Any palaeobiological investigation needs to take into account the completeness of the dataset. If the primary data that comprises the fossil record, for example, is spatially variable in its completeness, then any attempt to extract a meaningful signal from this biased data set will tend to deliver a view of the past that is artefactual. This is the case with the North American dinosaur diversity record, which is skewed towards better preserved areas.”

4. Line 84: ‘Gains more space’ is strangely phrased. Consider rewording.

The reviewer probably confused line 84 (where there is no trace of this association of words) with line 184. We have changed the passage from “The lower threshold of habitability (>0.2) gains more space in the early Maastrichtian...” to “In the lower threshold of habitability (>0.2), suitable space increases in the early Maastrichtian...”

5. Line 125: This sentence is unclear.

We have changed this passage from: “Thus, because biodiversity is spatially sensitive to abiotic constraints, ENM can provide an additional metric for understanding deep time response of organisms to environmental changes.” to “Thus, because biogeographical patterns are spatially sensitive to abiotic constraints, ENM can provide an additional metric for understanding deep time responses of organisms to environmental changes.” We also preferred to replace “biodiversity” with “biogeographical pattern” to satisfy a subsequent request to de-emphasise the link between “modelled climatic niche” and biodiversity (point 9).

6. In general, the results are quite confusing and difficult follow. I needed to reference the figures to understand the main message. I would consider revising them thoroughly. Example suggestions include: making it clearer that lines ~165–174 refer to the training/calibration region (regions with rock outcrop), and not the projections, and making it clear how the section starting with line 186 differs from the results presented in lines 165-174. Currently they seem to be stating essentially the same thing.

We have now modified this section, adding clear and unambiguous references to the relevant figures. We thank the reviewer for the recommendation and hope that these additions, particularly in the online reference system, will help the reader to more easily keep track of the results. In addition, we have added further specifications when we are discussing the training and projection regions whenever first mentioned in the results (lines 165 and 175 respectively in the former submission).

7. Line 503: Are references needed here?

We have added references to previously cited papers, plus a newly published review on this argument (Smith et al. 2019). Please note that this review advocates the advantages of supra-specific ecological niche modelling, a conclusion independently proposed by ourselves throughout this submission process.

Reference:

Smith, A. S. et al. Niche estimation above and below the species level. Trends in Ecology and Evolution. In press (2019). DOI: <https://doi.org/10.1016/j.tree.2018.10.012>.

8. Line 1015: Incomplete sentence.

We have fixed this: we have shortened the first sentence, replacing the semicolon with a full stop, added the preposition “to” after “configured”, and corrected the word “meter” with the

appropriate spelling (metre). Please note that the caption has also now been formatted, adding brackets to reference letters, to conform with Nature Communications guidelines.

9. The authors need to be more explicit regarding some of the assumptions and caveats of their study. For example, they study suitability patterns only in North America, while dinosaurs were found globally. Global patterns may not mirror North American patterns. I think additional verbiage is also warranted on the disconnect between extent of suitable habitat and diversity. The authors test whether declines in the extent of suitable habitat could have caused declines in dinosaur diversity. They did not test whether that diversity was declining. As indicated previously, ecological modelling could suggest that regions are suitable for dinosaurs, but these areas may not have been inhabited by dinosaurs because of dispersal constraints or other biotic factors. This situation would then invalidate the conclusions of the authors.

We are quite surprised that the reviewer raises these issues here, when this has been highlighted throughout the text, beginning with the Abstract and Introduction. We report here some of the arguments clarifying this for the sake of anyone consulting this peer review file. As highlighted not only by us, but also from other authors (e.g. Brusatte et al. 2015) “only North America boasts a detailed record of correlative, stratigraphically stacked faunas, in many cases accurately dated (References therein)” and in Brusatte et al (2012) “Finally, understanding the global picture of Late Cretaceous dinosaur evolution may be complicated by a traditional focus on the North American record (Reference therein), particularly the abundance and distribution of dinosaurs within a single formation (Hell Creek Formation). This is understandable, as the Hell Creek is one of the few units that globally preserves a dinosaur-dominated ecosystem and a precisely located Cretaceous–Palaeogene boundary (References therein), but may be only partially informative, if there were region-specific trends in biodiversity.”

This has relevance, not only for the research which attempted to study North American diversity trends through time analytically, but also because the availability of continuous sedimentary succession allows empirical investigation in order to test hypotheses derived from modelling investigations. We make this already quite clear for the reader since the Introduction in lines 65-66. In addition, as North America is the only latest Cretaceous continuous sedimentary succession containing the K/Pg boundary, it is also useful in tracking the faunal dynamics both on the lead-up to the End-Cretaceous mass extinction. This is the reason why an hypothesis of dinosaur demise due to habitat degradation caused by climatic changes was probably first proposed based on the North American record (again, as evidenced in our introduction and throughout the text; Sloan et al. 1986). The K/Pg boundary and the chronostratigraphic units preceding this event are unfortunately missing in many late Maastrichtian sections in Europe, Asia, and other regions of the world. In addition, some of the taxa that are majorly representative of end-Mesozoic dinosaur faunas are not present outside North America, e.g. the ceratopsids, which are only represented by a fragmentary specimen and single taxon in China and nowhere else outside western North America. We also would like to mention that even our abstract states that: “We suggest that Maastrichtian North American dinosaur diversity is therefore likely to be underestimated...”

Last, but not least, many palaeobiological researchers have recently shifted to investigating macroecological dynamics in regional areas, rather than globally, as interpretation of global patterns without considering structural changes on a regional level can provide several artefacts (Tennant, Chiarenza & Baron, 2018; Close et al., 2017).

We tried to make clear throughout the text that what we are modelling is the abiotic niche through time, and how this shows that climatic fluctuations are not to be considered responsible for long term decline in diversity before the end-Cretaceous mass extinction. We removed any ambiguity between fluctuations in habitat suitability and diversity. We still mention diversity, however, to make the point that we show that a large amount of potentially suitable habitat is missing in our palaeontological reconstructions due to fossil biases. This surely causes artefact in our reconstruction of palaeodiversity. Nonetheless, to follow the reviewer's advice, after an additional read in search of problematic statements in the text, whenever referring to "what ENM predicts", we have removed references to diversity, such as on line 125 (see point 5 above) and specified that we map potential biogeographical patterns.

References:

Brusatte, S. L. et al. The extinction of the dinosaurs. *Biol. Rev.* 90, 628–642 (2015).

Brusatte, S. L. et al. Dinosaur morphological diversity and the end-Cretaceous extinction. *Nat. Commun.* 3, (2012).

Close, R. A., Benson, R. B. J., Upchurch, P. & Butler, R. J. Controlling for the species–area effect supports constrained long–term Mesozoic terrestrial vertebrate diversification. *Nat. Commun.* 8, 1–11 (2017).

Sloan et al. Gradual dinosaur extinction and simultaneous ungulate radiation in the Hell Creek Formation. *Science* 232, 629–33 (1986).

Tennant JP, Chiarenza AA, Baron M. 2018. How has our knowledge of dinosaur diversity through geologic time changed through research history? *PeerJ* 6:e4417 <https://doi.org/10.7717/peerj.4417>

“As indicated previously, ecological modelling could suggest that regions are suitable for dinosaurs, but these areas may not have been inhabited by dinosaurs because of dispersal constraints or other biotic factors. This situation would then invalidate the conclusions of the authors.”

We strongly doubt that this would be the case, as we already know that these areas were indeed inhabited by non-avian dinosaurs, although the incompleteness due to the poor preservational regimes (also discussed by ourselves in the text) doesn't allow a proper detailed taxonomic resolution for these remains. Please note, that in some cases, the systematic position of such fragmentary remains has been referred to supra-generic taxa typically of Maastrichtian/Laramidian origin (Farke and Philips, 2017). For a review on Appalachian dinosaurs, please see Brownstein (2018). We nevertheless tried to tone down some assumptions on dispersals, as we have not directly tested this in our paper, with passages like lines 400-424: “Although we do not exclude a biotic influence of tectonic and eustatic changes on ‘true’ dinosaur diversity in tandem (e.g. allopatric speciation in the Campanian, and faunal mixing and greater area in the Maastrichtian), we contend that this played a relatively minor role when compared with their abiotic effects on observed diversity.”

After a previous reviewer's recommendation, we also remarked that we are referring to abiotic niches, habitat distribution and potential niche. Please note that the methodological caveats offered by our approach are already highlighted throughout the text (particularly in the Introduction, Results, Methods and Supplementary Information), for example for this specific case in lines 120-121. We have evidenced the uncertainty of interpreting

macroecological trends when based on a spatiotemporally biased fossil record and also our own interpretation based on our methods.

References:

Brownstein, C. D. The biogeography and ecology of the Cretaceous non-avian dinosaurs of Appalachia. *Palaeo. Elect.* 21, 1–56. (2018).

Farke AA, Phillips GE. 2017. The first reported ceratopsid dinosaur from eastern North America (Owl Creek Formation, Upper Cretaceous, Mississippi, USA) *PeerJ* 5:e3342 <https://doi.org/10.7717/peerj.3342>

10. The extent of the projection region within North America was unclear. Did it change through time? Did the authors include only those regions known to be terrestrial, or did they project to the entire North American continent, regardless of whether regions were marine (even though these regions could not support terrestrial dinosaurs)? This calculation will obviously affect amount of suitable habitat through time, since extent of the Western Interior Seaway, among other things, was changing over this interval (noted by the authors themselves).

We apologise if this basic point wasn't made clear enough in the previous submissions. We projected and quantified habitat suitability models in terrestrial only regions, as this is an important point in our interpretation: the fact that the increase of terrestrial space due to the regression of the Western Interior Seaway and Sevier Orogeny expanded the amount of terrestrial environments potentially suitable to these dinosaur clades at continental scale. This in turn reduces the amount of depositional environments potentially favourable for vertebrate fossil preservation, generating the spatial bias that we discuss in this paper, driving an artefactual "diversity decline".

To avoid any further confusion, we have specified it further in a few passages throughout the text, for example in line 175, where we added "terrestrial extent" for clarity before "of the North American continent". We have also added the same words in methods (line 613) from "and then projected it to the whole latest Cretaceous of North America" to "then projected it to the whole terrestrial extent of latest Cretaceous North America.". Please note that the same word (terrestrial) was already used in the previous submission in similar contexts through the text where it might have been ambiguous not referring to terrestrial extents (e.g. line 313: "expanse of suitable areas" to "expanse of suitable terrestrial areas" and line 353 where "when a continental projection" was changed into "when a continental terrestrial projection").

Information on resubmission

Number of words in the abstract of the submitted manuscript: 153

Number of words in the main text of the submitted manuscript (with headings): 5538

Number of words in the Methods of the submitted manuscript (with headings): 3028

Number of figures provided as vector files (.psd): 6

Number of tables: 2 (included in MS file)

Additional materials: a manuscript checklist complete and signed, an editorial policy checklist complete and signed, a reporting summary complete and signed, a point-by-point response letter to the reviewer, 4 multimedia license complete and signed (for Figures 1, 3, 4 and 6), original 'track changes' version of the manuscript in word, a clean version of the new resubmission, 1 Supplementary Information File and this cover letter with point-by-point response to the Editor.

Total number of submitted files: 18

Expected number of published pages in Nature Communications: 10

We thank the editor and referee again for the thorough review job, which helped to improve the final version of this study.

Yours sincerely,

Alfio Alessandro Chiarenza,

On behalf of all the co-authors.